# Understanding the Training and Generalization of Pre-trained Transformer for Sequential Decision Making

## Abstract

In this paper, we consider the supervised pre-trained transformer for a class of sequential decision-making problems. The class of considered problems is a subset of the general formulation of reinforcement learning in that there is no transition probability matrix; though seemingly restrictive, the subset class of problems covers bandits, dynamic pricing, and newsvendor problems as special cases. Such a structure enables the use of optimal actions/decisions in the pre-training phase, and the usage also provides new insights for the training and generalization of the pre-trained transformer. We first note the training of the transformer model can be viewed as a performative prediction problem, and the existing methods and theories largely ignore or cannot resolve an out-of-distribution issue. We propose a natural solution that includes the transformer-generated action sequences in the training procedure, and it enjoys better properties both numerically and theoretically. The availability of the optimal actions in the considered tasks also allows us to analyze the properties of the pre-trained transformer as an algorithm and explains why it may lack exploration and how this can be automatically resolved. Numerically, we categorize the advantages of pre-trained transformers over the structured algorithms such as UCB and Thompson sampling into three cases: (i) it better utilizes the prior knowledge in the pre-training data; (ii) it can elegantly handle the misspecification issue suffered by the structured algorithms; (iii) for short time horizon such as $T \leq 50$, it behaves more greedy and enjoys much better regret than the structured algorithms designed for asymptotic optimality.

## 1 Introduction

In recent years, transformer-based models [48] have achieved great success in a wide range of tasks such as natural language processing [46, 12], computer vision [20], and also reinforcement learning (RL) [13]. In particular, an offline RL problem can be framed as a sequence prediction problem [19] that predicts the optimal/near-optimal/human action based on the observed history. [13, 29] pioneer this approach with the Decision Transformer (DT), which leverages the auto-regressive nature of transformers to maximize the likelihood of trajectories in offline datasets. In this paper, we focus on a subset class of reinforcement learning problems, which we call sequential decision-making problems, that has no transition probability matrix compared to a general RL problem. This subset of problems are general enough to capture a wide range of applications including stochastic and linear bandit problems [35], dynamic pricing [16, 15], and newsvendor problem [37, 9]. Most importantly, the structure brings two benefits: (i) pre-training – the optimal action is either in closed form or efficiently solvable and thus the optimal action can be used as the prediction target in the pre-training data and (ii) understanding – the setting is more aligned with the existing study [53, 22] on the in-context learning ability of the transformer which gives more insight to the working machinery of transformer on decision-making tasks. Moreover, as a side product, our paper extends pre-trained transformers to the application contexts of business, economics, and operations research which are less studied by the existing works.

To summarize, we work on the following aspects in this paper:

- The pre-training procedure (Section 3): We mathematically formulate the learning setup of supervised pre-trained transformer for sequential decision-making problems, and propose a standardized procedure to generate the pre-training data. In addition, for the training process, we identify an out-of-distribution (OOD) issue largely ignored or unsolved by the existing literature. Specifically, the pre-training data involves two streams of actions – one is the actions in the history (features),

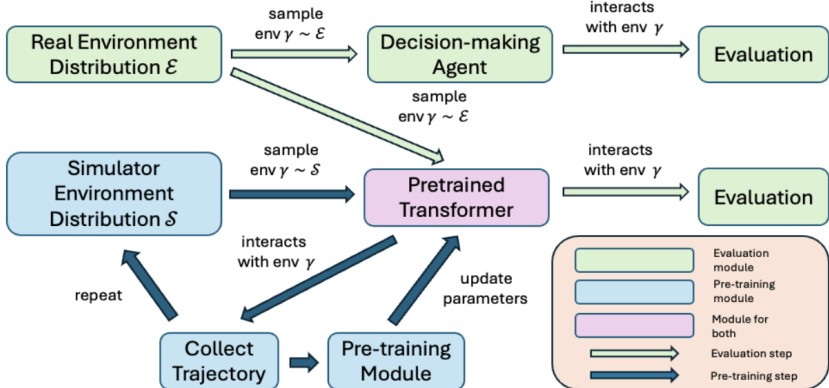

Figure 1: Comparison between the pre-trained transformer framework and traditional sequential decision-making methods (such as the structured algorithm of UCB and Thompson sampling). For traditional methods, the decision-making agent (or policy) interacts directly with a single environment sampled from the real environment distribution, focusing on exploration and exploitation within that specific environment. In contrast, the pre-trained transformer is trained across multiple environments sampled from a simulator distribution. During pre-training, the transformer collects trajectories and updates its parameters by interacting with diverse environments. Once pre-trained, the transformer functions works as an algorithm and can be applied effectively in the real environment just as the structured algorithms. But unlike the structured algorithms, the pre-trained transformer leverages the huge amount of pre-training data sampled from the simulation environment.

and the other is the action to predict (target). [38] show that when both streams are generated by one structured algorithm (say UCB or Thompson sampling), the pre-training of the transformer will result in an imitation learning procedure that imitates the structured algorithm at best. If one desires to obtain a better algorithm than the structured algorithm, like using the optimal action [36] as the target to predict in our case, this will cause the OOD issue (reflected by an $\Theta(\exp T)$ factor in the regret bound of [38]). To resolve this matter, we propose an algorithmic solution by injecting the transformer-generated action sequence into the pre-training data, and this also draws a connection with the problem of performative prediction [43].

- Understanding transformers as decision makers (Section 4): The structure of sequential decision-making problems enables the usage of optimal actions in pre-training data, and as noted earlier and will be exemplified throughout our paper, this brings benefits for both training and theoretical understanding. First, we connect the pre-trained transformer with the existing literature studying in-context learning [53, 56, 6] and we establish the pre-trained transformer as a near Bayes-optimal predictor for the optimal action in a Bayesian sense. Next, we show that (i) why the transformer model exhibits a lack of exploration and (ii) how it achieves a paramount or better performance than the structured algorithms. Other than these, an important aspect is that the problem structure of sequential decision-making renders the setup of our pre-trained transformer different from the existing works that also involve interactions with the underlying environment under the general RL setup. For example, the Online Decision Transformer (ODT) [57] lets the transformers interact with the underlying environment during the training phase, but the interactions are limited to a single environment and are designed to explore better trajectories or policies within that environment than the given offline data. We defer a more detailed comparison between the existing usage of pre-trained transformers on general RL problems and our framework to Appendix A.2.

- Numerically, when and why the pre-trained transformer is better than the structured algorithms (Section 5): While there are many structured algorithms such as UCB and Thompson sampling for each of the decision-making problems that the transformer model is applied to, the question is whether and when the transformer can achieve a better performance than these algorithms. We use numerical experiments to attribute the advantage of the pre-trained transformer into three scenarios: (i) it better utilizes the prior knowledge in the pre-training data; (ii) it can elegantly handle the misspecification issue suffered by the structured algorithms; (iii) for short time horizon such as

$T \leq 50$ or $100$, it behaves more greedy and enjoys much better regret than the structured algorithms which are designed for asymptotic optimality.

We defer more discussions on the related works to Section A.

## 2 PROBLEM SETUP

In this section, we introduce a general formulation for sequential decision-making and describe the setup of supervised pre-training. An agent (decision maker) makes decisions in a sequential environment. The full procedure is described by the sequence

$$(X_1, a_1, o_1, X_2, a_2, o_2, \ldots, X_T, a_T, o_T).$$

Here $T$ is the horizon. At each time $t = 1, \ldots, T$, the agent first observes a context vector $X_t \in \mathcal{X} \subset \mathbb{R}^d$ and takes the action/decision $a_t \in \mathcal{A}$, and then observes an outcome $o_t \in \mathcal{O} \subset \mathbb{R}^k$. Define the history

$$H_t := (X_1, a_1, o_1, \ldots, X_{t-1}, a_{t-1}, o_{t-1}, X_t),$$

and the agent's decision $a_t$ is based on $H_t$, i.e., non-anticipatory. In this light, we can model the decision as a function of $H_t$,

$$a_t = \text{TF}_\theta(H_t)$$

where $\theta$ encapsulates all the parameters of the function. That is, the agent learns from the past interactions with the underlying *environment* – tuples of $(X_t, a_t, o_t)$'s to optimize future decisions, and this process that transfers the past knowledge to the future is described by the function $\text{TF}_\theta$. The function $\text{TF}_\theta$ can take as inputs a variable-length sequence ($H_t$ can be of variable length), which is the case for the generative pre-trained transformers [46].

### 2.1 ENVIRONMENT AND PERFORMANCE METRICS

The function $\text{TF}_\theta$ describes how the actions $a_t$'s are generated. Now we describe how $X_t$'s and $o_t$'s are generated. Assume the context $X_t$'s are i.i.d. and $o_t$ follows a conditional distribution given $X_t$ and $a_t$

$$X_t \overset{\text{i.i.d.}}{\sim} \mathbb{P}_\gamma(\cdot), \; o_t \overset{\text{i.i.d.}}{\sim} \mathbb{P}_\gamma(\cdot|X_t, a_t) \tag{1}$$

where $\gamma$ encapsulates all the parameters for these two probability distributions. In particular, we emphasize that the parameter $\gamma$ is unknown to the agent.

At each time $t$, the agent collects a random reward $R_t = R(X_t, a_t)$ which depends on $X_t$, $a_t$, and also some possible exogenous randomness. The observation $o_t$ includes $R_t$ as its first coordinate. The expected reward is defined by $r(X_t, a_t) = \mathbb{E}[R(X_t, a_t)|X_t, a_t]$. Specifically, this reward function $r(\cdot, \cdot)$ is unknown to the agent and it also depends on the underlying environment $\gamma$.

The performance of the agent, namely, the decision function $\text{TF}_\theta$, is usually measured by the notion of *regret*, which is defined by

$$\text{Regret}(\text{TF}_\theta; \gamma) := \mathbb{E}\left[\sum_{t=1}^{T} r(X_t, a_t^*) - r(X_t, a_t)\right] \tag{2}$$

where the action $a_t^*$ is the optimal action that maximizes the reward

$$a_t^* := \arg\max_{a \in \mathcal{A}} r(X_t, a) \tag{3}$$

and the optimization assumes the knowledge of the underlying environment, i.e., the parameter $\gamma$. In the regret definition equation 2, the expectation is taken with respect to the underlying probability distribution equation 1 and possible randomness in the agent's decision function $\text{TF}_\theta$. We use the arguments $\text{TF}_\theta$ and $\gamma$ to reflect the dependency of the regret on the decision function $\text{TF}_\theta$ and the environment $\gamma$. Also, we note that the benchmark is defined as a dynamic oracle where $a_t^*$ maybe different over time. Regret is not a perfect performance measure for the problem, and we also do not believe the algorithm design for sequential decision-making problems should purely aim for getting a better regret bound. Despite this, it serves as a good monitor for the effectiveness of the underlying learning and the optimization procedure.

## 2.2 SUPERVISED PRE-TRAINING

In the classic study of these sequential decision-making problems, the common paradigm is to assume some structure for the underlying environment and/or the reward function, and to design algorithms accordingly. These algorithms are often known as online or reinforcement learning algorithms and are usually hard to combine with prior knowledge such as pre-training data. Comparatively, the supervised pre-training formulates the decision-making problem as a supervised learning prediction task that predicts the optimal or a near-optimal action, and thus it can utilize the vast availability of pre-training data that can be generated from simulated environments.

Specifically, the supervised pre-training first constructs a pre-training dataset

$$\mathcal{D}_{\text{PT}} := \left\{ \left( H_1^{(i)}, a_1^{(i)*} \right), \left( H_2^{(i)}, a_2^{(i)*} \right), \ldots, \left( H_T^{(i)}, a_T^{(i)*} \right) \right\}_{i=1}^n.$$

In particular, we assume the parameter $\gamma$ (see equation 1) that governs the generation of $X_t$ and $o_t$ are generated from some environment distribution $\mathcal{P}_\gamma$. Then, we first generate $n$ environments denoted by $\gamma_1, \ldots, \gamma_n$. Then there are two important aspects with regard to the generation of $\mathcal{D}_{\text{PT}}$:

- Action $a_t^{(i)*}$: For this paper, we generate $a_t^{(i)*}$ as the optimal action specified by equation 3 and this requires the knowledge of the underlying environment $\gamma_i$. A common property of all these sequential decision-making problems is that the optimal action can be easily or often analytically computed. Other option is also possible, say, $a_t^{(i)*}$ can be generated based on some expert algorithms such as Thompson sampling or UCB algorithms. In this case, the action $a_t^{(i)*} = \text{Alg}(H_t^{(i)})$ where $\text{Alg}(\cdot)$ represents the algorithm that maps from $H_t^{(i)}$ to the action without utilizing knowledge of $\gamma_i$.

- History $H_t^{(i)} = \left( X_1^{(i)}, a_1^{(i)}, o_1^{(i)}, \ldots, X_{t-1}^{(i)}, a_{t-1}^{(i)}, o_{t-1}^{(i)}, X_t^{(i)} \right)$. Note that $X_t^{(i)}$'s and $o_t^{(i)}$'s are generated based on the parameter $\gamma_i$ following equation 1. The action $a_t^{(i)}$ is generated based on some pre-specified decision function $f$ in a recursive manner:

$$a_t^{(i)} = f\left( H_t^{(i)} \right), \quad H_{t+1}^{(i)} = \left( H_t^{(i)}, a_t^{(i)}, o_t^{(i)}, X_{t+1}^{(i)} \right) \tag{4}$$

for $t = 1, \ldots, T$.

Formally, we define a distribution $\mathcal{P}_{\gamma, f}$ that generates $H_t$ and $a_t^*$ as

$$X_\tau \sim \mathbb{P}_\gamma(\cdot), \ a_\tau = f(H_\tau), \ o_\tau \sim \mathbb{P}_\gamma\left(\cdot | X_\tau, a_\tau\right), \ H_{\tau+1} = \left( H_\tau, a_\tau, o_\tau, X_{\tau+1} \right). \tag{5}$$

for $\tau = 1, \ldots, t-1$ and $a_t^*$ is specified by equation 3. The parameter $\gamma$ and the function $f$ jointly parameterize the distribution $\mathcal{P}_{\gamma, f}$. The parameter $\gamma$ governs the generation of $X_t$ and $o_t$ just as in equation 1. The decision function $f$ maps from the history to the action and it governs the generation of $a_\tau$'s in the sequence $H_t$. In this way, the pre-training data is generated with the following flow:

$$\mathcal{P}_\gamma \rightarrow \gamma_i \rightarrow \mathcal{P}_{\gamma_i, f} \rightarrow \left\{ \left( H_1^{(i)}, a_1^{(i)*} \right), \ldots, \left( H_T^{(i)}, a_T^{(i)*} \right) \right\}$$

where $f$ is a prespecified decision function used to generate the data.

Based on the dataset $\mathcal{D}_{\text{PT}}$, the supervised pre-training refers to the learning of the decision function $\text{TF}_\theta$ through minimizing the following empirical loss

$$\hat{\theta} := \arg\min_\theta \frac{1}{nT} \sum_{i=1}^n \sum_{t=1}^T l\left( \text{TF}_\theta\left( H_t^{(i)} \right), a_t^{(i)*} \right) \tag{6}$$

where the loss function $l(\cdot, \cdot) : \mathcal{A} \times \mathcal{A} \rightarrow \mathbb{R}$ specifies the prediction loss. Thus the key idea of supervised pre-training is to formulate the sequential decision-making task as a *sequence modeling* task that predicts the optimal action $a_t^{(i)*}$ given the history $H_t^{(i)}$.

In the test phase, a new environment parameter $\gamma$ is generated from $\mathcal{P}_\gamma$ and we apply the learned decision function $\text{TF}_{\hat{\theta}}$. Specifically, the procedure is described by

$$X_\tau \sim \mathbb{P}_\gamma(\cdot), \ a_\tau = \text{TF}_{\hat{\theta}}(H_\tau), \ o_\tau \sim \mathbb{P}_\gamma\left(\cdot | X_\tau, a_\tau\right), \ H_{\tau+1} = \left( H_\tau, a_\tau, o_\tau, X_{\tau+1} \right). \tag{7}$$

The only difference between the test dynamics equation 7 and the training dynamics equation 5 is that the action $a_t$ (or $a_\tau$) is generated by $\text{TF}_{\hat{\theta}}$ instead of the pre-specified decision function $f$. In this

way, we can write $(H_t, a_t^*) \sim \mathcal{P}_{\gamma, \text{TF}_\theta}$ for the test phase. Accordingly, we can define the expected loss of the decision function $\text{TF}_\theta$ under the environment specified by $\gamma$ as

$$L\left(\text{TF}_\theta; \gamma\right) \coloneqq \mathbb{E}_{(H_t, a_t^*) \sim \mathcal{P}_{\gamma, \text{TF}_\theta}} \left[ \sum_{t=1}^{T} l\left(\text{TF}_\theta\left(H_t\right), a_t^*\right) \right] \tag{8}$$

where the expectation is taken with respect to $X_t$, $o_t$ and the possible randomness in $a_t$. As before, $a_t^*$ are the optimal action defined by equation 3 and obtained from the knowledge of $\gamma$.

We defer discussions on how various sequential decision-making problems can be formulated under the general setup above and how the pre-training loss is related to the regret to Section B.

## 3 PRE-TRAINING AND GENERALIZATION

We first point out a subtle point of the supervised pre-training setting presented in the previous section. Unlike supervised learning, the expected loss equation 8 may not be equal to the expectation of the empirical loss equation 6. As noted earlier, this is because in the sequence $H_t$, the action $a_t = \text{TF}_{\hat{\theta}}(H_t)$ as in equation 7 during the test phase as opposed to that $a_t = f(H_t)$ as in equation 5 for the pertraining data $\mathcal{D}_{\text{PT}}$. Taking expectation with respect to $\gamma$ in equation 8, we can define the expected loss as

$$L(\text{TF}_\theta) \coloneqq \mathbb{E}_{\gamma \sim \mathcal{P}_\gamma} \left[ L(\text{TF}_\theta; \gamma) \right].$$

However, the expectation of the empirical loss equation 6 is

$$L_f(\text{TF}_\theta) \coloneqq \mathbb{E}_{\gamma \sim \mathcal{P}_\gamma} \left[ L_f(\text{TF}_\theta; \gamma) \right] \tag{9}$$

where

$$L_f(\text{TF}_\theta; \gamma) \coloneqq \mathbb{E}_{(H_t, a_t^*) \sim \mathcal{P}_{\gamma, f}} \left[ \sum_{t=1}^{T} l\left(\text{TF}_\theta\left(H_t\right), a_t^*\right) \right].$$

Here $f$ is the pre-specified decision function used in generating the pre-training data $\mathcal{D}_{\text{PT}}$. We note that the discrepancy between $L(\text{TF}_\theta)$ and $L_f(\text{TF}_\theta)$ is caused by the difference in the generations of the samples $(H_t, a_t^*)$, namely, $\mathcal{P}_{\gamma, \text{TF}_\theta}$ versus $\mathcal{P}_{\gamma, f}$; which essentially reduces to how the actions $a_t$'s are generated in $H_t$. In the training phase, this is generated based on $f$, while measuring the test loss requires that the actions to be generated are based on the transformer $\text{TF}_\theta$. This discrepancy is somewhat inevitable because when generating the training data, there is no way we can know the final parameter $\hat{\theta}$. Consequently, there is no direct guarantee on the loss for $L(\text{TF}_{\hat{\theta}}; \gamma)$ or $L(\text{TF}_{\hat{\theta}})$ with $\hat{\theta}$ being the final learned parameter. This out-of-distribution (OOD) issue motivates the design of our algorithm.

Algorithm 1 describes our algorithm to resolve the above-mentioned OOD issue between training and test. It consists of two phases, an early training phase and a mixed training phase. For the early training phase, the data are generated from the distribution $\mathcal{P}_{\gamma_i, f}$, while for the mixed training phase, the data are generated from both $\mathcal{P}_{\gamma_i, f}$ and $\mathcal{P}_{\gamma_i, \text{TF}_{\theta_t}}$ with a ratio controlled by $\kappa$. The decision function $f$ can be chosen as a uniform distribution over the action space as in the literature [13, 54, 36], or other more complicated options (details on the $f$ used in our experiments are provided in Appendix E.1.2). When there is no mixed phase, i.e., $M_0 = M$, the algorithm recovers that of [36]. Intuitively, the benefits of involving the data generated from $\mathcal{P}_{\gamma_i, \text{TF}_{\theta_t}}$ is to make the sequences $H_t^{(i)}$'s in the training data closer to the ones generated from the transformer. For the mixed training phase, a proportion of the data is still generated from the original $\mathcal{P}_{\gamma_i, f}$ because the generation from $\mathcal{P}_{\gamma_i, \text{TF}_{\theta_t}}$ costs relatively more time. For our numerical experiments, we choose $\kappa = 1/3$.

As far as we know, this OOD matter is largely ignored in the existing empirical works on supervised pre-training transformers. Theoretically, [36] directly assumes the pre-trained transformer learns (for discrete action space) the optimal decision function $\text{Alg}^*$ defined in the next section and develops their theoretical results accordingly. [38] show that when (i) $a_t^{(i)*}$'s in the training data are no longer the optimal actions $a_t^*$ in our setting but generated from some algorithm $\text{Alg}$ and (ii) the decision function $f = \text{Alg}$ in the distribution $\mathcal{P}_{\gamma_i, f}$, the pre-trained transformer $\text{TF}_{\hat{\theta}} \to \text{Alg}$ as the number of training samples goes to infinity. We note that this only implies that the transformer model can *imitate* an existing algorithm $\text{Alg}$. In other words, setting $f = \text{Alg}$ brings the theoretical

---

**Algorithm 1** Supervised pre-training transformers for sequential decision making

---

**Require:** Iterations $M$, early training phase $M_0 \in [1, M]$, number of training sequences per iteration $n$, ratio $\kappa \in [0, 1]$, pre-training decision function $f$, prior distribution $\mathcal{P}_\gamma$, initial parameter $\theta_0$

1: Initialize $\theta_1 = \theta_0$
2: **for** $m = 1, \ldots, M_0$ **do**
  %% Early training phase
3:   Sample $\gamma_1, \gamma_2, \ldots, \gamma_n$ from $\mathcal{P}_\gamma$ and $S_i = \left\{ \left( H_1^{(i)}, a_1^{(i)*} \right), \ldots, \left( H_T^{(i)}, a_T^{(i)*} \right) \right\}$ from $\mathcal{P}_{\gamma_i, f}$
4:   Optimize over the generated dataset $\mathcal{D}_m = \{S_i\}_{i=1}^n$ and obtain the updated parameter $\theta_{m+1}$
5: **end for**
6: **for** $m = M_0 + 1, \ldots, M$ **do**
  %% Mixed training phase
7:   Sample $\gamma_1, \gamma_2, \ldots, \gamma_n$ from $\mathcal{P}_\gamma$
8:   For $i = 1, \ldots, \kappa n$, sample $S_i = \left\{ \left( H_1^{(i)}, a_1^{(i)*} \right), \ldots, \left( H_T^{(i)}, a_T^{(i)*} \right) \right\}$ from $\mathcal{P}_{\gamma_i, f}$
9:   For $i = \kappa n, \ldots, n$, sample $S_i = \left\{ \left( H_1^{(i)}, a_1^{(i)*} \right), \ldots, \left( H_T^{(i)}, a_T^{(i)*} \right) \right\}$ from $\mathcal{P}_{\gamma_i, \mathrm{TF}_{\theta_m}}$
10:   Optimize over the generated dataset $\mathcal{D}_m = \{S_i\}_{i=1}^n$ and obtain the updated parameter $\theta_{m+1}$
11: **end for**
12: **Return**: $\hat{\theta} = \theta_{M+1}$

---

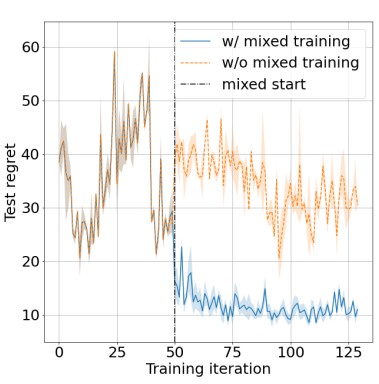
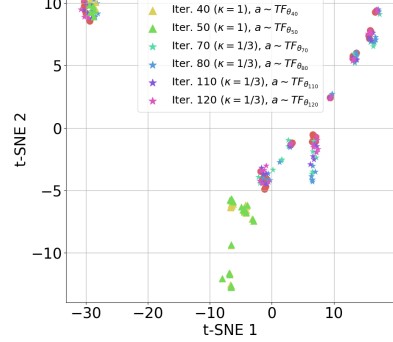

(a) Training dynamics       (b) t-SNE of the generated $H_{30}$'s

Figure 2: (a) Training dynamics. Orange: $M_0 = M = 130$. Blue: $M_0 = 50$ and $M = 130$. It shows the effectiveness of injecting/mixing the transformer-generated sequence into the training procedure. (b) A visualization of the $H_t$ with $a_\tau$'s in $H_t$ generated from various $\mathrm{TF}_{\theta_m}$. For each $\mathrm{TF}_{\theta_m}$, we generate 30 sequences. The decision function $\mathrm{Alg}^*$ is defined in the next section. We observe (i) there is a shift over time in terms of the transformer-generated action sequence, and thus the training should adaptively focus more on the recently generated sequence like the design in Algorithm 1; (ii) the action sequence gradually gets closer to the optimal decision function $\mathrm{Alg}^*$. The experiment setups are deferred to Appendix E.4.

benefits of consistency, but it excludes the possibility of using the optimal $a_t^*$ in the pre-training and thus makes it impossible for the pre-trained transformer $\mathrm{TF}_{\hat{\theta}}$ to perform better than the algorithm $\mathrm{Alg}$. When $f \neq \mathrm{Alg}$, the OOD issue is reflected by the distribution ratio in Definition 5 of [38]; the ratio can be as large as $\Theta(\exp T)$ and thus renders the regret bound to be a vacuous one on the order of $\Theta(\exp T)$. Compared to these works, our algorithm provides an algorithmic approach to resolve the OOD issue.

Algorithm 1 from an optimization perspective can be viewed as an iterative procedure to optimize

$$\mathbb{E}_{\gamma \sim \mathcal{P}_\gamma} \left[ \mathbb{E}_{(H_t, a_t^*) \sim \kappa \mathcal{P}_{\gamma, f} + (1-\kappa) \mathcal{P}_{\gamma, \mathrm{TF}_\theta}} \left[ \sum_{t=1}^T l\left( \mathrm{TF}_\theta\left( H_t \right), a_t^* \right) \right] \right]$$

where both the input data pair $(H_t, a_t^*)$ and the prediction function involve the parameter $\theta$. This falls into the paradigm of performative prediction [43, 28, 39] where the prediction model may affect the data generated to be predicted. In the literature of performative prediction, a critical matter is the instability issue which may cause the parameter $\theta_m$ to oscillate and not converge. [43] shows that the matter can be solved with strong conditions such as smoothness and strong convexity on the objective function. However, we do not encounter this instability in our numerical experiment, and we make an argument as the following claim. That is, when the underlying function class, such as $\{\text{TF}_\theta : \theta \in \Theta\}$ is rich enough, one does not need to worry about such instability.

**Claim 3.1.** The instability of a performative prediction algorithm and the resultant non-convergence does not happen when the underlying prediction function is rich enough.

The claim will be further reinforced by the optimal decision function $\text{Alg}^*$ and Proposition 4.1 in the next section. We also defer more discussions on the claim to Section D.1. While it brings a peace of mind from the optimization perspective, the following proposition justifies the mixed training procedure from a statistical generalization perspective.

**Proposition 3.2.** *Suppose $\hat{\theta}$ is determined by equation 6 where $\kappa n$ data sequences are from $\mathcal{P}_{\gamma, f}$ and $(1 - \kappa)n$ data sequences are from $\mathcal{P}_{\gamma, \text{TF}_{\tilde{\theta}}}$ for some parameter $\tilde{\theta}$. For a Lipschitz and bounded loss $l$, the following generalization bound holds with probability at least $1 - \delta$,*

$$L(\text{TF}_{\hat{\theta}}) \leq \hat{L}(\text{TF}_{\hat{\theta}}) + \sqrt{\frac{\text{Comp}(\{\text{TF}_\theta : \theta \in \Theta\})}{nT}}$$

$$+ O\left(\kappa T \mathbb{E}_{\gamma \sim \mathcal{P}_\gamma}\left[W_1\left(\mathcal{P}_{\gamma, f}, \mathcal{P}_{\gamma, \text{TF}_{\hat{\theta}}}\right)\right] + (1 - \kappa)T \mathbb{E}_{\gamma \sim \mathcal{P}_\gamma}\left[W_1\left(\mathcal{P}_{\gamma, \text{TF}_{\tilde{\theta}}}, \mathcal{P}_{\gamma, \text{TF}_{\hat{\theta}}}\right)\right] + \sqrt{\frac{\log\left(\frac{1}{\delta}\right)}{nT}}\right)$$

*where $\text{Comp}(\cdot)$ denotes some complexity measure and $W_1(\cdot, \cdot)$ is the Wasserstein-1 distance. The notation $O(\cdot)$ omits the boundedness parameters.*

The proof of Proposition 3.2 is not technical and follows the standard generalization bound [40] together with the OOD analysis [8]. We hope to use the result to provide more insights into the training procedure. The parameter $\tilde{\theta}$ can be interpreted as close to $\hat{\theta}$ as in the design of our algorithm. This will result in the term related to the distance between $\mathcal{P}_{\gamma, \text{TF}_{\tilde{\theta}}}$ and $\mathcal{P}_{\gamma, \text{TF}_{\hat{\theta}}}$ small. At the two ends of the spectrum, when $\kappa = 1$, it will result in a large discrepancy between the training and the test data and thus the right-hand-side will explode; when $\kappa = 0$, this will result in the tightest bound but it will cause more computational cost in the training procedure.

## 4 LEARNED DECISION FUNCTION AS AN ALGORITHM

In the previous section, we discuss the training and generalization aspects of the pre-trained transformer. Now, we return to the perspective of sequential decision-making and discuss the properties of the decision function when it is used as an algorithm.

In the ideal case, we define the Bayes-optimal decision function as follow

$$\text{Alg}^*(H) \coloneqq \underset{a \in \mathcal{A}}{\arg\min}\, \mathbb{E}\left[l(a, a^*(H, \gamma))|H\right]$$

$$= \underset{a \in \mathcal{A}}{\arg\min} \int_\gamma l(a, a^*(H, \gamma))\mathbb{P}(H|\gamma)\text{d}\mathcal{P}_\gamma$$

where $\mathcal{P}_\gamma$ is the environment distribution that generates the environment parameter $\gamma$, the optimal action $a^*(H, \gamma)$ depends on both the history $H$ and the environment parameter $\gamma$ (as defined in equation 3), and $\mathbb{P}(H|\gamma)$ is proportional to the likelihood of observing the history $H$ under the environment $\gamma$

$$\mathbb{P}(H|\gamma) \propto \prod_{t=1}^{\tau} \mathbb{P}_\gamma(X_t) \cdot \prod_{t=1}^{\tau-1} \mathbb{P}_\gamma(o_t|X_t, a_t)$$

where $H = (X_1, a_1, o_1, \ldots, X_{\tau-1}, a_{\tau-1}, o_{\tau-1}, X_\tau)$ for some $\tau \in \{1, \ldots, T\}$ and the distributions $\mathbb{P}_\gamma(\cdot)$ and $\mathbb{P}_\gamma(\cdot|X_t, a_t)$ are as in equation 1. In the definition, the integration is with respect to

the environment distribution $\mathcal{P}_\gamma$, and each possible environment $\gamma$ is weighted with the likelihood $\mathbb{P}(H|\gamma)$.

The following proposition relates $\texttt{Alg}^*(H)$ with the pre-training objective which justifies why it is called the *optimal* decision function. The loss function $L_f(\texttt{Alg})$ is defined as equation 9; recall that the decision function $f$ is used to generate the actions in $H_t$ and the decision function $\texttt{Alg}$ is used to predict $a_t^*$.

**Proposition 4.1.** *The following holds for any distribution $\mathcal{P}_{\gamma,f}$:*

$$\texttt{Alg}^*(\cdot) \in \underset{\texttt{Alg} \in \mathcal{F}}{\arg\min}\ L_f(\texttt{Alg}) = \mathbb{E}_{\gamma \sim \mathcal{P}_\gamma}\left[\mathbb{E}_{(H_t, a_t^*) \sim \mathcal{P}_{\gamma,f}}\left[\sum_{t=1}^{T} l\left(\texttt{Alg}\left(H_t\right), a_t^*\right)\right]\right]$$

*where $\mathcal{F}$ is the family of all measurable functions (on a properly defined space that handles variable-length inputs). In particular, we can choose the decision function $f$ properly to recover the test loss $L(\texttt{TF}_\theta)$ or the expected training loss $L_f(\texttt{TF}_\theta)$ in Section 3.*

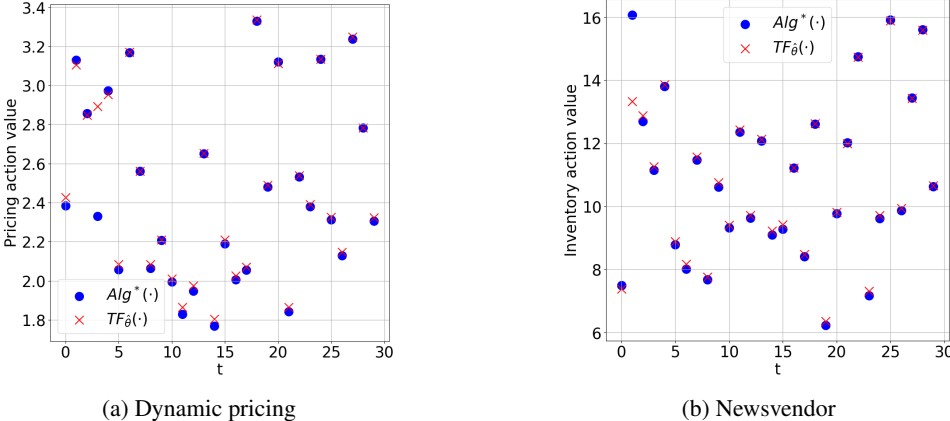

(a) Dynamic pricing          (b) Newsvendor

Figure 3: The pre-trained transformer $\texttt{TF}_{\hat{\theta}}$ well matches the optimal decision function $\texttt{Alg}^*$. For both figures, we plot decision trials for $\texttt{Alg}^*$ and $\texttt{TF}_{\hat{\theta}}$. The optimal actions change over time because $X_t$'s are different for different time $t$. The experiment setup is deferred to Appendix E.4.

Proposition 4.1 states that the function $\texttt{Alg}^*$ is one minimizer of the expected loss under any distribution $\mathcal{P}_{\gamma,f}$. To see this, $\texttt{Alg}^*$ is defined in a pointwise manner for every possible $H$. The probability $\mathcal{P}_{\gamma,f}$ defines a distribution over the space of $H$. The pointwise optimality of $\texttt{Alg}^*$ easily induces the optimality of $\texttt{Alg}^*$ for any $\mathcal{P}_{\gamma,f}$. Furthermore, this implies that $\texttt{Alg}^*$ is also the optimal solution to the expected test loss and the expected training loss. This means that if the transformer class of $\texttt{TF}_\theta$ is rich enough to cover $\texttt{Alg}^*$, and with an infinite amount of data, the supervised pre-training will result in $\texttt{Alg}^*$ at best. Thus we can interpret the property of the supervised pre-trained transformer by $\texttt{Alg}^*$ as a proxy. Such a Bayes-optimal function and the Bayesian perspective have been discussed under various settings in the in-context learning literature [53, 56, 30].

**Example 4.2.** $\texttt{Alg}^*$ behaves as (a) posterior sampling under the cross-entropy loss; (b) posterior averaging under the squared loss; (c) posterior median under the absolute loss; where the posterior distribution is with respect to $\gamma$ and it is defined by $\mathcal{P}(\gamma|H) \propto \mathbb{P}(H|\gamma) \cdot \mathcal{P}_\gamma(\gamma)$.

We defer more details for Example 4.2 to Appendix D.4. Part (a) has been directly employed as Assumption 1 in [36] for their theoretical developments. Indeed, part (a) corresponds to a discrete action space such as multi-armed bandits while part (b) and part (c) correspond to the continuous action space such as linear bandits, pricing, and newsvendor problems.

**Proposition 4.3.** *There exist a linear bandits problem instance and a dynamic pricing problem instance, i.e., an environment distribution $\mathcal{P}_\gamma$, such that the optimal decision function learned under the squared loss incurs a linear regret for every $\gamma$, i.e., $\text{Regret}(\texttt{Alg}^*; \gamma) = \Omega(T)$ for every $\gamma$.*

Proposition 4.3 states a negative result that the optimal decision function $\texttt{Alg}^*$, albeit being optimal in a prediction sense (for predicting the optimal action), does not serve as a good algorithm

for sequential decision-making problems. To see the intuition, the decision function is used recursively in the test phase (see equation 7). While it gives the best possible prediction, it does not conduct *exploration* which is critical for the concentration of the posterior measure. This highlights one key difference between the sequential decision-making problem and the regression problem [22] when applying the pre-trained transformer. For the regression problem, the in-context samples (in analogy context-action-observation tuples in $H_t$) are i.i.d. generated, and this results in a convergence/concentration of the posterior to the true model/environment $\gamma$ [23]. Yet for sequential decision-making problems, the dynamics equation 7 induces a dependency throughout $H_t$, and it may result in a non-concentration behavior for the posterior. And this is the key for constructing the linear-regret example in Proposition 4.3 and the examples are also inspired from and share the same spirit as the discussions in [24]. Such lack of exploration does not happen for discrete action space [36]; this is because for discrete action space, the output $\text{TF}_{\hat{\theta}}$ gives the posterior distribution as a distribution over the action space. The distribution output will lead to a randomized action and thus automatically perform exploration.

**Proposition 4.4.** *Consider a finite prior $\mathcal{P}_\gamma$ supported on $\Gamma = \{\gamma_1, ..., \gamma_k\}$. Suppose there exists constants $\Delta_{Exploit}$ and $\Delta_{Explore}$ such that for any $\gamma \in \Gamma$ and $H_t$, $\text{TF}_{\hat{\theta}}$ and $\text{Alg}^*(H_t)$ satisfies*

- $\mathbb{E}[r(X_t, \text{Alg}^*(H_t)) - r(X_t, \text{TF}_{\hat{\theta}}(H_t))|H_t] \leq \Delta_{Exploit}$, *where the expectation is taken with respect to the possible randomness in $\text{Alg}^*$ and $\text{TF}_{\hat{\theta}}$.*

- *The KL divergence*

$$KL\left(\mathbb{P}_\gamma(\cdot|X_t, \text{TF}_{\hat{\theta}}(H_t))\|\mathbb{P}_{\gamma'}(\cdot|X_t, \text{TF}_{\hat{\theta}}(H_t))\right) \geq \Delta_{Explore}$$

*for any $\gamma' \in \Gamma$, and the log-likelihood ratio between $\mathbb{P}_\gamma(\cdot|X_t, \text{TF}_{\hat{\theta}}(H_t))$ and $\mathbb{P}_{\gamma'}(\cdot|X_t, \text{TF}_{\hat{\theta}}(H_t))$ is $C_{\sigma^2}\Delta_{Explore}$-sub-Gaussian for any $\gamma' \in \Gamma$.*

*Then under conditions on the boundedness of the reward and environment parameters, we have*

$$\text{Regret}(\text{TF}_{\hat{\theta}}; \gamma) \leq O\left(\Delta_{Exploit}T + \frac{\log k}{\Delta_{Explore}}\right)$$

*for any $\gamma \in \Gamma$, where the notation $O(\cdot)$ omits logarithm terms in $T$ and the parameters in the boundedness assumptions.*

There are simple remedies to the linear regret behavior of the Bayes-optimal decision function, for example, using an $\epsilon$-greedy version of $\text{Alg}^*$ (and $\text{TF}_{\hat{\theta}}$). Numerically, we do not observe such behavior for any experiment even without any further modification of the pre-trained transformer $\text{TF}_{\hat{\theta}}$. Proposition 4.4 provides a simple explanation for why one does not have to be too pessimistic. Intuitively, the pre-trained model $\text{TF}_{\hat{\theta}}$ can be viewed as the optimal decision function $\text{Alg}^*$ plus some random noise, and the random noise actually plays the role of exploration and helps the posterior to concentration. Proposition 4.4 materializes such an intuition and it involves two conditions. The first condition is that $\text{TF}_{\hat{\theta}}$ behaves closely to $\text{Alg}^*$ and the (reward) deviation between these two is captured by $\Delta_{Exploit}$. This condition can be justified by Figure 3. The second condition characterizes the KL divergence between the conditional distribution of the observation $o_t$ for two environments: the true testing environment $\gamma$ and any other one $\gamma' \in \Gamma$. Note that this KL divergence critically depends on $\text{TF}_{\hat{\theta}}$ because the ability to distinguish between two environments depends on the intensity of exploration of the underlying algorithm. These conditions work more like a stylized model to understand the behavior of the transformer. It is a largely simplified model; say, the parameter $\Delta_{Explore}$ should have a dependency on $t$ for many online learning algorithms such as UCB or Thompson sampling. Nevertheless, the conditions give us a perspective to understand how the transformer works as an online algorithm. In particular, $\text{TF}_{\hat{\theta}}$ deviates from $\text{Alg}^*$ by an amount of $\Delta_{Exploit}$ in terms of the reward, and this deviation accumulates over time into the first term in the regret bound. On the other hand, this deviation also gives an exploration whose intensity is measured by $\Delta_{Explore}$, and the second term captures the regret caused during the concentration of the posterior onto the true environment. In Appendix D.7, we provide some further justifications for these conditions. We also note that the finiteness of $\Gamma$ is not essential. The KL divergence condition can be modified for a continuous $\Gamma$ and combined with a covering argument to obtain the bound accordingly.

## 5 NUMERICAL EXPERIMENTS AND DISCUSSIONS

The following figure summarizes the experiments for our model against benchmarks for each task. We note that the prior environment distribution $\mathcal{P}_\gamma$ takes an infinite support which means the environment $\gamma$ in the test phase is different from the environment $\gamma_i$'s used for training with probability 1. This provides a fair and possibly more challenging setup to examine the ability of the transformer.

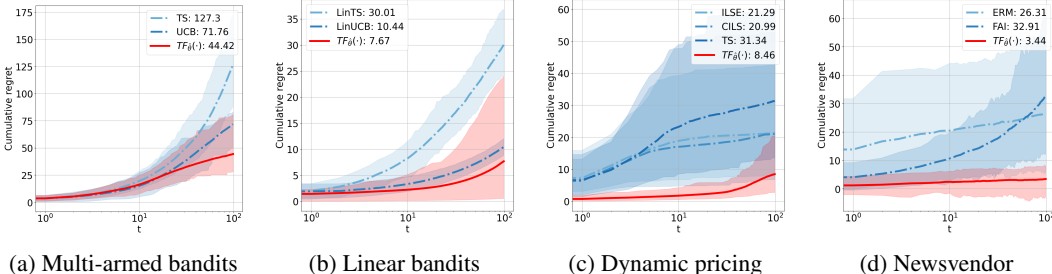

|(a) Multi-armed bandits|(b) Linear bandits|(c) Dynamic pricing|(d) Newsvendor|

Figure 4: The average out-of-sample regret of $\text{TF}_{\hat{\theta}}$ against benchmark algorithms (see details in Appendix E.3) calculated based 100 runs. The numbers in the legend bar are the final regret at $t = 100$. The shaded area in the plots indicates the 90% (empirical) confidence interval for the regrets. The prior distribution $\mathcal{P}_\gamma$ is continuous (infinitely many possible $\gamma$). The problem dimension: number of arms for MAB =20, dimension of linear bandits = 2, dimension of $X_t$ for pricing = 6, dimension of $X_t$ for newsvendor = 4.

We make the following observations on the numerical experiment. The transformer achieves a better performance than the structured algorithms for each task. This result is impressive to us in that the transformer is not replicating/imitating the performance of these benchmark algorithms but it discovers a new and more effective algorithm. This in fact shows the effectiveness of using the optimal actions $a_t^*$ as the target variable for training the transformer. Moreover, we attribute the advantage of the transformer against the benchmark algorithms to two aspects: prior knowledge from pre-training data and more greedy exploitation. First, the transformer well utilizes the pre-training data and can be viewed as tailored for the pre-training distribution. Comparatively, all the benchmark algorithms are cold-start. Second, the benchmark algorithms are all designed in ways to cater for asymptotic optimality and thus may sacrifice short-term rewards. Comparatively, the transformer behaves more greedy; just like $\text{Alg}^*$ is a decision function that does not explicitly encourage exploration. And such greedy behavior brings better reward when $T$ is small such as $T \leq 100$.

We provide more numerical experiments to Section C. In Section C.1, we present additional experiments on training dynamics, including an ablation study on $f$ and $\kappa$ as well as further results extended from Figure 2 (a). Section C.2 provides more results regarding the alignment of $\text{TF}_{\hat{\theta}}$ and $\text{Alg}^*$. Section C.3 compares the performance of $\text{TF}_{\hat{\theta}}$ and $\text{Alg}^*$, showing that $\text{TF}_{\hat{\theta}}$ can outperform $\text{Alg}^*$ in certain scenarios. In Section C.4, we demonstrate that $\text{TF}_{\hat{\theta}}$ can be a solution to model misspecifications. We also evaluate the performance of $\text{TF}_{\hat{\theta}}$ under environments with varying levels of problem complexity (Section C.5), longer testing horizons (Section C.6), and the case of model misspecification when the test environment is sampled from a distribution different from pre-training (i.e., its OOD performance) (Section C.7). Finally, we conduct an ablation study on the model architecture, model size, and task complexity in Section C.8 and Section C.9.

## 6 CONCLUSION

In this paper, we study the pre-trained transformer for sequential decision-making problems. Such problems enable using the optimal action as the target variable for training the transformer. We mathematically formulate the setup of the training pipeline and propose a way to utilize simulation environments to generate pre-training data. We investigate the training and generation aspect of the transformer and identify an OOD issue that is largely ignored by the existing literature. Theoretically, we interpret how the transformer works as an online algorithm through the lens of the Bayes-optimal decision function. The numerical experiments are encouraging in that the trained transformer learns a decision function significantly better than all the benchmark algorithms.

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

## A   RELATED WORKS

### A.1   LITERATURE REVIEW

In this section, we discuss more related works which complement our discussions in Section 1. In particular, we elaborate the literature in three streams.

**Understanding transformer and in-context learning.**

A remarkable characteristic of pre-trained transformers is their ability to perform in-context learning (ICL). Once pre-trained on a vast corpus, transformers can solve new tasks with just a few examples, without updating their parameters. ICL has captured the attention of the theoretical machine learning community, leading to considerable efforts into understanding ICL from different theoretical perspectives [53, 4, 49, 38]. A stream of literature that is related to our work explains the transformer's behavior as Bayesian inference. See [53, 56] for the Bayesian argument in regression context, [41] for approximating a large set of posteriors, and [36] for the Bayesian analysis in decision-making. Another stream of literature aims to develop generalization bound or other types of convergence results. See [30, 6] for such results in the regression context and [38] in the decision-making context.

**Pre-trained transformer for RL.**

The earliest effort of applying transformers to reinforcement learning lies in the area of offline RL [29, 13]. By autoregressively maximizing the likelihood of trajectories in the offline dataset, this paradigm essentially converts offline RL to a supervised learning problem. When evaluating the policy, the actions are sampled according to the likelihood of PT conditioned on the so called "return-to-go", which is commonly chosen to be the cumulative reward of "good" trajectories in the dataset. By doing so, PT retrieves information from trajectories in training data with similar return-to-go values and performs imitation learning to high-reward offline trajectories. Based on this observation, several works further examine the validity of return-to-go conditioning [21, 3, 11] and propose alternative methods that improve this approach or extend to other settings [34, 5, 57, 52]. For example, one of the drawbacks of the return-to-go conditioning is that it often fails in trajectory-stitching when offline data comes from sub-optimal policies. One recent work [42] finds that pre-trained LLMs, although trained by text data, can behave like a follow-the-perturbed-leader (FTPL) algorithm in bandits and games. It further introduces a regret-based training loss, through which transformers can be trained in an unsupervised way.

**Sequential decision-making.**

We carry out experiments involving some stylized sequential decision-making models including dynamic pricing and inventory management. Dynamic pricing entails setting prices to discern the underlying revenue function, usually defined as the product of demand and price [16]. Algorithms that purely exploit based on historical data tend to be suboptimal and have been found converging to non-optimal prices with a positive probability [33, 32]. Therefore, random exploration is essential for a more accurate understanding of the demand function and to avoid settling on suboptimal prices. See [24, 17, 10, 15] for algorithms that tackle this exploration-exploitation trade-off and achieving near-optimal regret under different settings.

For inventory management, demand estimation is different from that in dynamic pricing due to the censored demand [26]. When demand exceeds inventory levels, the unmet demand is not observed, leading to lost sales. The optimal decision-making policy must balance exploration and exploitation, which involves occasionally setting inventory levels higher than usual to gather more demand data, but not so high that the additional costs of exploration become prohibitive. Regarding near-optimal

policies, various works [27, 55, 2] have developed algorithms that achieve an $O(\sqrt{T})$ regret bound with different assumptions about lead times.

## A.2 TF$_{\hat{\theta}}$ V.S. ONLINE DECISION TRANSFORMER

In this section, we compare TF$_{\hat{\theta}}$ with online decision transformer (ODT) [57]. As shown in Figure 5, although both approaches involve a pre-training phase and an (online) interaction phase, they are indeed very different and designed to address different problems. Specifically:

- ODT interacts with a specific testing environment to fine-tune the policy, whereas TF$_{\hat{\theta}}$ (with Algorithm 1) interacts with different environments and does the learning in the pre-training stage. As a result, ODT is tailored to optimize performance in a single test environment, while TF$_{\hat{\theta}}$ learns various policies across different environments during pre-training, enabling it to perform well in online evaluations across different environments.

- ODT has been shown to underperform in stochastic environments. Although some RL environments—such as Atari and Gym—are relatively less stochastic, sequential decision-making often involves more stochasticity. This is why ODT has rarely been applied to sequential decision-making problems like bandit problems. More specifically, ODT, along with its precursor Decision Transformer (DT) [13], belongs to the class of "return-conditioned supervised learning" methods, which are not suitable even for offline reinforcement learning in stochastic environments. In such cases, the transformers (ODT or DT) is prone to learning from a "survival bias", as demonstrated in [11], leading to suboptimal performance.

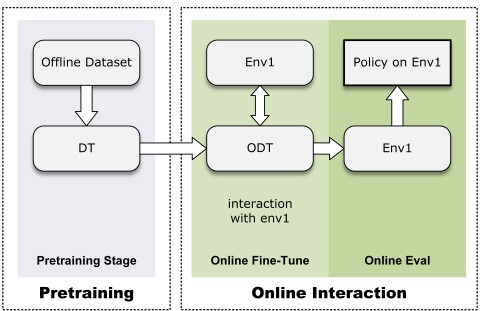
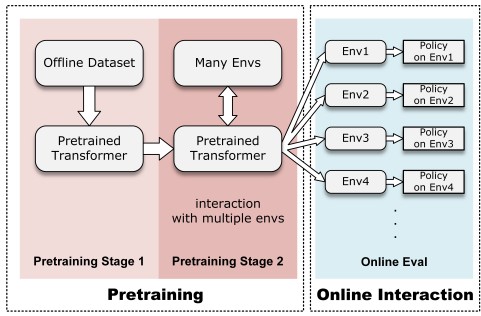

(a) Online Decision Transformer (ODT)  (b) TF$_{\hat{\theta}}$ with Algorithm 1 (ours)

Figure 5: Comparison between training frameworks of Online Decision Transformer (ODT) [57] and TF$_{\hat{\theta}}$. The online interactions of ODT is in the (single) test environment to explore a (single) good policy in the same environment, while our proposed training framework (Algorithm 1) has online interactions in multiple environments to mitigate the OOD issue in the pre-training data (as discussed in Section 3), and TF$_{\hat{\theta}}$ can handle different test environments.

We further conduct an experiment comparing our framework with ODT as shown in Figure 6 to highlight their differences. Specifically, we use a noiseless linear bandits task (i.e., no reward noise) with dimension $d = 2$ and a horizon of $T = 20$, where the test environments are entirely unseen during the training phase, including the online fine-tuning phase for ODT. It is important to emphasize that this problem setting differs from the one for which ODT is designed (as shown in Figure 5(a), where the test environment is identical to the environment used during online fine-tuning), and aligns with the setting we study in this work. We set fair configurations for training ODT and TF$_{\hat{\theta}}$ (which will be specified later). As expected, Figure 6 demonstrates that ODT fails in this setting, performing no better than a random policy that uniformly samples actions.

**Setup**: for both ODT and TF$_{\hat{\theta}}$, we set the transformer architecture with 4 layers, 4 heads, 512 embedding dimensions, and a 20-length context window, which are the same as in [57]. For the offline dataset, we sample 100,000 environments by the same method in our paper, and then use Thompson sampling to generate trajectories for each environment. And for the online interactions, the environments are sampled from the same distribution as the offline data. The testing environments are still sampled from the same distribution. We set the offline training phase (pre-training stage for



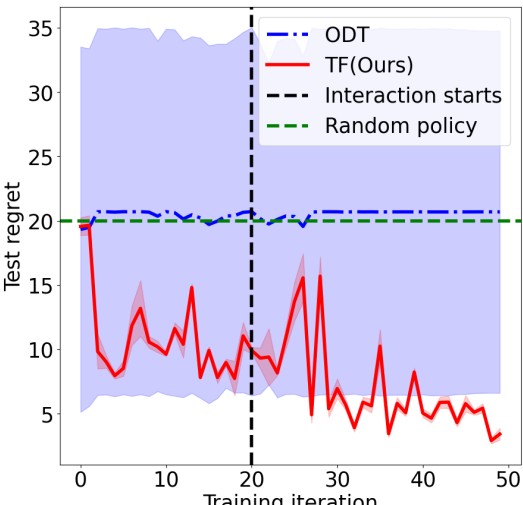

Figure 6: We test the ODT and our method in a same linear bandits task, where test environments are unseen before the evaluation phase. The ODT fails in the test environments, which are different from the online fine-tuning phase's.

ODT and stage 1 for $\texttt{TF}_{\hat{\theta}}$) with 20 iterations (each iteration has 500 updates/gradient descents on parameters), and set the online interaction phase (online fine-tuning for ODT and stage 2 for $\texttt{TF}_{\hat{\theta}}$) with 30 iterations (each iteration has 50 updates/gradient descents on parameters). The other training parameters (e.g., learning rate) are set the same as in [57] except the return-to-go (RTG): we set the eval RTG as 55, which is the maximum RTG observed in the offline dataset, and online RTG as 110, which doubles the eval RTG as used in [57].

## B    PROBLEM EXAMPLES FOR THE GENERAL SETUP IN SECTION 2

Throughout the paper, we make the following boundedness assumption.

**Assumption B.1.** Assume the context space $\mathcal{X}$, the action space $\mathcal{A}$, and the observation space $\mathcal{O}$ is bounded (under the Euclidean norm) by $D$.

We provide a few examples as special cases of the framework in Section 2. We hope these examples make the actions, observations, and context in the general setup more tangible.

- Stochastic multi-armed bandits : There is no context, $X_t = $ null for all $t$. The action $a_t \in \mathcal{A} = \{1, 2, \ldots, k\}$ denotes the index of the arm played at time $t$. The observation (also the random reward) is generated following the distribution $P_{a_t}$. The parameter $\gamma$ encapsulates the distributions $P_1, \ldots, P_k$.

- Linear bandits: There is no context, $X_t = $ null for all $t$. The action $a_t \in \mathcal{A} \in \mathbb{R}^d$ is selected from some pre-specified domain $\mathcal{A}$. The random reward $R(X_t, a_t) = w^\top a_t + \epsilon_t$ where $\epsilon_t$ is some noise random variable and $w$ is a vector unknown to the agent. The observation $o_t = R(X_t, a_t)$ and the expected reward $r(X_t, a_t) = \mathbb{E}\left[R(X_t, a_t) | X_t, a_t\right] = w^\top a_t$. The parameter $\gamma$ encapsulates the vector $w$ and the noise distribution.

- Dynamic pricing: The context vector $X_t$ describes the market-related information at time $t$. Upon the reveal of $X_t$, the agent takes the action $a_t$ as the pricing decision. Then the agent observes the demand $d_t = d(X_t, a_t) + \epsilon_t$ where $\epsilon_t$ is some noise random variable and $d(\cdot, \cdot)$ is a demand function unknown to the agent. The random reward $R_t(X_t, a_t) = d_t \cdot a_t$ is the revenue collected from the sales at time $t$ (which equals the demand times the price), and the observation $o_t = d_t$. The expected reward $r(X_t, a_t) = \mathbb{E}\left[R(X_t, a_t) | X_t, a_t\right] = d(X_t, a_t) \cdot a_t$. The parameter $\gamma$ governs the generation of $X_t$'s and also contains information about the demand function $d$ and the noise distribution.

- Newsvendor problem: As the dynamic pricing problem, the context vector $X_t$ describes the market-related information at time $t$. Upon its reveal, the agent takes the action $a_t$ which represents the number of inventory prepared for the sales at time $t$. Then the agent observes the demand $d_t = d(X_t, a_t) + \epsilon_t$ where $\epsilon_t$ is some noise random variable and $d(\cdot, \cdot)$ is a demand function unknown to the agent. The observation $o_t = d_t$, and the random reward is the negative cost $R(X_t, a_t) = -h \cdot (a_t - d_t)^+ - l \cdot (d_t - a_t)^+$ where $(\cdot)^+$ is the positive-part function, $h$ is the left-over cost, and $l$ is the lost-sale cost. The expected reward $r(X_t, a_t) = \mathbb{E}[R(X_t, a_t)|X_t, a_t]$ with the expectation taken with respect to $\epsilon_t$. The parameter $\gamma$ governs the generation of $X_t$'s and encodes the demand function $f$ and the noise distribution.

For the general problem setup in Section 2, the following proposition relates the pre-training and the regret through the lens of loss function.

**Proposition B.2** (Surrogate property). *We say the loss function $l(\cdot, \cdot)$ satisfies the surrogate property if there exists a constant $C > 0$ such that*

$$\text{Regret}(f; \gamma) \leq C \cdot L(f; \gamma)$$

*holds for any $f$ and $\gamma$. We have*

*(a) The cross-entropy loss satisfies the surrogate property for the multi-armed bandits problem.*

*(b) The squared loss satisfies the surrogate property for the dynamic pricing problem.*

*(c) The absolute loss satisfies the surrogate property for the linear bandits and the newsvendor problem.*

Proposition B.2 gives a first validity of the supervised pre-training approach. The surrogate property is a property of the loss function and it is not affected by the distribution or the decision function $\text{TF}_\theta$. Specifically, it upper bounds the regret with the action prediction loss. It is also natural in that a closer prediction of the optimal action gives a smaller regret. One implication is that the result guides the choice of the loss function for different underlying problems.

### B.1 PROOF OF PROPOSITION B.2

To prove Proposition B.2, we assume the following mild conditions. First, for all these problems, the noise random variable $\epsilon_t$ has mean 0, is i.i.d. across time, and is independent of the action $a_t$. Second, for the linear bandits problem, the $L_\infty$ norm of the unknown parameter $w$ is bounded by $D$. Third, for the dynamic pricing problem, we assume the demand function is differentiable with respect to the action coordinate, and both the absolute value of the first- and second-order derivatives is bounded by $D$. Further details regarding the problem formulations can be found in Appendix E.2.

*Proof.* Since both the regret $\text{Regret}(f; \gamma)$ and the prediction error $L(f; \gamma)$ are the sums of single-step regret and prediction error over the horizon, it is sufficient to prove the single-step surrogate property of the loss function. Therefore, in the following proof, we will focus on proving the single-step surrogate property and omit the subscript $t$ for simplicity.

- For the multi-arm bandits problem, without loss of generality, we assume $\mathcal{A} = \{a_1, \ldots, a_J\}$ and the optimal arm is $a_1$. For $1 \leq i \leq J$, let $\mu_j$ be the mean of the reward distribution for arm $j$, and define $\Delta_j = \mu_1 - \mu_j$ as the reward gap for arm $j$. Define $\Delta_{\max} = \max\{\Delta_1, \cdots, \Delta_J\}$ as the maximum action gap. Under cross-entropy loss, we denote the output distribution of decision function $f$ is $(p_1, \ldots, p_J)$. Then the single-step regret can be bounded by:

$$\mathbb{E}[r(X, a^*) - r(X, f(H))|H] = \mu_1 - \sum_{j=1}^{J} p_j \mu_j = \sum_{j=2}^{J} p_j \Delta_j \leq \Delta_{\max}(1 - p_1) \leq -\Delta_{\max} \cdot \log(p_1),$$

where the last step is by the inequality $1 - x \leq -\log(x)$ for $0 < x \leq 1$. Then by the definition of cross-entropy loss, we finish the proof.

- For the dynamic pricing problem, we have

$$r(X, a^*) - r(X, f(H)) = \nabla_a r(X, a^*)(a^* - f(H)) + \nabla_a^2 r(X, \tilde{a})|f(H) - a^*|^2 \leq D|f(H) - a^*|^2$$

where $\tilde{a}$ is in the line of $f(H)$ and $a^*$. Here, the first step is by the Taylor expansion, and the second step is by the first order condition of $a^*$ and the assumption. Then by noting $|f(H) - a^*|^2$ is the square loss, we finish the proof.

- For the linear bandits problem, we have

$$r(X, a^*) - r(X, f(H)) = w^\top a^* - w^\top f(H) \leq ||w||_\infty ||f(H) - a^*||_1 \leq D||f(H) - a^*||_1,$$

where the second step is by the Holder's inequality and the last step is by the assumption. Then by noting $||f(H) - a^*||_1$ is the absolute loss, we finish the proof.

- For the newsvendor problem, we have

$$r(X, a^*) - r(X, f(H)) \leq \max\{h, l\} \, \mathbb{E}_X \left[ |a^* - d(X) - f(H) + d(X)| \right]$$
$$= \max\{h, l\} \, |a^* - f(H)|$$

where the first line is because the (random) reward function is $\max\{h, l\}$-Lipschitz in $a - d(X)$. Then by noting $|f(H) - a^*|$ is the absolute loss, we finish the proof.

$\square$

## C  MORE EXPERIMENTS

### C.1  MORE EXPERIMENTS ON TRAINING DYNAMICS

#### C.1.1  ABLATION STUDY ON $f$ AND $\kappa$

In this section, we present ablation studies to explore the impact of two key factors: the decision function $f$ used for generating pre-training data, and the mix ratio $\kappa$ applied in Algorithm 1. Figure 7 summarizes the results, where we pre-train and test models in multi-armed bandits problems defined in Appendix E.2, and only tune $f$ or $\kappa$ during the pre-training and keep all other parameters the same as in Appendix E.1.2.

- **Effect of $f$**: We evaluate the influence of different decision functions $f$ on the performance of the transformer in a multi-armed bandit task. Three types of decision functions are considered: the one we designed, as defined in Appendix E.1.2 , the UCB algorithm for multi-armed bandits [35], and a 50/50 mixture of both. The training dynamics for each function are shown in Figure 7a. We observe that all three approaches achieve similar final performance once training ends. While it is possible that some decision functions could lead to faster convergence or better performance after pre-training, we did not encounter any training failures due to the choice of $f$.

- **Effect of $\kappa$**: We also conduct an experiment to numerically investigate the impact of different mix ratios $\kappa$. The results are presented in Figure 7b. We test four values of $\kappa$ $(10\%, 33\%, 66\%, 90\%)$, keeping all other hyperparameters constant for a multi-armed bandit task. The findings show that transformer's performance remains robust across different values of $\kappa$, except when $\kappa = 0.9$, where performance declines. This supports our earlier discussion in Algorithm 1, highlighting the importance of mixing a non-trivial amount of self-generated data (i.e., non-trivial value for $1 - \kappa$).

#### C.1.2  ADDITIONAL RESULTS ON FIGURE 2 (A)

In addition to Figure 2 (a), we further evaluate the effectiveness of Algorithm 1 across various tasks and architectures, as illustrated in Figure 8.

Figure 8 presents comparisons of the transformer's testing regret (see definition and implementation in Appendix E.1) with and without self-generated data (i.e., using Algorithm 1 or not) during the training, colored by blue and orange respectively. The comparisons are conducted on (a) linear

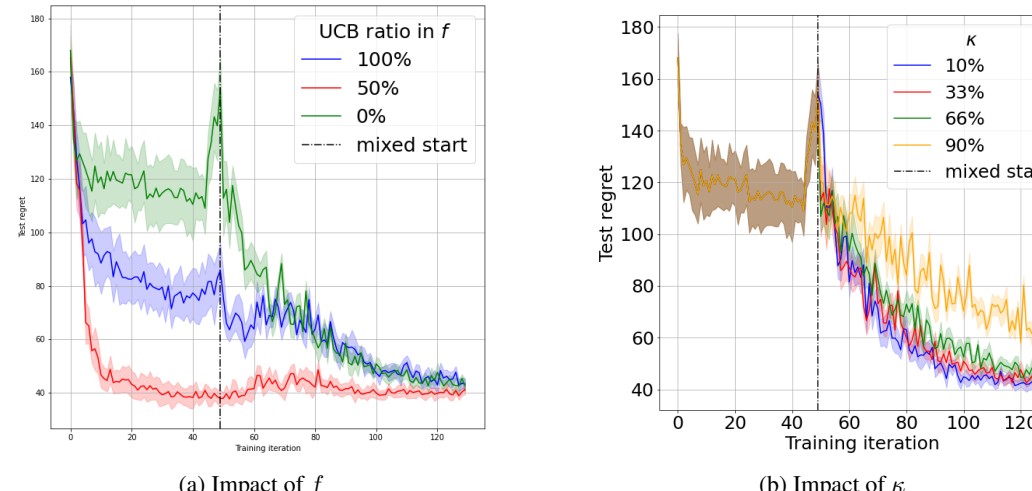

(a) Impact of $f$                           (b) Impact of $\kappa$

Figure 7: (a) The effect of $f$ (decision function for generating the pre-training data), where we mix the UCB algorithm [35] and decision function designed in Appendix E.1.2 with different ratios to create the pre-training data. (b) The effect of $\kappa$ (ratio of samples generated by $f$ in the mixed training phase).

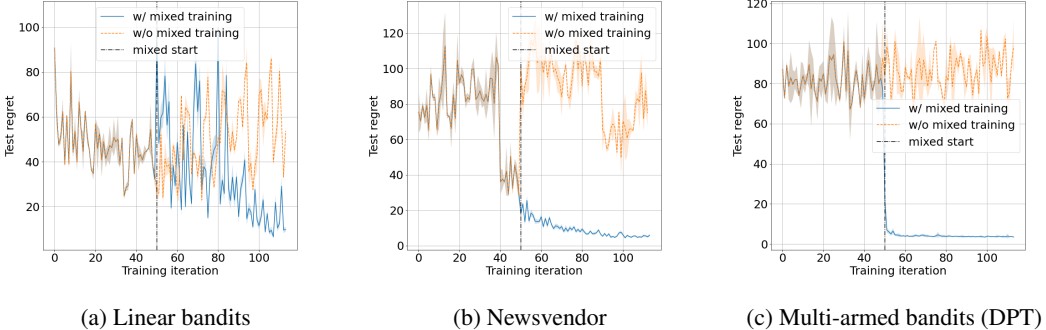

(a) Linear bandits              (b) Newsvendor              (c) Multi-armed bandits (DPT)

Figure 8: Effectiveness of Algorithm 1. The values are averaged out-of-sample regrets based on 128 runs, where the shaded area indicates the standard deviation.

bandits and (b) newsvendor tasks. Additionally, (c) includes a similar comparison conducted in a model with different architecture, the Decision Pre-trained Transformer (DPT) [36], on the multi-armed bandits task. The results consistently demonstrate the same pattern observed in Figure 2 (a): incorporating transformer-generated data significantly reduces the testing regret loss at each iteration $m > 50$ compared to not using it. This shows the effectiveness of Algorithm 1 (i) across different tasks and (ii) for various model architectures.

**Setup.** In Figure 8 (a) and (b), we consider two tasks: a linear bandits task (with 2-dimensional action space) and a newsvendor task (with 4-dimensional contexts). Both tasks have infinite support of the prior environment distribution $\mathcal{P}_\gamma$ (i.e., infinite possible environments in both the pre-training and testing). The details of the tasks can be found in Appendix E.2. For each task, we independently train two transformer models by Algorithm 1. For both models, the training parameters are identical except for the $M_0$: the first curve (blue, thick line) uses $M_0 = 50$, while the second curve (orange, dashed line) uses $M_0 = 130$, meaning no transformer-generated data is utilized during training. All other parameters follow the configuration detailed in Appendix E.1. The figure shows the mean testing regret at each training iteration across 128 randomly sampled environments from $\mathcal{P}_\gamma$, with the shaded areas representing the standard deviation. We conduct the same experiment on the multi-armed bandits task (with infinite possible environments and 20 arms) for a different architecture for

sequential decision making, Decision Pre-trained Transformer (DPT) [36], in Figure 8 (c). We refer to the original paper for details on the DPT's architecture.

## C.2 MORE EXPERIMENTS ON MATCHINGS OF $\text{TF}_{\hat{\theta}}$ AND $\text{ALG}^*$

This section provides more results regarding the matching of $\text{TF}_{\hat{\theta}}$ and $\text{Alg}^*$ across different tasks: Figure 9 provides more examples comparing decisions made by $\text{TF}_{\hat{\theta}}$ and $\text{Alg}^*$ at sample path levels, while Figure 10 compares them at population levels.

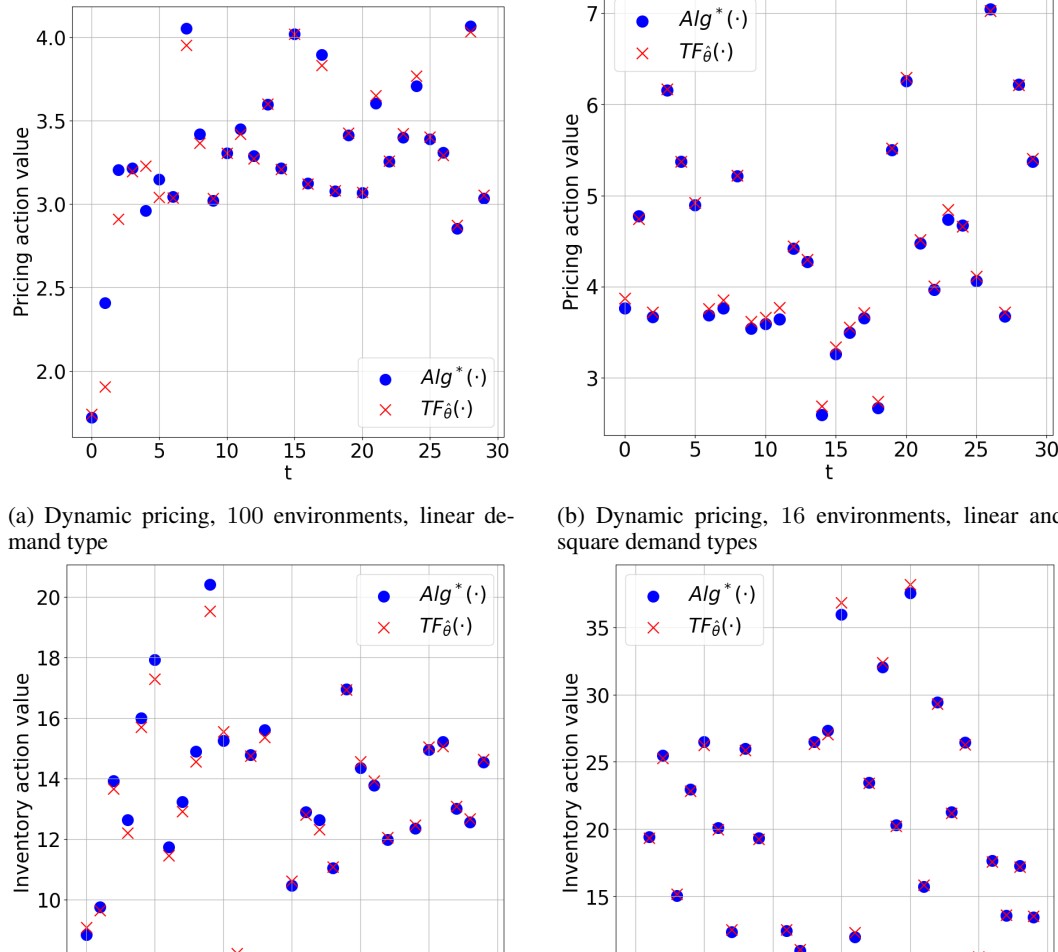

(a) Dynamic pricing, 100 environments, linear demand type

(b) Dynamic pricing, 16 environments, linear and square demand types

(c) Newsvendor, 100 environments, linear demand type

(d) Newsvendor, 16 environments, linear and square demand types

Figure 9: Examples to compare the actions from the transformer $\text{TF}_{\hat{\theta}}$ and the optimal decision function $\text{Alg}^*$.

We plot the actions generated from $\text{TF}_{\hat{\theta}}$ and $\text{Alg}^*$ on the same environment with the same contexts in Figure 9 across dynamic pricing and newsvendor tasks. Each subfigure has a different number of possible environments or possible demand types (which are shared in both the pre-training and testing phases). The population-level differences $\{\text{TF}_{\hat{\theta}}(H_t) - \text{Alg}^*(H_t)\}_{t=1}^T$ are presented in Figure 10. We observe that across different tasks: (1) $\text{TF}_{\hat{\theta}}$ nearly matches $\text{Alg}^*$, but (2) the matchings are not perfect. The actions from $\text{TF}_{\hat{\theta}}$ and $\text{Alg}^*$ are not exactly the same, and their differences increase

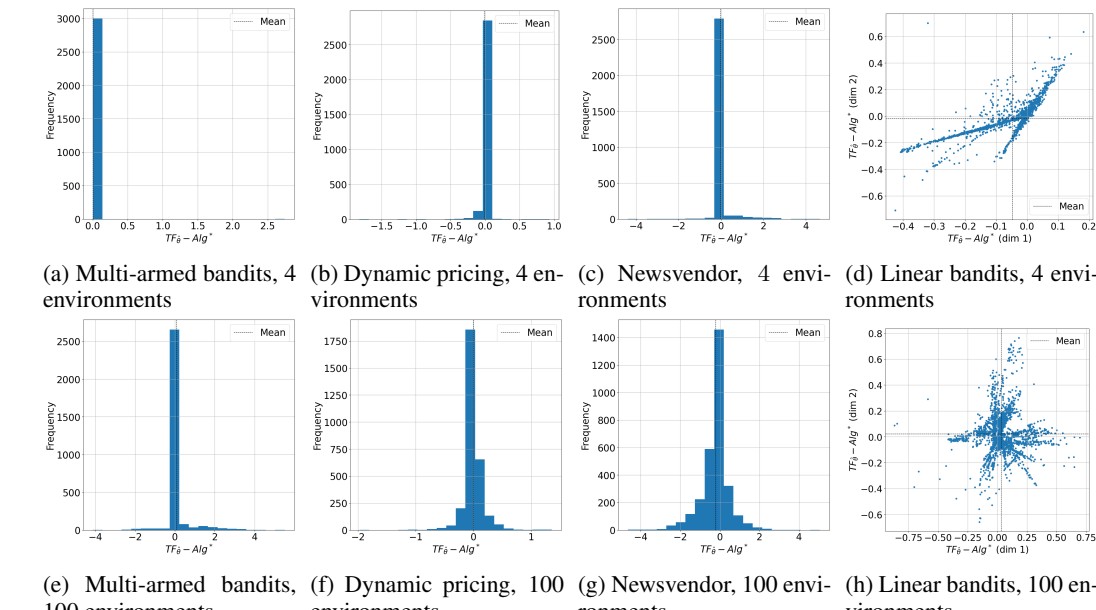

(a) Multi-armed bandits, 4 environments
(b) Dynamic pricing, 4 environments
(c) Newsvendor, 4 environments
(d) Linear bandits, 4 environments

(e) Multi-armed bandits, 100 environments
(f) Dynamic pricing, 100 environments
(g) Newsvendor, 100 environments
(h) Linear bandits, 100 environments

Figure 10: $\text{TF}_{\hat{\theta}}(H_t) - \text{Alg}^*(H_t)$ across different tasks with various numbers of possible environments. More possible environments lead to a harder decision-making problem.

as the underlying problems become more complex, as seen by comparing the first and second rows in Figure 10. This latter observation numerically supports the conditions in Proposition 4.4.

**Setup.** For Figure 9, each subfigure is based on a sampled environment with a sampled sequence of contexts $\{X_t\}_{t=1}^{30}$ from the corresponding task. For tasks in (a),(c), the data generation process follows the description detailed in Appendix E.2, and the architecture of $\text{TF}_{\hat{\theta}}$ and the definition of $\text{Alg}^*$ can be found in Appendix E.1 and Appendix E.3 respectively. For tasks in (b), (d), there are 16 environments included in the support of $\mathcal{P}_\gamma$, where 8 environments have demand functions of the linear type and the other 8 have demand functions of the square type (see Appendix E.2 for the definitions of these two types). Both the pre-training and testing samples are drawn from $\mathcal{P}_\gamma$, i.e., from these 16 environments. The remaining setups for these tasks follow the methods provided in Appendix E.2.

For each task shown in Figure 10, we generate 100 sequences of $\{\text{TF}_{\hat{\theta}}(H_t) - \text{Alg}^*(H_t)\}_{t=1}^{100}$, i.e., each subfigure is based on 10000 samples of $\text{TF}_{\hat{\theta}}(H_t) - \text{Alg}^*(H_t)$. For dynamic pricing and newsvendor, actions are scalars and thus $\text{TF}_{\hat{\theta}}(H_t) - \text{Alg}^*(H_t)$ is also a scalar; For multi-armed bandits, the action space is discrete and we use the reward difference associated with the chosen actions (by $\text{TF}_{\hat{\theta}}(H_t)$ or $\text{Alg}^*(H_t)$) as the value of $\text{TF}_{\hat{\theta}}(H_t) - \text{Alg}^*(H_t)$. We use histograms to summarize the samples from them. For linear bandits, the actions are 2-dimensional and thus we use the scatters to show $\text{TF}_{\hat{\theta}}(H_t) - \text{Alg}^*(H_t)$ in Figure 10 (d) or (h).

### C.3 REGRET COMPARISON WITH ALG*

In this subsection, we study the behavior of $\text{TF}_{\hat{\theta}}$ by comparing regret performances on $\text{TF}_{\hat{\theta}}$ and $\text{Alg}^*$, which utilizes the posterior distribution $\mathcal{P}(\gamma|H)$ of environments. As shown in Section 4, $\text{Alg}^*$'s are defined as the Bayes-optimal decision functions.

Algorithm 2 presents a general framework for the Bayes-optimal decision functions/algorithms in Example 4.2: posterior averaging, sampling, and median. For simplicity of notation, we omit the subscript $t$ and only consider a finite space $\Gamma$ of environments, while the algorithms can be easily extended to the infinite case. For the posterior median algorithm, we assume the action space $\mathcal{A} \subseteq \mathbb{R}$.

Figure 11 presents the testing regret and action suboptimality for $\text{TF}_{\hat{\theta}}$ compared to the Bayes-optimal decision functions/algorithms. Generally, all algorithms exhibit good regret performance. However,

---

**Algorithm 2** Posterior averaging/sampling/median at time $t$

---

**Require:** Posterior probability $\mathcal{P}(\gamma|H)$ and optimal action $a_\gamma^*$ for each environment $\gamma \in \Gamma$, algorithm $\text{Alg} \in \{$posterior averaging, posterior sampling, posterior median$\}$
    %% Posterior averaging
1: **if** $\text{Alg}=$posterior averaging **then**
2:      $a = \sum_{\gamma \in \Gamma} \mathcal{P}(\gamma|H) \cdot a_\gamma^*$
    %% Posterior sampling
3: **else if** $\text{Alg}=$posterior sampling **then**
4:      $a = a_{\tilde{\gamma}}^*$, where $\tilde{\gamma} \sim \mathcal{P}(\cdot|H)$
    %% Posterior median
5: **else if** $\text{Alg}=$posterior median **then**
6:      Sort and index the environments as $\{\gamma_i\}_{i=1}^{|\Gamma|}$ such that the corresponding optimal actions are in ascending order: $a_{\gamma_1}^* \leq a_{\gamma_2}^* \leq \ldots \leq a_{\gamma_{|\Gamma|}}^*$.
7:      Choose $a = \min \left\{ a_{\gamma_i}^* \,\middle|\, \sum_{i'=1}^{i} \mathcal{P}(\gamma_{i'}|H) \geq 0.5 \right\}$
8: **end if**
9: **Return:** $a$

---

we observe that $\text{TF}_{\hat{\theta}}$ is (i) superior to posterior sampling in multi-armed bandits, as shown in (a), and (ii) outperforms posterior averaging in linear bandits, as shown in (c).

Specifically, (b) shows that posterior sampling has higher action suboptimality during the initial time steps compared to $\text{TF}_{\hat{\theta}}$. This suggests that the exploration inherent in the sampling step of posterior sampling can introduce additional regret, which might be unnecessary for simpler problems. Conversely, (d) indicates that the decisions from posterior averaging do not converge to the optimal action $a_t^*$ as quickly as those from $\text{TF}_{\hat{\theta}}$ and posterior sampling. This suggests that posterior averaging may be too greedy, thus failing to sufficiently explore the environment. This second observation also numerically supports Proposition 4.3.

These observations illustrate that $\text{TF}_{\hat{\theta}}$ can outperform Bayes-optimal decision functions in certain scenarios. This further demonstrates that the advantage of $\text{TF}_{\hat{\theta}}$ over benchmark algorithms is not solely due to its use of prior knowledge about the task. Rather, $\text{TF}_{\hat{\theta}}$ discovers a new decision rule that achieves better short-term regret than oracle posterior algorithms. It can be more greedy than posterior sampling while exploring more than posterior averaging.

**Setup.** We consider a multi-armed bandits task (with 20 arms) and linear bandits task (with 2-dimensional actions) with 4 environments. The results are based on 100 runs. The details of tasks with their specific oracle posterior algorithms are provided in Appendix E.2 and Appendix E.3.

## C.4    $\text{TF}_{\hat{\theta}}$ AS A SOLUTION TO MODEL MISSPECIFICATION

Most sequential decision-making algorithms, including all the benchmark algorithms in our experiments, rely on structural or model assumptions about the underlying tasks. For instance, demand functions in pricing and newsvendor problems are typically assumed to be linear in context. Applying these algorithms in misspecified environments, where these assumptions do not hold, can lead to degraded performance.

In contrast, $\text{TF}_{\hat{\theta}}$ offers a potential solution to model misspecifications. By generating pre-training samples from all possible types of environments (e.g., newsvendor models with both linear and non-linear demand functions), $\text{TF}_{\hat{\theta}}$ leverages its large capacity to make near-optimal decisions across different types. Figure 12 demonstrates the performance of $\text{TF}_{\hat{\theta}}$ trained on tasks with two types of demand functions: linear and square (definitions provided in Appendix E.2). It compares $\text{TF}_{\hat{\theta}}$ with benchmark algorithms designed solely for linear demand cases. The performance is tested in environments where demand functions (i) can be either type with equal probability (first column); (ii) are strictly square type (second column); and (iii) are strictly linear type (third column). The benchmark algorithms face model misspecification issues in the first two cases.

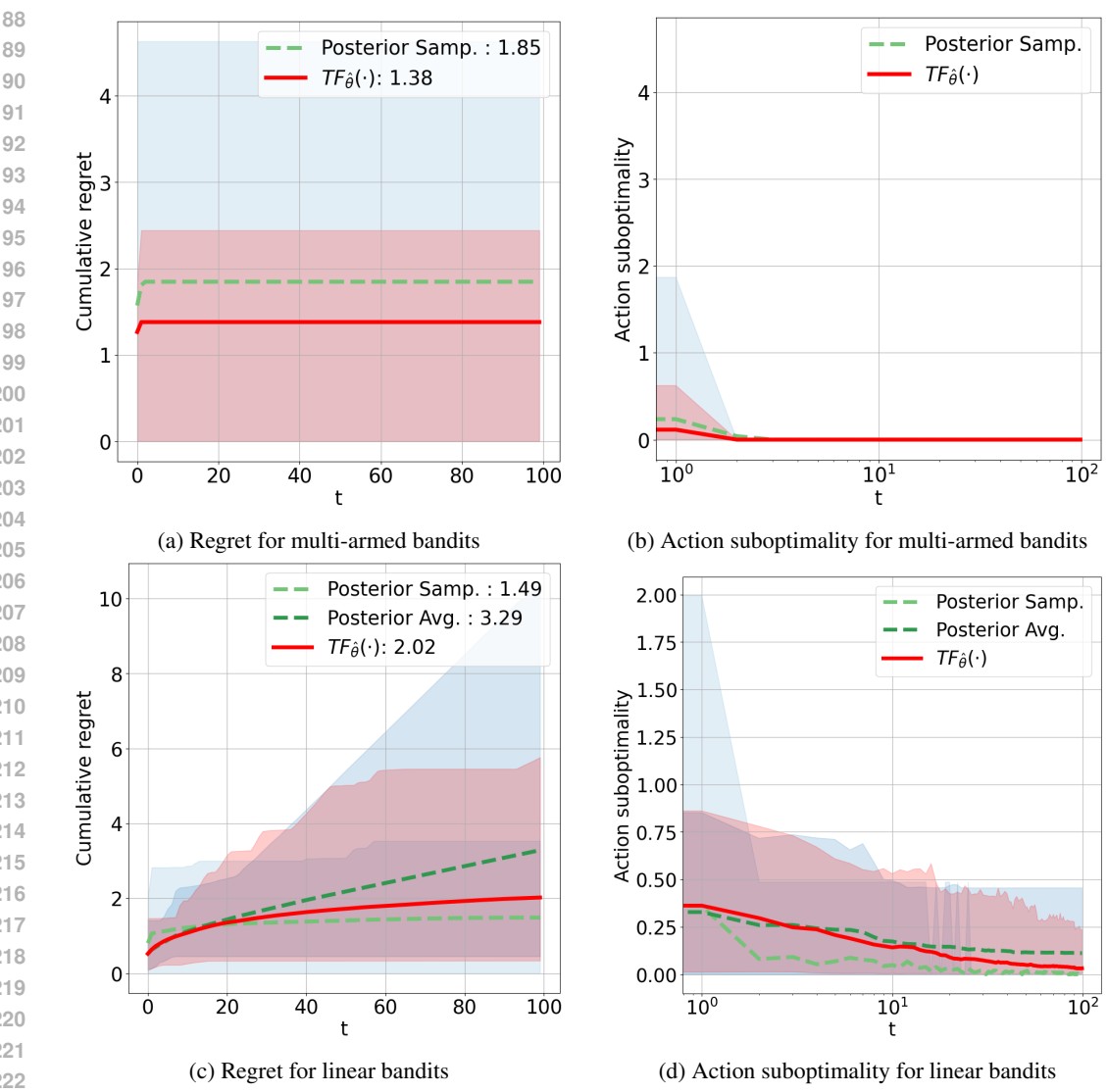

(a) Regret for multi-armed bandits

(b) Action suboptimality for multi-armed bandits

(c) Regret for linear bandits

(d) Action suboptimality for linear bandits

Figure 11: Performances of $\text{TF}_{\hat{\theta}}$ and Bayes-optimal decision functions. The numbers in the legend bar are the final regret at $t = 100$ and the shaded areas indicate the $90\%$ (empirical) confidence intervals.

For both pricing and newsvendor tasks, $\text{TF}_{\hat{\theta}}$ consistently outperforms all benchmarks across the three scenarios, especially in the first two cases. This result highlights the potential of $\text{TF}_{\hat{\theta}}$ to effectively handle model misspecifications.

**Setup.** For both the pricing and newsvendor tasks, during the pre-training phase of $\text{TF}_{\hat{\theta}}$, the pre-training data are generated from two types of demand functions: linear and square, with each type having a half probability. The support of $\mathcal{P}_\gamma$ is infinite, meaning the parameters in the demand functions are not restricted. The context dimension is 6 for the pricing task and 4 for the newsvendor task. The generating details and definitions of these two types of demands are provided in Appendix E.2. Additionally, when generating data using $\text{TF}_{\theta_m}$ in Algorithm 1, the sampled environment also has an equal probability of having a linear or square demand type.

Each subfigure in Figure 12 is based on 100 runs. We consider three different cases during testing: (i) the environment can have either the square or linear demand type (each with half probability); (ii) the environment can only have the linear demand type; and (iii) the environment can only have the square demand type. In the first case, the testing environment is sampled in the same way as the

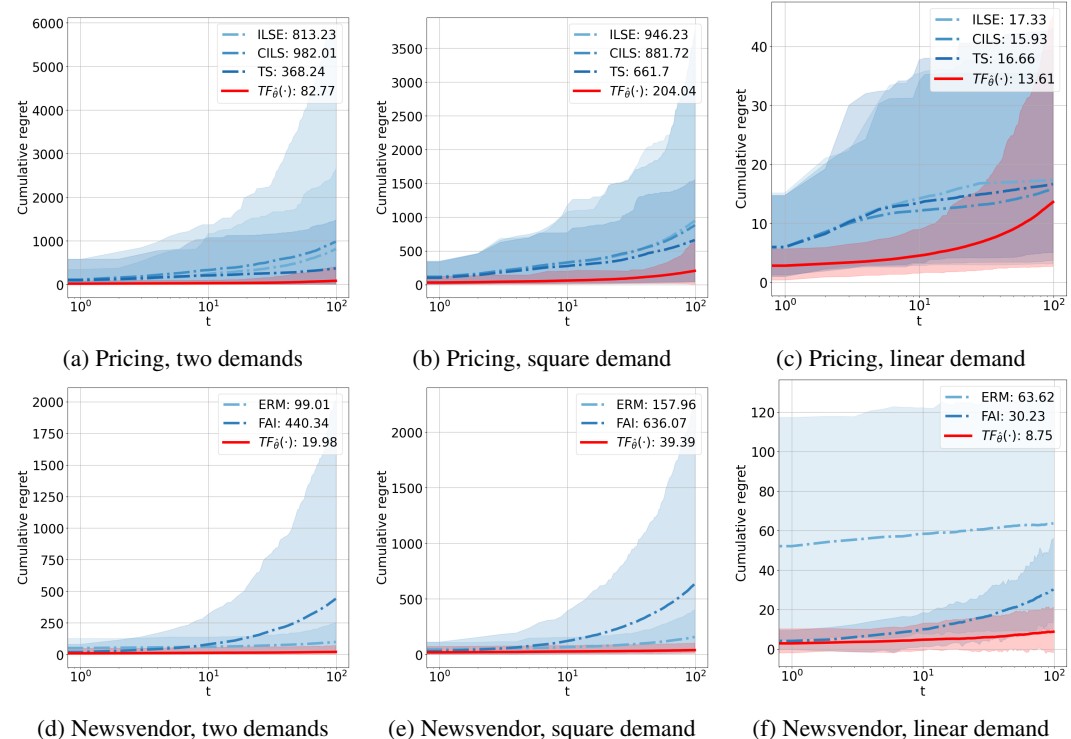

Figure 12: The average out-of-sample regret on dynamic pricing (first row) and newsvendor (second row) with two possible demand types: linear and square. The numbers in the legend bar are the final regret at $t = 100$ and the shaded areas indicate the $90\%$ (empirical) confidence intervals. The details of benchmarks can be found in Appendix E.3.

pre-training environment, while in the other two cases, we only include the sampled environments with the specified demand type until the total testing samples reach 100.

## C.5 ADDITIONAL RESULTS ON $\text{TF}_{\hat{\theta}}$ PERFORMANCES

This subsection presents further results on the performance of $\text{TF}_{\hat{\theta}}$. Specifically, we compare $\text{TF}_{\hat{\theta}}$ with benchmark algorithms on (i) simple tasks, which include only 4 possible environments in both pre-training and testing samples (Figure 13), and (ii) more complex tasks, which involve either 100 possible environments (Figure 14 (a), (b)) or 16 environments with two possible types of demand functions (Figure 14 (c), (d)).

Figures 13 and 14 illustrate the consistently and significantly superior performance of $\text{TF}_{\hat{\theta}}$ and $\text{Alg}^*$ across all tasks compared to the benchmark algorithms. These results underscore the advantage of leveraging prior knowledge about the tested environments.

**Setup.** All figures are based on 100 runs. We consider four tasks with 4 environments (i.e., the support of $\mathcal{P}_\gamma$ contains only 4 environments) in Figure 13 and two tasks with 100 environments in Figure 14 (a), (b). The setup for these tasks follows the generation methods provided in Appendix E.2.

For the tasks in Figure 14 (c), (d), there are 16 environments included in the support of $\mathcal{P}_\gamma$, where 8 environments have demand functions of the linear type and the other 8 have demand functions of the square type (see Appendix E.2 for the definitions of these two types). Both the pre-training and testing samples are drawn from $\mathcal{P}_\gamma$, i.e., from these 16 environments. The remaining setups for these tasks follow the methods provided in Appendix E.2.

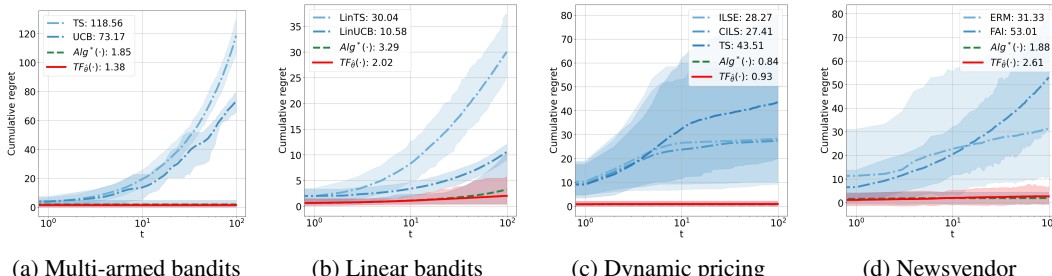

(a) Multi-armed bandits     (b) Linear bandits     (c) Dynamic pricing     (d) Newsvendor

Figure 13: The average out-of-sample regret on tasks with simpler environments, where each task only has 4 possible environments. The numbers in the legend bar are the final regret at $t = 100$ and the shaded areas indicate the $90\%$ (empirical) confidence intervals. The details of benchmarks can be found in Appendix E.3.

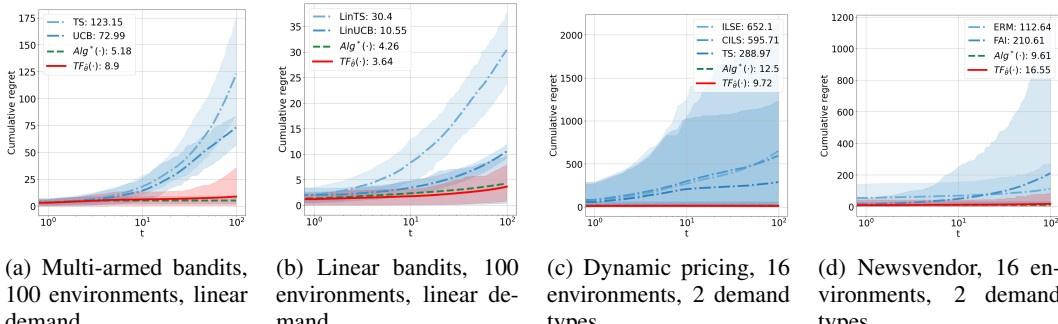

(a) Multi-armed bandits, 100 environments, linear demand

(b) Linear bandits, 100 environments, linear demand

(c) Dynamic pricing, 16 environments, 2 demand types

(d) Newsvendor, 16 environments, 2 demand types

Figure 14: The average out-of-sample regret on harder tasks: with 100 environments in (a) multi-armed bandits and (b) linear bandits; or with 16 environments, 2 demand types in (c) dynamic pricing and (d) newsvendor. The numbers in the legend bar are the final regret at $t = 100$ and the shaded areas indicate the $90\%$ (empirical) confidence intervals. The details of benchmarks can be found in Appendix E.3

### C.6 PERFORMANCE ON LONGER TESTING HORIZON

Our paradigm can be generalized to a longer testing horizon (beyond the length of the pre-training data) by introducing a *context window*. Specifically, we define a context window size $W$ and limit the input to the last $\min\{W, t\}$ timesteps of the input sequence $H_t$ to predict $a_t^*$ during both the pre-training and testing phases. By considering only the latest $\min\{W, t\}$ timesteps, we enable the model to handle testing sequences of any length, even when $t > T$. This technique is commonly used in the literature (e.g., Chen et al. [13]).

In this part, we evaluate $\text{TF}_{\hat{\theta}}$'s generalization ability when the testing horizon is longer than that seen during pre-training. As for a shorter testing horizon, $\text{TF}_{\hat{\theta}}$ can simply run until the end of the horizon, and previous results demonstrate strong performances in such cases, particularly when the horizon is less than 100. Figure 15 shows the performance when the testing horizon is extended from 100 (pre-training horizon) to 200 in the newsvendor problem. The results demonstrate that $\text{TF}_{\hat{\theta}}$ generalizes well in this extended horizon scenario, even in the censored demand setting where exploration is essential. Its actions remain nearly optimal beyond $t = 100$, resulting in lower regret compared to benchmark algorithms. For this experiment, all the pre-training and testing setups follow Appendix E, except the testing horizon length.

### C.7 OUT-OF-DISTRIBUTION PERFORMANCE

In this section, we evaluate $\text{TF}_{\hat{\theta}}$'s generalization ability under out-of-distribution (OOD) conditions, using dynamic pricing problems as the test case.

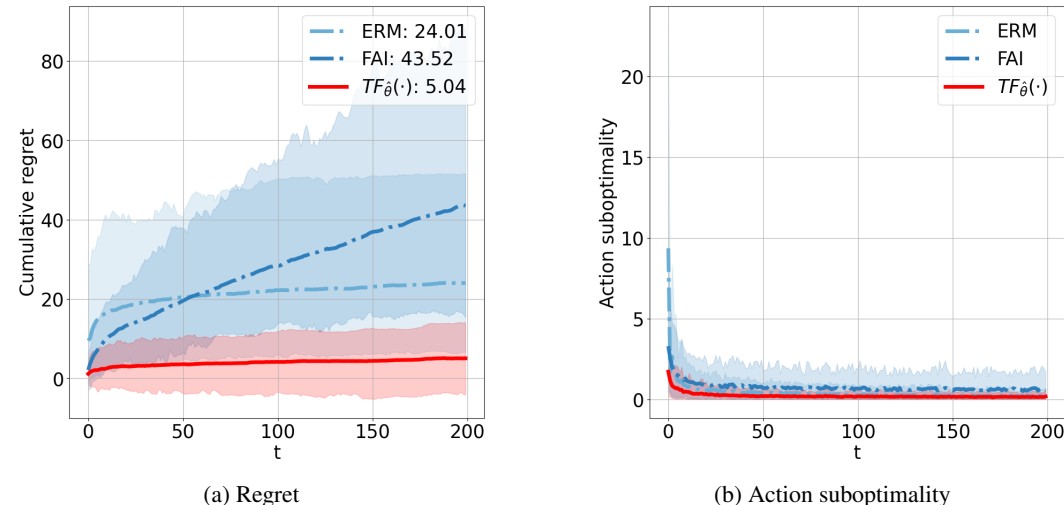

(a) Regret

(b) Action suboptimality

Figure 15: Performance under horizon generalization in newsvendor problems, where the pre-training samples have a horizon of 100 but the testing samples have a horizon of 200. It shows the average out-of-sample regret (first row) and action suboptimality, i.e., $|a_t^* - \mathtt{Alg}(H_t)|$, (second row) of $\mathtt{TF}_{\hat{\theta}}$ against benchmark algorithms. The numbers in the legend bar are the final regret at $t = 200$.

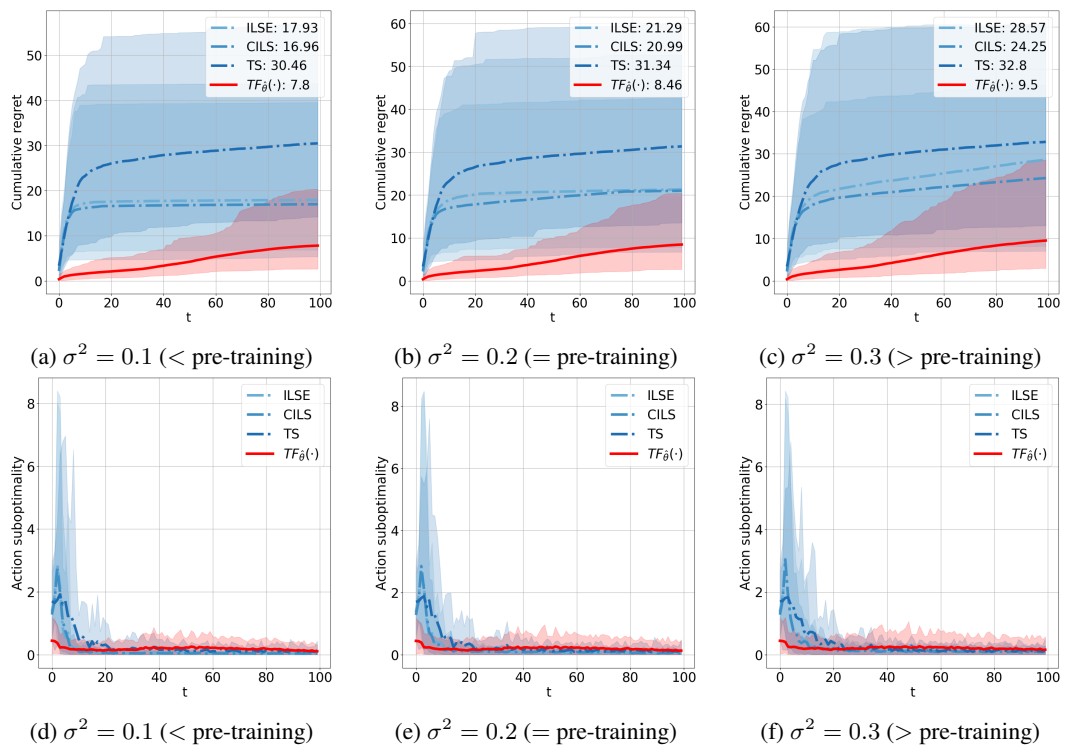

(a) $\sigma^2 = 0.1$ (< pre-training)    (b) $\sigma^2 = 0.2$ (= pre-training)    (c) $\sigma^2 = 0.3$ (> pre-training)

(d) $\sigma^2 = 0.1$ (< pre-training)    (e) $\sigma^2 = 0.2$ (= pre-training)    (f) $\sigma^2 = 0.3$ (> pre-training)

Figure 16: OOD performance under different testing noise variances, which may deviate from the pre-training variance of $\sigma^2 = 0.2$. It shows the average out-of-sample regret (first row) and action suboptimality, i.e., $|a_t^* - \mathtt{Alg}(H_t)|$, (second row) of $\mathtt{TF}_{\hat{\theta}}$ against benchmark algorithms. The numbers in the legend bar are the final regret at $t = 100$.

We test $\mathtt{TF}_{\hat{\theta}}$ in three different problem settings with varying testing noise variances while keeping the pre-training variance as 0.2 for all of them: $\sigma^2 = 0.2$, which matches the noise variance used

in pre-training; $\sigma^2 = 0.1$, which is smaller than that used in pre-training; and $\sigma^2 = 0.3$, which is larger. Figure 16 provides the results. We do not observe any significant sign of failure in $\text{TF}_{\hat{\theta}}$'s OOD performance: across all three settings, the benchmark algorithms consistently incur higher regret than $\text{TF}_{\hat{\theta}}$. Although $\text{TF}_{\hat{\theta}}$ shows some variation in mean regret across OOD settings (specifically, lower regret when $\sigma^2 = 0.2$ and higher regret when $\sigma^2 = 0.3$), this is partially due to the varying "difficulty" of the underlying tasks. As expected, higher noise variances, which need more data for accurately estimating the demand function compared to lower variance cases, lead to worse performance for all algorithms tested. However, these variations should not be interpreted as evidence of $\text{TF}_{\hat{\theta}}$ failing to handle OOD issues. In fact, all benchmark algorithms show similar performance fluctuations under these conditions, further demonstrating $\text{TF}_{\hat{\theta}}$'s good OOD performances. For this experiment, all the pre-training and testing setups follow Appendix E, except the noise variance in the testing samples.

## C.8 IMPACT OF NETWORK ARCHITECTURE

Here we investigate the effect of different network architectures on performance. Specifically, we replace the transformer/GPT-2 architecture used in $\text{TF}_{\hat{\theta}}$ with Long Short-Term Memory (LSTM) [25].

We maintain the overall architecture of $\text{TF}_{\hat{\theta}}$, as described in Appendix E.1 except for replacing the transformer/GPT-2 module with LSTM. We evaluate two LSTM variants: a 5-layer LSTM and a 12-layer LSTM (for comparison, the tested transformer also has 12 layers). Typically, LSTM architectures should be shallower than transformers in practice, which is why we also include the 5-layer version. Other hyperparameters like embedding space dimension are kept the same across all models. We pre-train and test these models on dynamic pricing problems (see Appendix E.2 for the experimental setup), and the pre-training procedure is identical to the one outlined in Appendix 3.

Figure 17 presents the testing regret for the three models. The results indicate no significant difference between the 5-layer and 12-layer LSTM models. However, the transformer consistently performs better than both LSTM variants. This outcome aligns with the transformer's superior performance in other domains, such as natural language processing, and suggests that transformers are more effective for sequential decision making tasks as well.

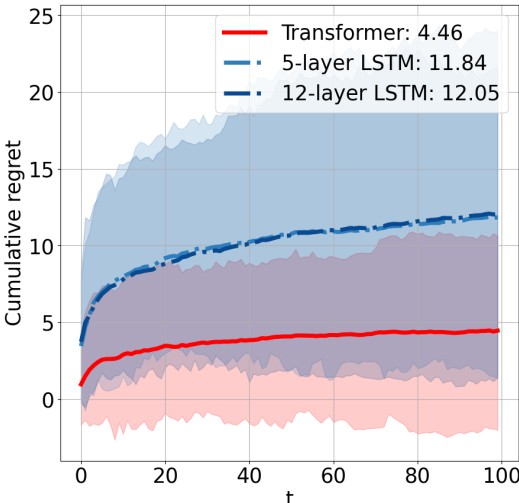

Figure 17: Comparison of the average out-of-sample regret between using the Transformer and LSTM architectures.

## C.9 ABLATION STUDY ON MODEL SIZE AND TASK COMPLEXITY

In this section, we explore the performance changes when tuning both model size and task complexity. Specifically, we adjust the number of layers to control the model size and the problem dimension

(the dimension of the context) in a dynamic pricing task to control the task complexity, where the number of unknown parameters is twice the context dimension. Since the optimal reward may scale differently across dimensions, we provide a relatively fair comparison by evaluating the advantage of $\mathtt{TF}_{\hat{\theta}}$ relative to the best-performing benchmark algorithm. This advantage is defined as the reward improvement rate of $\mathtt{TF}_{\hat{\theta}}$ compared to the best benchmark algorithm (the one among ILSE, CILS, and TS that achieves the highest reward). Thus, a positive value indicates $\mathtt{TF}_{\hat{\theta}}$ outperforms all benchmark algorithms, and a larger value indicates a better performance of $\mathtt{TF}_{\hat{\theta}}$. We test three model sizes (4, 8, and 12 layers) and three problem dimensions (4, 10, and 20), while keeping other hyperparameters the same. The results are presented in Figure 18.

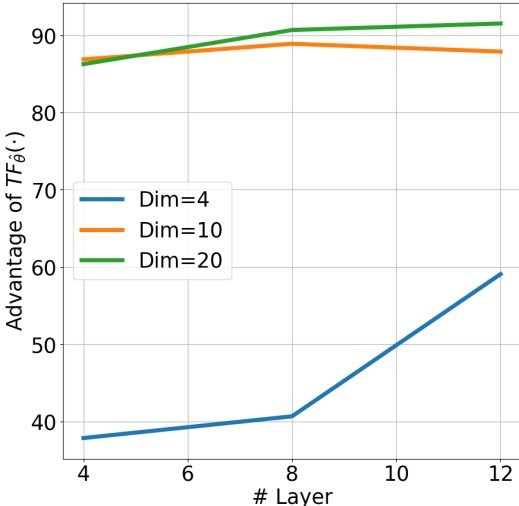

Figure 18: The advantage of $\mathtt{TF}_{\hat{\theta}}$ compared to the best benchmark algorithm in dynamic pricing, across different model size (number of layers) and problem complexity (problem dimensions).

From Figure 18, we observe the following: (i) $\mathtt{TF}_{\hat{\theta}}$ consistently outperforms the benchmark algorithms, and this advantage grows as the complexity of the problem increases. (ii) Increasing model size consistently improves the performance of $\mathtt{TF}_{\hat{\theta}}$ across all problem dimensions, indicating that larger models are generally preferred. This highlights $\mathtt{TF}_{\hat{\theta}}$'s superior ability to handle more complex tasks and these observations align with the scaling law [31] for large language models.

# D PROOFS

## D.1 DISCUSSIONS ON CLAIM 3.1

Consider the following setup where a feature-target pair $(X, Y)$ is generated from some distribution $\mathcal{P}_{X,Y}(\theta)$. Then we consider the minimization of the following loss

$$\min_{\theta} \mathbb{E}_{\mathcal{P}_{X,Y}(\theta)}[l(f_\theta(X), Y)]$$

for some loss function $l : \mathbb{R} \times \mathbb{R} \to \mathbb{R}$.

Under this formulation, both the data generation distribution $\mathcal{P}_{X,Y}(\theta)$ and the prediction function $f_\theta$ are parameterized by $\theta$. This describes the *performative prediction* problem [43].

In addition, we note that in our pre-trained transformer setting, the joint distribution can be factorized in

$$\mathcal{P}_{X,Y}(\theta) = \mathcal{P}_X(\theta) \cdot \mathcal{P}_{Y|X}.$$

That is, the parameter $\theta$ only induces a covariate shift by affecting the marginal distribution of $X$, but the conditional distribution $\mathcal{P}_{Y|X}$ remains the same for all the $\theta$. This exactly matches our setting of pre-trained transformer; to see this, different decision functions, $f$ or $\mathtt{TF}_{\theta_m}$, only differ in terms of the generation of the actions $a_\tau$'s in $H_t$ (for $\tau = 1, ..., t-1$) which is the features $X$ in this formulation but the optimal action $a_t^*$ will only be affected by $X_t$ in $H_t$.

Also, under this factorization, we define the Bayes-optimal estimator as

$$f^*(X) = \min_y \mathbb{E}_{\mathcal{P}_{Y|X}}[l(y, Y)|X].$$

Then it is easy to note that when $f^* = f_{\theta^*}$ for some $\theta^*$, and have

$$\theta^* \in \arg\min_\theta \mathbb{E}_{\mathcal{P}_{X,Y}(\theta')}[l(f_\theta(X), Y)]$$

for any $\theta'$.

In this light, the oscillating behavior of the optimization algorithms in [43] will not happen. Because for all the data generation distribution $\mathcal{P}_{X,Y}(\theta')$, they all point to one optimal $\theta^*$.

Back to the context of the pre-trained transformer, such a nice property is contingent on two factors: (i) the transformer function class is rich enough to cover $f^*$; (ii) there are infinitely many samples/we can use the expected loss. Also, the above argument is connected to the argument of Proposition 4.1.

### D.2  PROOF OF PROPOSITION 3.2

As noted in the proposition, let $b$ denote the bound of the loss function, and let the loss function $l(\text{TF}_\theta(H), a)$ is $D$-Lipschitz with respect to $H$ and $a$. We remark that the constant $D$ could be very large for the transformer model; the bound derived here serves more to illustrate the relationship between the generalization error and various problem parameters, but not really as an empirical bound to predict the test error of the underlying transformer.

Step 1. We aim to show with probability at least $1 - h$,

$$\kappa L_f(\text{TF}_{\hat\theta}) + (1 - \kappa)L_{\text{TF}_{\hat\theta}}(\text{TF}_{\hat\theta}) \leq \frac{1}{nT} \sum_{i=1}^n \sum_{t=1}^T l\left(\text{TF}_{\hat\theta}\left(H_t^{(i)}\right), a_t^{(i)*}\right)$$

$$+ \sqrt{\frac{\text{Comp}(\{\text{TF}_\theta : \theta \in \Theta\})}{nT}} + \sqrt{\frac{2b^2}{nT} \log\left(\frac{4}{h}\right)},$$

where $\text{Comp}(\{\text{TF}_\theta : \theta \in \Theta\})$ is some inherent constant describing the complexity of $\{\text{TF}_\theta : \theta \in \Theta\}$. Note that when samples follow the generation process (actions are generated by $f$)

$$\mathcal{P}_\gamma \to \gamma_i \to \mathcal{P}_{\gamma_i, f} \to \left\{\left(H_1^{(i)}, a_1^{(i)*}\right), \ldots, \left(H_T^{(i)}, a_T^{(i)*}\right)\right\},$$

and

$$L_f(\text{TF}_\theta) = \mathbb{E}_{\gamma \sim \mathcal{P}_\gamma}[L_f(\text{TF}_\theta; \gamma)] = \mathbb{E}_{\gamma \sim \mathcal{P}_\gamma}\left[\mathbb{E}_{(H_t, a_t^*) \sim \mathcal{P}_{\gamma, f}}\left[\sum_{t=1}^T l\left(\text{TF}_\theta\left(H_t\right), a_t^*\right)\right]\right].$$

Since $l(\text{TF}_\theta(\cdot), \cdot)$ is bounded by $b$, we can apply the McDiarmid's inequality (see Theorem D.8 in [40] for more details) with constant $\frac{2b}{nT}$ to $\sup_{\theta \in \Theta}\left|\frac{1}{nT}\sum_{i=1}^n \sum_{t=1}^T l\left(\text{TF}_\theta\left(H_t^{(i)}\right), a_t^{(i)*}\right) - L_f(\text{TF}_\theta)\right|$ such that with probability at least $1 - h$, we have

$$\sup_{\theta \in \Theta}\left|\frac{1}{nT}\sum_{i=1}^n \sum_{t=1}^T l\left(\text{TF}_\theta\left(H_t^{(i)}\right), a_t^{(i)*}\right) - L_f(\text{TF}_\theta)\right|$$

$$\leq \mathbb{E}_{(H_t, a_t^*) \sim \mathcal{P}_{\gamma, f}}\left[\sup_{\theta \in \Theta}\left|\frac{1}{nT}\sum_{i=1}^n \sum_{t=1}^T l\left(\text{TF}_\theta\left(H_t^{(i)}\right), a_t^{(i)*}\right) - \mathbb{E}_{(H_t, a_t^*) \sim \mathcal{P}_{\gamma, f}}\left[\sum_{t=1}^T l\left(\text{TF}_\theta\left(H_t\right), a_t^*\right)\right]\right|\right] \tag{10}$$

$$+ \sqrt{\frac{2b^2}{nT} \log\left(\frac{2}{h}\right)}$$

$$\leq 2\mathbb{E}_{(H_t, a_t^*) \sim \mathcal{P}_{\gamma, f}, w_{it}}\left[\sup_{\theta \in \Theta}\left|\frac{1}{nT}\sum_{i=1}^n \sum_{t=1}^T w_{it} l\left(\text{TF}_\theta\left(H_t^{(i)}\right), a_t^{(i)*}\right)\right|\right] + \sqrt{\frac{2b^2}{nT} \log\left(\frac{2}{h}\right)}$$

$$\leq \sqrt{\frac{\text{Comp}(\{\text{TF}_\theta : \theta \in \Theta\})}{nT}} + \sqrt{\frac{2b^2}{nT} \log\left(\frac{2}{h}\right)}, \tag{11}$$

where the second inequality uses the symmetric tricks (see Theorem 3.3 in [40] for more details), and $w_{it}$, so-called Rademacher variables are independent uniform random variables taking values in $\{-1, 1\}$.

Similarly, when samples follow the generation process (actions are generated by $\text{TF}_{\bar{\theta}}$)

$$\mathcal{P}_\gamma \to \gamma_i \to \mathcal{P}_{\gamma_i, \text{TF}_{\bar{\theta}}} \to \left\{ \left( H_1^{(i)}, a_1^{(i)*} \right), \dots, \left( H_T^{(i)}, a_T^{(i)*} \right) \right\},$$

by using the same tricks, we can have that with probability at least $1 - h$,

$$\sup_{\theta \in \Theta} \left| \frac{1}{nT} \sum_{i=1}^n \sum_{t=1}^T l \left( \text{TF}_\theta \left( H_t^{(i)} \right), a_t^{(i)*} \right) - L_{\text{TF}_{\bar{\theta}}}(\text{TF}_\theta) \right| \leq \sqrt{\frac{\text{Comp}(\{\text{TF}_\theta : \theta \in \Theta\})}{nT}} + \sqrt{\frac{2b^2}{nT} \log \left( \frac{2}{h} \right)}.$$

(12)

Since $\hat{\theta}$ is determined by equation 6 where $\kappa n$ data sequences are from $\mathcal{P}_{\gamma, f}$ and $(1 - \kappa)n$ data sequences are from $\mathcal{P}_{\gamma, \text{TF}_{\bar{\theta}}}$, by taking the uniform bound of equations (11) and (12), we have with probability at least $1 - h$,

$$\kappa L_f(\text{TF}_{\hat{\theta}}) + (1 - \kappa) L_{\text{TF}_{\bar{\theta}}}(\text{TF}_{\hat{\theta}}) \leq \frac{1}{nT} \sum_{i=1}^n \sum_{t=1}^T l \left( \text{TF}_{\hat{\theta}} \left( H_t^{(i)} \right), a_t^{(i)*} \right)$$
$$+ \sqrt{\frac{\text{Comp}(\{\text{TF}_\theta : \theta \in \Theta\})}{nT}} + \sqrt{\frac{2b^2}{nT} \log \left( \frac{4}{h} \right)}.$$

Step 2. Now we show

$$L(\text{TF}_{\hat{\theta}}) \leq L_f(\text{TF}_{\hat{\theta}}) + TD \cdot \mathbb{E}_{\gamma \sim \mathcal{P}_\gamma} \left[ W_1 \left( \mathcal{P}_{\gamma, f}, \mathcal{P}_{\gamma, \text{TF}_{\hat{\theta}}} \right) \right],$$
$$L(\text{TF}_{\hat{\theta}}) \leq L_f(\text{TF}_{\hat{\theta}}) + TD \cdot \mathbb{E}_{\gamma \sim \mathcal{P}_\gamma} \left[ W_1 \left( \mathcal{P}_{\gamma, \text{TF}_{\bar{\theta}}}, \mathcal{P}_{\gamma, \text{TF}_{\hat{\theta}}} \right) \right],$$

where $W_1(\cdot, \cdot)$ is the Wasserstein-1 distance.

By Kantorovich-Rubinstein's theorem, for distributions $\mathcal{D}_1$ and $\mathcal{D}_2$, we have

$$W_1(\mathcal{D}_1, \mathcal{D}_2) = \sup \left\{ \left| \mathbb{E}_{X \sim \mathcal{D}_1} g(X) - \mathbb{E}_{X \sim \mathcal{D}_2} g(X) \right| \, \big| \, g : \mathbb{R}^p \to \mathbb{R}, \, g \text{ is 1-Lipschitz} \right\}.$$

Since $l(\text{TF}_\theta(H), a)$ is $D$-Lipschitz with respect to $H$ and $a$ for all $\theta \in \Theta$, we have

$$L \left( \text{TF}_{\hat{\theta}}; \gamma \right) - L_f \left( \text{TF}_{\hat{\theta}}; \gamma \right) = \mathbb{E}_{(H_t, a_t^*) \sim \mathcal{P}_{\gamma, \text{TF}_{\hat{\theta}}}} \left[ \sum_{t=1}^T l \left( \text{TF}_{\hat{\theta}} (H_t), a_t^* \right) \right] - \mathbb{E}_{(H_t, a_t^*) \sim \mathcal{P}_{\gamma, f}} \left[ \sum_{t=1}^T l \left( \text{TF}_{\hat{\theta}} (H_t), a_t^* \right) \right]$$
$$\leq TD \cdot W_1 \left( \mathcal{P}_{\gamma, f}, \mathcal{P}_{\gamma, \text{TF}_{\hat{\theta}}} \right),$$

which implies that $L(\text{TF}_{\hat{\theta}}) \leq L_f(\text{TF}_{\hat{\theta}}) + TD \cdot \mathbb{E}_{\gamma \sim \mathcal{P}_\gamma} \left[ W_1 \left( \mathcal{P}_{\gamma, f}, \mathcal{P}_{\gamma, \text{TF}_{\hat{\theta}}} \right) \right]$. Similarly, we have $L(\text{TF}_{\hat{\theta}}) \leq L_f(\text{TF}_{\hat{\theta}}) + TD \cdot \mathbb{E}_{\gamma \sim \mathcal{P}_\gamma} \left[ W_1 \left( \mathcal{P}_{\gamma, \text{TF}_{\bar{\theta}}}, \mathcal{P}_{\gamma, \text{TF}_{\hat{\theta}}} \right) \right]$.

Therefore, by combining step 1 and step 2, we can conclude that with probability at least $1 - h$,

$$L(\text{TF}_{\hat{\theta}}) \leq \hat{L}(\text{TF}_{\hat{\theta}}) + \sqrt{\frac{\text{Comp}(\{\text{TF}_\theta : \theta \in \Theta\})}{nT}}$$
$$+ \kappa TD \cdot \mathbb{E}_{\gamma \sim \mathcal{P}_\gamma} \left[ W_1 \left( \mathcal{P}_{\gamma, f}, \mathcal{P}_{\gamma, \text{TF}_{\hat{\theta}}} \right) \right]$$
$$+ (1 - \kappa) TD \cdot \mathbb{E}_{\gamma \sim \mathcal{P}_\gamma} \left[ W_1 \left( \mathcal{P}_{\gamma, \text{TF}_{\bar{\theta}}}, \mathcal{P}_{\gamma, \text{TF}_{\hat{\theta}}} \right) \right] + \sqrt{\frac{2b^2}{nT} \log \left( \frac{4}{h} \right)},$$

which completes the proof.

### D.3 PROOF OF PROPOSITION 4.1

*Proof.* The proof can be done with the definition of $\text{Alg}^*$. Also, we refer more discussion to Section D.1. $\square$

## D.4 PROOF OF EXAMPLE 4.2

*Proof.* Since $\mathtt{Alg}^*$ is a function of any possible history $H_t$, we omit the subscrpit $t$ for the optimal actions for notation simplicity, and use $a^{(\gamma)*}$ as the optimal action of $\gamma$.

- **Posterior sampling under the cross-entropy loss.** We assume that $\mathcal{A} = \{a_1, \ldots, a_J\}$ and the output of $\mathtt{Alg}^*$ is a probability vector $(p_1, \ldots, p_J)$ such that $\sum_{j=1}^{J} p_j = 1$, meaning the probability of choosing each action. Then

$$\mathtt{Alg}^*(H) = \underset{\sum_{j=1}^{J} p_j = 1}{\arg\min} \int_\gamma -\log\left(p_{a^{(\gamma)*}}\right) \mathbb{P}(H|\gamma)\mathrm{d}\mathcal{P}_\gamma$$

$$= \underset{\sum_{j=1}^{J} p_j = 1}{\arg\min} \sum_{j=1}^{J} \left(-\log\left(p_j\right)\right) \int_{\{\gamma: \, a^{(\gamma)*}=j\}} \mathbb{P}(H|\gamma)\mathrm{d}\mathcal{P}_\gamma.$$

  By nothing $\int_{\{\gamma: \, a^{(\gamma)*}=j\}} \mathbb{P}(H|\gamma)\mathrm{d}\mathcal{P}_\gamma$ is the probability such that action $j$ is the optimal action conditional on $H$ (i.e., the posterior distribution of the optimal action), we are indeed minimizing the cross-entropy of the posterior distribution of the optimal action relative to the decision variables $p_j$. Thus, the optimal solution $p_j^* = \int_{\{\gamma: \, a^{(\gamma)*}=j\}} \mathbb{P}(H|\gamma)\mathrm{d}\mathcal{P}_\gamma$ for each $j$ and $\mathtt{Alg}^*(H_t)$ behaves as the posterior sampling.

- **Posterior averaging under the squared loss.** By definition,

$$\mathtt{Alg}^*(H) = \underset{a \in \mathcal{A}}{\arg\min} \int_\gamma ||a - a^{(\gamma)*}||_2^2 \mathbb{P}(H|\gamma)\mathrm{d}\mathcal{P}_\gamma$$

$$= \frac{\int_\gamma a^{(\gamma)*}\mathbb{P}(H|\gamma)\mathrm{d}\mathcal{P}_\gamma}{\int_\gamma \mathbb{P}(H|\gamma)\mathrm{d}\mathcal{P}_\gamma}$$

$$= \mathbb{E}_\gamma[a^{(\gamma)*}|H]$$

  where the second line is by the first order condition.

- **Posterior median under the absolute loss.** By definition,

$$\mathtt{Alg}^*(H) = \underset{a \in \mathcal{A}}{\arg\min} \int_\gamma |a - a^{(\gamma)*}|\mathbb{P}(H|\gamma)\mathrm{d}\mathcal{P}_\gamma.$$

  Then by zero-subgradient condition, we can conclude that $\mathtt{Alg}^*(H)$ satisfies

$$\int_{\{\gamma: \, \mathtt{Alg}^*(H) \leq a^{(\gamma)*}\}} \mathbb{P}(H|\gamma)\mathrm{d}\mathcal{P}_\gamma = \int_{\{\gamma: \, \mathtt{Alg}^*(H) \geq a^{(\gamma)*}\}} \mathbb{P}(H|\gamma)\mathrm{d}\mathcal{P}_\gamma.$$

  Divide both sides of this equation by $\int_\gamma \mathbb{P}(H|\gamma)\mathrm{d}\mathcal{P}_\gamma$, we can conclude that

$$\int_{\{\gamma: \, \mathtt{Alg}^*(H) \leq a^{(\gamma)*}\}} \mathrm{d}\mathcal{P}\left(\gamma|H\right) = \int_{\{\gamma: \, \mathtt{Alg}^*(H) \geq a^{(\gamma)*}\}} \mathrm{d}\mathcal{P}\left(\gamma|H\right)$$

  where $\mathcal{P}\left(\gamma|H_t\right)$ is the posterior distribution of $\gamma$. Hence $\mathtt{Alg}^*(H)$ is the posterior median.

$\square$

## D.5 PROOF OF PROPOSITION 4.3

*Proof.* **For the linear bandits problem**, we consider the following example. Suppose we have two environments, $\gamma_1, \gamma_2$, each with standard normal distributed noise and unknown parameters $w_{\gamma_1} = (1, 0)$ and $w_{\gamma_2} = (0, 1)$, respectively. Let the action space be $\mathcal{A} = [-1, 1] \times [-1, 1]$. The optimal actions for the two environments are $a^{(1)*} = (1, 0)$, $a^{(2)*} = (0, 1)$. We assume the prior distribution $\mathcal{P}_\gamma$ is given by $\mathcal{P}_\gamma(\gamma_1) = \mathcal{P}_\gamma(\gamma_2) = \frac{1}{2}$.

Then if $\texttt{Alg}^*$ is the posterior averaging, we have

$$a_t = \texttt{Alg}^*(H_t) = \mathcal{P}_\gamma\left(\gamma_1|H_t\right) \cdot a^{(1)*} + \mathcal{P}_\gamma\left(\gamma_2|H_t\right) \cdot a^{(2)*},$$

where the posterior distribution $\mathcal{P}_\gamma\left(\gamma_i|H_t\right)$ is given by

$$\mathcal{P}_\gamma\left(\gamma_i|H_t\right) = \frac{\prod_{\tau=1}^{t-1} \mathbb{P}_{\gamma_i}(o_\tau|a_\tau)}{\sum_{i'=1}^{2} \prod_{\tau=1}^{t-1} \mathbb{P}_{\gamma_{i'}}(o_\tau|a_\tau)}, \quad i = 1, 2.$$

We now use induction to show $\mathcal{P}_\gamma\left(\gamma_i|H_t\right) = \frac{1}{2}$ for any $t \geq 1$ and $i = 1, 2$, where $H_t$ can be generated by either $\gamma_1$ or $\gamma_2$. This results in $a_t = \frac{1}{2}a^{(1)*} + \frac{1}{2}a^{(2)*} = \left(\frac{1}{2}, \frac{1}{2}\right)$, which does not change with respect to $t$ and will cause regret linear in $T$.

**Step 1.** Since $\mathcal{P}_\gamma\left(\gamma_i|H_0\right) = \mathcal{P}_\gamma\left(\gamma_i\right) = \frac{1}{2}$, we have $a_1 = \frac{1}{2}a^{(1)*} + \frac{1}{2}a^{(2)*} = \left(\frac{1}{2}, \frac{1}{2}\right)$, and the conclusion holds for $t = 1$.

**Step 2.** Now assume the conclusion holds for $t$, i.e., $\mathcal{P}_\gamma\left(\gamma_i|H_t\right) = \frac{1}{2}$, and $a_t = \left(\frac{1}{2}, \frac{1}{2}\right)$. Since $w_{\gamma_1}^T a_t = w_{\gamma_2}^T a_t = \frac{1}{2}$, we have $\mathbb{P}_{\gamma_i}(o_t|a_t) = \frac{1}{\sqrt{2\pi}} \exp\left(-\frac{\left(o_t - \frac{1}{2}\right)^2}{2}\right)$ for $i = 1, 2$. Observe that

$$\mathcal{P}_\gamma(\gamma_i|H_{t+1}) = \frac{\mathbb{P}_{\gamma_i}(o_{t+1}|a_{t+1})\mathcal{P}_\gamma(\gamma_i|H_t)}{\sum_{i'=1}^{2} \mathbb{P}_{\gamma_{i'}}(o_{t+1}|a_{t+1})\mathcal{P}_\gamma(\gamma_{i'}|H_t)} = \frac{1}{2},$$

which implies $a_{t+1} = \left(\frac{1}{2}, \frac{1}{2}\right)$, and the conclusion holds for $t + 1$.

Thus, the conclusion holds for all $t \geq 1$. Then the regret is

$$\text{Regret}(\texttt{Alg}^*; \gamma_i) = \mathbb{E}\left[\sum_{t=1}^{T} r(X_t, a_t^*) - r(X_t, a_t)\right] = \frac{1}{2}T$$

for $i = 1, 2$.

**For the dynamic pricing problem**, we can construct a similar example. Suppose we have two environments without context $X_t$, denoted by $\gamma_1, \gamma_2$. The demands $o_t$ of them are set to be $o_t = 2 - a_t + \epsilon_t$ and $o_t = \frac{4}{5} - \frac{1}{5} \cdot a_t + \epsilon_t$ respectively, where $\epsilon_t \overset{i.i.d.}{\sim} \mathcal{N}(0, 1)$ and $a_t$ is the price. Accordingly, the optimal actions are then $a^{(1)*} = 1$ and $a^{(2)*} = 2$.

Then if $\texttt{Alg}^*$ is the posterior averaging, we have

$$a_t = \texttt{Alg}^*(H_t) = \mathcal{P}_\gamma\left(\gamma_1|H_t\right) \cdot a^{(1)*} + \mathcal{P}_\gamma\left(\gamma_2|H_t\right) \cdot a^{(2)*},$$

where the posterior distribution $\mathcal{P}_\gamma\left(\gamma_i|H_t\right)$ is given by

$$\mathcal{P}_\gamma\left(\gamma_i|H_t\right) = \frac{\prod_{\tau=1}^{t-1} \mathbb{P}_{\gamma_i}(o_\tau|a_\tau)}{\sum_{i'=1}^{2} \prod_{\tau=1}^{t-1} \mathbb{P}_{\gamma_{i'}}(o_\tau|a_\tau)}, \quad i = 1, 2.$$

with $\mathbb{P}_{\gamma_1}(o_\tau|a_\tau)$ as the normal distribution $\mathcal{N}\left(2 - a_t, 1\right)$ and $\mathbb{P}_{\gamma_2}(o_\tau|a_\tau)$ as $\mathcal{N}\left(\frac{4-a_t}{5}, 1\right)$.

Observe that $a_1 = \frac{1}{2}a^{(1)*} + \frac{1}{2}a^{(2)*} = \frac{3}{2}$ satisfies $2 - a_1 = \frac{4-a_1}{5}$. Following a similar analysis as in the linear bandits example, we can conclude that $\mathcal{P}_\gamma\left(\gamma_i|H_t\right) = \frac{1}{2}$ for any $t$ and $i = 1, 2$, where $H_t$ can be generated by either $\gamma_1$ or $\gamma_2$. Therefore, $a_t = \frac{3}{2}$ for all $t \geq 1$ and the regret is

$$\text{Regret}(\texttt{Alg}^*; \gamma_i) = \mathbb{E}\left[\sum_{t=1}^{T} r(X_t, a_t^*) - r(X_t, a_t)\right] = \frac{1}{4}T \cdot \mathbb{I}\left\{\gamma_i = \gamma_1\right\} + \frac{1}{20}T \cdot \mathbb{I}\left\{\gamma_i = \gamma_2\right\},$$

for $i = 1, 2$.

$\square$

## D.6 PROOF OF PROPOSITION 4.4

We make the following additional assumptions:

- There exists a constant $\bar{r}$ such that $\sup_{x \in \mathcal{X}, a \in \mathcal{A}, \gamma \in \Gamma} r(x, a; \gamma) \leq \bar{r}$, where we use $r(x, a; \gamma)$ to denote the reward function of environment $\gamma$.

- There exists a constant $C_r$ such that $\text{Alg}^*$ satisfies
$$\mathbb{E}[r(X_t, a_t^*) - r(X_t, \text{Alg}^*(H_t))|H_t] \leq C_r \sum_{\gamma_i \neq \gamma} \mathcal{P}(\gamma_i|H_t),$$
where the expectation on the left side is taken with respect to the possible randomness in $\text{Alg}^*$.

- $\mathcal{P}_\gamma$ is a uniform distribution over $\Gamma$.

*Proof.* We first compute a concentration inequality for $\sum_{\tau=1}^{t} \log\left(\frac{\mathbb{P}_\gamma(o_\tau|X_\tau, a_\tau)}{\mathbb{P}_{\gamma'}(o_\tau|X_\tau, a_\tau)}\right)$. By the Bernstein-type concentration bound for a martingale difference sequence (Theorem 2.19 in [50]), under the given conditions, we have for any $\gamma' \in \Gamma$, $t > 0$ and $1 > \delta > 0$ with probability $1 - \delta$,

$$\sum_{\tau=1}^{t} \log\left(\frac{\mathbb{P}_\gamma(o_\tau|X_\tau, a_\tau)}{\mathbb{P}_{\gamma'}(o_\tau|X_\tau, a_\tau)}\right) - \mathbb{E}\left[\log\left(\frac{\mathbb{P}_\gamma(o_\tau|X_\tau, a_\tau)}{\mathbb{P}_{\gamma'}(o_\tau|X_\tau, a_\tau)}\right)\right] \geq -\sqrt{2C_{\sigma^2}\Delta_{\text{Explore}}t\log(1/\delta)},$$

where

$$\sum_{\tau=1}^{t} \mathbb{E}\left[\log\left(\frac{\mathbb{P}_\gamma(o_\tau|X_\tau, a_\tau)}{\mathbb{P}_{\gamma'}(o_\tau|X_\tau, a_\tau)}\right)\right] = \sum_{\tau=1}^{t} \text{KL}\left(\mathbb{P}_\gamma(\cdot|X_\tau, a_\tau)\|\mathbb{P}_{\gamma'}(\cdot|X_\tau, a_\tau)\right) \geq t\Delta_{\text{Explore}}.$$

We think about two situations:

- If $\sqrt{t\Delta_{\text{Explore}}} \geq 2\sqrt{2C_{\sigma^2}\log(1/\delta)}$, then

$$\sum_{\tau=1}^{t} \log\left(\frac{\mathbb{P}_\gamma(o_\tau|X_\tau, a_\tau)}{\mathbb{P}_{\gamma'}(o_\tau|X_\tau, a_\tau)}\right) \geq t\Delta_{\text{Explore}} - \frac{1}{2}t\Delta_{\text{Explore}}$$

$$= \frac{1}{2}t\Delta_{\text{Explore}}$$

$$> \frac{1}{2}t\Delta_{\text{Explore}} - 4C_{\sigma^2}\log(1/\delta).$$

- If $\sqrt{t\Delta_{\text{Explore}}} < 2\sqrt{2C_{\sigma^2}\log(1/\delta)}$, then

$$\sum_{\tau=1}^{t} \log\left(\frac{\mathbb{P}_\gamma(o_\tau|X_\tau, a_\tau)}{\mathbb{P}_{\gamma'}(o_\tau|X_\tau, a_\tau)}\right) > t\Delta_{\text{Explore}} - 4C_{\sigma^2}\log(1/\delta)$$

$$\geq \frac{1}{2}t\Delta_{\text{Explore}} - 4C_{\sigma^2}\log(1/\delta).$$

Thus with probability $1 - \delta$, we have

$$\sum_{\tau=1}^{t} \log\left(\frac{\mathbb{P}_\gamma(o_\tau|X_\tau, a_\tau)}{\mathbb{P}_{\gamma'}(o_\tau|X_\tau, a_\tau)}\right) > \frac{1}{2}t\Delta_{\text{Explore}} - 4C_{\sigma^2}\log(1/\delta).$$

Now we apply the union bound to $\gamma_i$ for $i = 1, \ldots, k$ and for $t = 1, \ldots, T$: with probability $1 - 1/T$, for all $i = 1, \ldots, k$, and $t = 1, \ldots, T$

$$\mathcal{P}(\gamma_i|H_t) \leq \mathcal{P}(\gamma|H_t) \cdot \exp\left(-\frac{1}{2}t\Delta_{\text{Explore}} + 4C_{\sigma^2}\log(1/\delta)\right).$$

Now we are ready to prove the regret bound. We first decompose the regret:

$$\text{Regret}(\text{TF}_{\hat{\theta}}; \gamma) = \sum_{t=1}^{T} \mathbb{E}[r(X_t, \text{Alg}^*(H_t)) - r(X_t, \text{TF}_{\hat{\theta}}(H_t))] + \sum_{t=1}^{T} \mathbb{E}[r(X_t, a_t^*) - r(X_t, \text{Alg}^*(H_t))]$$

$$\leq \Delta_{\text{Exploit}} T + C_r \mathbb{E}\left[\sum_{t=1}^{T} \sum_{\gamma_i \neq \gamma} \mathcal{P}(\gamma_i | H_t)\right].$$

Since $\mathcal{P}(\gamma_i | H_t) + \mathcal{P}(\gamma | H_t) \leq 1$, we have with probability $1 - 1/T$, for all $i = 1, \ldots, k$, and $t = 1, \ldots, T$,

$$\mathcal{P}(\gamma_i | H_t) \leq \frac{1}{1 + \exp\left(\frac{1}{2} t \Delta_{\text{Explore}} - 4C_{\sigma^2} \log(kT^2)\right)}.$$

Let $t_0 = \left\lceil \frac{16 C_{\sigma^2} \log(kT^2)}{\Delta_{\text{Explore}}} \right\rceil$, then we have with probability $1 - 1/T$, for all $i = 1, \ldots, k$, and $t = 1, \ldots, T$,

$$\mathcal{P}(\gamma_i | H_t) \leq \frac{1}{1 + \exp\left(\frac{1}{2} t \Delta_{\text{Explore}} - 4C_{\sigma^2} \log(kT^2)\right)} < \exp\left(-\frac{1}{4} t \Delta_{\text{Explore}}\right).$$

Thus, we can conclude that

$$\mathbb{E}\left[\sum_{t=1}^{T} \sum_{\gamma_i \neq \gamma} \mathcal{P}(\gamma_i | H_t)\right] < 1 + \frac{16 C_{\sigma^2} \bar{r} \log(kT^2)}{\Delta_{\text{Explore}}} + k\mathbb{E}\left[\sum_{t=t_0}^{T} \exp\left(-\frac{1}{4} t \Delta_{\text{Explore}}\right)\right]$$

$$= O\left(\frac{\log k}{\Delta_{\text{Explore}}}\right)$$

and finish the regret bound.

$\square$

### D.7 EXAMPLE/JUSTIFICATION FOR THE CONDITIONS IN PROPOSITION 4.4

We use a set of linear bandits problems as a an example to illustrate Proposition 4.4. Consider a set of linear bandits problems with $o_t = R_t = w^\top a_t + \epsilon_t$, where $\epsilon_t$ is i.i.d. from a standard Normal distribution. Recall the environment parameter $\gamma = w$ here. We assume it has dimension $d$ and both $\mathcal{A}$ and $\Gamma$ are bounded with respect to the $L2$ norm. Further, we assume $\text{Alg}^*(H_t)$ is posterior averaging and for each time $t$,

$$\text{TF}_{\hat{\theta}}(H_t) = \text{Alg}^*(H_t) + \Delta_t,$$

where $\Delta_t$ follows a uniform distribution in $[-1, 1]^d$, i.i.d. across time and independent of $H_t$, $a_t$, and $\gamma$. This follows the numerical observations in Figure 3.

In this setting, we verify the conditions in Proposition 4.4. To simplify notations, we denote $\tilde{a}_t^* = \text{Alg}^*(H_t)$.

**First condition.** Since

$$\mathbb{E}[r(X_t, \text{Alg}^*(H_t)) - r(X_t, \text{TF}_{\hat{\theta}}(H_t)) | H_t] = \mathbb{E}[\gamma^\top \tilde{a}_t^* - \gamma^\top \tilde{a}_t^* - \gamma^\top \Delta_t | H_t] = 0,$$

we have $\Delta_{\text{Exploit}} = 0$.

**Second condition.** Since $\epsilon_t$ is i.i.d. from a standard Normal distribution, we have for $o_t = \gamma^\top a_t + \epsilon_t$ and $a_t = \text{TF}_{\hat{\theta}}(H_t)$,

$$\log\left(\frac{\mathbb{P}_\gamma(o_t | X_t, a_t)}{\mathbb{P}_{\gamma'}(o_t | X_t, a_t)}\right) = \frac{(o_t - \gamma'^\top a_t)^2 - (o_t - \gamma^\top a_t)^2}{2}$$

$$= \frac{((\gamma - \gamma')^\top a_t)^2}{2} - (\gamma - \gamma')^\top a_t \epsilon_t.$$

Thus, the KL divergence, which is the expectation of the above term with respect to $\epsilon_t$ and $\Delta_t$ (which are independent of $H_t$), is

$$\mathbb{E}\left[\frac{((\gamma - \gamma')^\top (\tilde{a}_t^* + \Delta_t))^2}{2}\bigg| H_t\right] \geq \frac{1}{6}\|\gamma - \gamma'\|_2^2.$$

And further, for the "noise" term $(\gamma - \gamma')^\top a_t \epsilon_t$, since its variance is bounded by $O(\|\gamma - \gamma'\|_2^2)$ (due to the assumptions on the boundedness of $a_t$ and that $\epsilon_t$ follows a standard Normal distribution), we can set $\Delta_{\text{Explore}} = \min_{\gamma' \in \Gamma} \|\gamma - \gamma'\|_2^2$.

Further, since

$$\mathbb{E}[r(X_t, a_t^*) - r(X_t, \texttt{Alg}^*(H_t))|H_t] \leq \bar{r} \sum_{\gamma_i \neq \gamma} \mathcal{P}(\gamma_i|H_t),$$

we can set $C_r = \bar{r}$.

And thus, we can conclude the regret bound is $\text{Regret}(\texttt{TF}_{\hat{\theta}}, \gamma) = O\left(\frac{\log k}{\min_{\gamma' \in \Gamma} \|\gamma - \gamma'\|_2^2}\right)$.

# E DETAILS FOR NUMERICAL EXPERIMENTS

## E.1 TRANSFORMER ARCHITECTURE AND ALGORITHM 1 IMPLEMENTATION

### E.1.1 TRANSFORMER FOR SEQUENTIAL DECISION MAKING

We adopt the transformer architecture for regression tasks from [22] to solve sequential decision-making tasks, with modifications tailored to our specific setting.

**Prompt.** At time $t$, the prompt consists of two types of elements derived from the history $H_t$: (i) the "feature" elements, which stack the context $X_\tau \in \mathbb{R}^d$ at each time $\tau \leq t$ and the observation $o_{\tau-1} \in \mathbb{R}^k$ from $\tau - 1$ (with $o_0$ set as a $k$-dimensional zero vector in our experiments), i.e., $\{(X_\tau, o_{\tau-1})\}_{\tau=1}^t \subset \mathbb{R}^{d+k}$. These elements serve as the "features" in the prediction; and (ii) the "label" elements, which are the actions $\{a_\tau\}_{\tau=1}^{t-1}$ in $H_t$. Therefore, the prompt contains $2t - 1$ elements in total.

**Architecture.** The transformer is based on the GPT-2 family [45]. It uses two learnable linear transformations to map each "feature" and "label" element into vectors in the embedding space respectively (similar to tokens in language models). These vectors are processed through the GPT-2's attention mechanism, resulting in a vector that encapsulates relevant contextual information. This vector then undergoes another learnable linear transformation, moving from the embedding space to the action $\mathcal{A}$, ultimately resulting in the prediction of $a_t^*$. In our experiments, the GPT-2 model has 12 layers, 16 attention heads, and a 256-dimensional embedding space. The experiments are conducted on 2 A100 GPUs with `DistributedDataParallel` method of Pytorch.

### E.1.2 ALGORITHM 1 IMPLEMENTATION AND PRE-TRAINING DETAILS

During pre-training, we use the AdamW optimizer with a learning rate of $10^{-4}$ and a weight decay of $10^{-4}$. The dropout rate is set to 0.05. For implementing Algorithm 1 in our experiments, we set the total iterations $M = 130$ with an early training phase of $M_0 = 50$. The number of training sequences per iteration is $n = 1500 \times 64$, which are randomly split into 1500 batches with a batch size of 64. The transformer's parameter $\theta$ is optimized to minimize the averaged loss of each batch. As suggested in Proposition B.2, we select the cross-entropy loss for the multi-armed bandits, squared loss for the dynamic pricing, and absolute loss for the linear bandits and newsvendor problem.

To reduce computation costs from sampling histories during the early training phase, instead of sampling a new dataset $D_m$ in each iteration (as described in Algorithm 1), we initially sample $10^6$ data samples before the pre-training phase as an approximation of $\mathcal{P}_{\gamma,f}$. Each sample in the batch is uniformly sampled from these $10^6$ data samples.

To reduce transformer inference costs during the mixed training phase, we reset the number of training sequences to $n = 15 \times 64$ per iteration and set the ratio $\kappa = 1/3$, meaning $10 \times 64$ samples are from $\mathcal{P}_{\gamma, \texttt{TF}_{\theta_m}}$ and $5 \times 64$ samples are from $\mathcal{P}_{\gamma,f}$ (sampled from the pool of pre-generated $10^6$

samples as before). Consequently, we adjust the number of batches from 1500 to 50 to fit the smaller size of training samples, keeping the batch size at 64.

**Decision Function $f$ in Pre-Training.** To mitigate the OOD issue mentioned in Section 3, we aim for the decision function $f$ to approximate $\mathtt{Alg}^*$ or $\mathtt{TF}_{\hat{\theta}}$. However, due to the high computation cost of $\mathtt{Alg}^*$ and the unavailability of $\mathtt{TF}_{\hat{\theta}}$, we set $f(H_t) = a_t^* + \epsilon_t'$, where $\epsilon_t'$ is random noise simulating the suboptimality of $\mathtt{Alg}^*$ or $\mathtt{TF}_{\hat{\theta}}$ and independent of $H_t$. As we expect such suboptimality to decrease across $t$, we also reduce the influence of $\epsilon_t'$ across $t$. Thus, in our experiments, $\epsilon_t'$ is defined as:

$$\epsilon_t' \sim \begin{cases} 0 & \text{w.p. } \max\{0, 1 - \frac{2}{\sqrt{t}}\}, \\ \mathrm{Unif}[-1, 1] & \text{w.p. } \min\{1, \frac{2}{\sqrt{t}}\}, \end{cases}$$

where for the multi-armed bandits we replace $\mathrm{Unif}[-1, 1]$ by $\mathrm{Unif}\{-2, -1, 1, 2\}$ and for the linear bandits with dimension $d > 1$ we apply the uniform distribution of the set $[-1, -1]^d$. We further project $f(H_t)$ into $\mathcal{A}$ when $f(H_t) \notin \mathcal{A}$.

**Curriculum.** Inspired by Garg et al. [22] in the regression task, we apply curriculum training to potentially speed up Algorithm 1. This technique uses "simple" task data initially and gradually increases task complexity during the training. Specifically, we train the transformer on samples with a smaller horizon $\tilde{T} = 20$ (generated by truncating samples from $\mathcal{P}_{\gamma,f}$ or $\mathcal{P}_{\gamma,\mathtt{TF}_{\theta_m}}$) at the beginning and gradually increase the sample horizon to the target $T = 100$. We apply this curriculum in both the early training phase ($m \leq 50$) and the mixed training phase ($m > 50$). The last 30 iterations focus on training with $\tilde{T} = 100$ using non-truncated samples from $\mathcal{P}_{\gamma,\mathtt{TF}_\theta}$ to let the transformer fit more on the non-truncated samples. The exact setting of $\tilde{T}$ is as follows:

$$\tilde{T} = \begin{cases} 20 \times (m\%10 + 1) & \text{when } m = 1, \ldots, 50, \\ 20 \times (m\%10 - 4) & \text{when } m = 51, \ldots, 100, \\ 100 & \text{when } m = 100, \ldots, 130. \end{cases}$$

### E.2 Environment Generation

Throughout the paper, we set the horizon $T = 100$ for all tasks. The environment distribution $\mathcal{P}_\gamma$ is defined as follows for each task:

- **Stochastic multi-armed bandit:** We consider the number of actions/arms $k = 20$. The environment parameter $\gamma$ encapsulates the expected reward $r_a$ of each arm $a = 1, \ldots, 20$, where the reward of arm $a$ is sampled from $\mathcal{N}(r_a, 0.2)$. Thus, the environment distribution $\mathcal{P}_\gamma$ is defined by the (joint) distribution of $(r_1, \ldots, r_{20})$. We set $r_a \overset{\text{i.i.d.}}{\sim} \mathcal{N}(0, 1)$ for each action $a$. The optimal arm is $a^* = \arg\max_a r_a$.

- **Linear bandits:** We consider the dimension of actions/arms $d = 2$ and $\mathcal{A} = \mathcal{B}(0, 1)$, i.e., a unit ball centered at the origin (with respect to the Euclidean norm). The environment parameter $\gamma$ encapsulates the (linear) reward function's parameter $w \in \{\tilde{w} \in \mathbb{R}^2 : \|\tilde{w}\|_2 = 1\}$, where the reward of arm $a$ is sampled from $\mathcal{N}(w^\top a, 0.2)$. Thus, the environment distribution $\mathcal{P}_\gamma$ is defined by the distribution of $w$. We set $w$ to be uniformly sampled from the unit sphere, and the corresponding optimal arm being $a^* = w$.

- **Dynamic pricing:** We set the noise $\epsilon_t \overset{\text{i.i.d.}}{\sim} \mathcal{N}(0, 0.2)$ and the context $X_t$ to be sampled i.i.d. and uniformly from $[0, 5/2]^d$, with $d = 6$ as the dimension of contexts, and $\mathcal{A} = [0, 30]$. We consider the linear demand function family $d(X_t, a_t) = w_1^\top X_t - w_2^\top X_t \cdot a_t + \epsilon_t$, where $w_1, w_2 \in \mathbb{R}^d$ are the demand parameters. Thus, the environment distribution $\mathcal{P}_\gamma$ is defined by the (joint) distribution of $(w_1, w_2)$. We set $w_1$ to be uniformly sampled from $[1/2, 3/2]^6$ and $w_2$ to be uniformly sampled from $[1/20, 21/20]^6$, independent of $w_1$. The optimal action $a_t^* = \frac{w_1^\top X_t}{2 \cdot w_2^\top X_t}$. We also consider a square demand function family, which will be specified later, to test the transformer's performance on a mixture of different demand tasks.

- **Newsvendor problem:** We set the context $X_t$ to be sampled i.i.d. and uniformly from $[0, 3]^d$ with $d = 4$, and $\mathcal{A} = [0, 30]$. We consider the linear demand function family $d(X_t, a_t) = w^\top X_t + \epsilon_t$, where $w \in \mathbb{R}^2$. The environment parameter $\gamma$ encapsulates (i) the upper bound $\bar{\epsilon}$ of the noise $\epsilon_t$, where $\epsilon_t \overset{\text{i.i.d.}}{\sim} \mathrm{Unif}(0, \bar{\epsilon})$; (ii) the left-over cost $h$ (with

the lost-sale cost $l$ being 1); and (iii) the demand parameter $w$. Accordingly, we set (i) $\bar{\epsilon} \sim \text{Unif}[1, 10]$; (ii) $h \sim \text{Unif}[1/2, 2]$; and (iii) $w$ uniformly sampled from $[0, 3]^2$. The optimal action can be computed by $a_t^* = w^\top X_t + \frac{\bar{\epsilon}}{1+h}$ [18], which is indeed the $\frac{1}{1+h}$ quantile of the the random variable $w^\top X_t + \epsilon_t$. We also consider a square demand function family, which will be specified later, to test the transformer's performance on a mixture of different demand tasks.

**Finite Pool of Environments.** To study the behavior and test the performance of the transformer on finite possible environments (e.g., to see if and how $\text{TF}_{\hat{\theta}}$ converges to $\text{Alg}^*$), we also consider a finite pool of environments as the candidates of $\gamma$. Specifically, for some tasks we first sample finite environments (e.g., $\gamma_1, \ldots, \gamma_4$) i.i.d. following the sampling rules previously described. We then set the environment distribution $\mathcal{P}_\gamma$ as a uniform distribution over the pool of sampled environments (e.g., $\{\gamma_1, \ldots, \gamma_4\}$). We should notice this pool of environments does not restrict the context generation (if any): in both the pre-training and testing, contexts are generated following the rules previously described and are independent of the given finite pool.

**Tasks with Two Demand Types.** To test the transformer's performance on tasks with a mixture of different demand types, we also consider the square demand function family for the dynamic pricing and newsvendor tasks. Specifically, besides the linear demand function family, we also consider $d^{\text{sq}}(X_t, a_t) = (w_1^\top X_t)^2 - (w_2^\top X_t) \cdot a_t + \epsilon_t$ in dynamic pricing and $d^{\text{sq}}(X_t, a_t) = (w^\top X_t)^2 + \epsilon_t$ in the newsvendor problem. Thus, with a slight abuse of notation, we can augment $\gamma$ such that it parameterizes the type of demand (linear or square). In experiments related to multi-type demands, the probability of each type being sampled is $1/2$, while the distributions over other parameters of the environment remain the same as in the linear demand case.

## E.3 BENCHMARK ALGORITHMS

### E.3.1 MULTI-ARMED BANDITS

- Upper Confidence Bound (UCB) [35]: Given $H_t$, the action $a_t$ is chosen by $a_t = \arg\max_{a \in \mathcal{A}} \hat{r}_a + \frac{\sqrt{2 \log T}}{\min\{1, n_a\}}$, where $n_a$ is the number of pulling times of arm $a$ before time $t$ and $\hat{r}_a$ is the empirical mean reward of $a$ based on $H_t$.

- Thompson sampling (TS) [47]: Given $H_t$, the action $a_t$ is chosen by $a_t = \arg\max_{a \in \mathcal{A}} \tilde{r}_a$, where each $\tilde{r}_a \sim \mathcal{N}\left(\hat{r}_a, \frac{\sqrt{2 \log T}}{\min\{1, n_a\}}\right)$ is the sampled reward of arm $a$ and $\hat{r}_a, n_a$ follow the same definitions as in UCB.

- $\text{Alg}^*$: Given $H_t$ from a task with a pool of $|\Gamma|$ environments $\{\gamma_1, \ldots, \gamma_{|\Gamma|}\}$, the action $a_t$ is chosen by the posterior sampling defined in Algorithm 2. The posterior distribution can be computed by

$$\mathcal{P}(\gamma_i | H_t) = \frac{\exp(-\frac{1}{\sigma^2} \sum_{\tau=1}^{t-1} (o_\tau - r_{a_\tau}^i)^2)}{\sum_{i'=1}^{|\Gamma|} \exp(-\frac{1}{\sigma^2} \sum_{\tau=1}^{t-1} (o_\tau - r_{a_\tau}^{i'})^2)},$$

where $r_a^i$ is the expected reward of $a$ in environment $\gamma_i$ and $\sigma^2$ is the variance of the noise (which equals to $0.2$ in our experiments).

### E.3.2 LINEAR BANDITS

- LinUCB [14]: Given $H_t$, we define $\Sigma_t = \sum_{\tau=1}^{t-1} a_\tau a_\tau^\top + \sigma^2 I_d$, where $\sigma^2$ is the variance of the reward noise. The action $a_t$ is chosen by $a_t \in \arg\max_{a \in \mathcal{A}} \hat{w}_t^\top a + \sqrt{2 \log T} \|a\|_{\Sigma_t^{-1}}$, where $\hat{w}_t = \Sigma_t^{-1}(\sum_{\tau=1}^{t-1} o_\tau \cdot a_\tau)$ is the current estimation of $w$ from $H_t$.

- LinTS [1]: Given $H_t$, the action $a_t$ is chosen by $a_t = \arg\max_{a \in \mathcal{A}} \tilde{w}_t^\top a$, where $\tilde{w}_t \sim \mathcal{N}\left(\hat{w}_t, \sqrt{2 \log T} \Sigma_t^{-1}\right)$ as the sampled version of $w$ and $\hat{w}_t, \Sigma_t$ follow the same definitions in LinUCB.

- $\text{Alg}^*$: Given $H_t$ from a task with a pool of $|\Gamma|$ environments $\{\gamma_1, \ldots, \gamma_{|\Gamma|}\}$, the action $a_t$ is chosen by the posterior median defined in Algorithm 2. The posterior distribution can be

computed by

$$\mathcal{P}(\gamma_i|H_t) = \frac{\exp(-\frac{1}{\sigma^2}\sum_{\tau=1}^{t-1}(o_\tau - w_i^\top a_\tau)^2)}{\sum_{i'=1}^{|\Gamma|}\exp(-\frac{1}{\sigma^2}\sum_{\tau=1}^{t-1}(o_\tau - w_{i'}^\top a_\tau)^2)},$$

where $w_i$ is the reward function parameter in environment $\gamma_i$ and $\sigma^2$ is the variance of the noise (which equals to $0.2$ in our experiments).

### E.3.3 DYNAMIC PRICING

All the benchmark algorithms presented below assume the demand model belongs to the linear demand function family as defined in Appendix E.2 (which can be mis-specified when we deal with dynamic pricing problems with two demand types).

- Iterative least square estimation (ILSE) [44]: At each $t$, it first estimates the unknown demand parameters $(w_1, w_2)$ by applying a ridge regression based on $H_t$, and chooses $a_t$ as the optimal action according to the estimated parameters and the context $X_t$. Specifically, we denote $(\hat{w}_{1,t}, -\hat{w}_{2,t}) = \Sigma_t^{-1}(\sum_{\tau=1}^{t-1} o_\tau \cdot z_\tau)$ as the estimation of $(w_1, -w_2)$ through a ridge regression, where $z_\tau = (X_\tau, a_\tau \cdot X_\tau)$ serves as the "feature vector" and $\Sigma_t = \sum_{\tau=1}^{t-1} z_\tau z_\tau^\top + \sigma^2 I_d$ ($\sigma^2$ is the variance of the demand noise). Then the action is chosen as the optimal one by treating $(\hat{w}_{1,t}, \hat{w}_{2,t})$ as the true parameter: $a_t = \frac{\hat{w}_{1,t}^\top X_t}{2\hat{w}_{2,t}^\top X_t}$.

- Constrained iterated least squares (CILS) [32]: It is similar to the ILSE algorithm except for the potential explorations when pricing. Specifically, we follow the notations in ILSE and denote $\hat{a}_t = \frac{\hat{w}_{1,t}^\top X_t}{2\hat{w}_{2,t}^\top X_t}$ as the optimal action by treating $(\hat{w}_{1,t}, \hat{w}_{2,t})$ as the true parameter (the chosen action of ILSE), and denote $\bar{a}_{t-1}$ as the empirical average of the chosen actions so far. Then the CILS chooses

$$a_t = \begin{cases} \bar{a}_{t-1} + \text{sgn}(\delta_t)\frac{t^{-\frac{1}{4}}}{10}, & \text{if } |\delta_t| < \frac{1}{10}t^{-\frac{1}{4}}, \\ \hat{a}_t, & \text{otherwise}, \end{cases}$$

where $\delta_t = \hat{a}_t - \bar{a}_{t-1}$. The intuition is that if the tentative price $\hat{a}_t$ stays too close to the history average, it will introduce a small perturbation around the average as price experimentation to encourage parameter learning.

- Thompson sampling for pricing (TS) [51]: Like the ILSE algorithm, it also runs a regression to estimate $(w_1, w_2)$ based on the history data, while the chosen action is the optimal action of a sampled version of parameters. Specifically, we follow the notations in ILSE and then TS chooses

$$a_t = \frac{\tilde{\alpha}}{2 \cdot \tilde{\beta}},$$

where $(\tilde{\alpha}, \tilde{\beta}) \sim \mathcal{N}((\hat{w}_{1,t}^\top X_t, \hat{w}_{2,t}^\top X_t), \tilde{\Sigma}_t^{-1}) \in \mathbb{R}^2$ are the sampled parameters of the "intercept" and "price coefficient" in the linear demand function and

$$\tilde{\Sigma}_t = \left( \begin{pmatrix} X_t & \mathbf{0} \\ \mathbf{0} & X_t \end{pmatrix}^\top \Sigma_t^{-1} \begin{pmatrix} X_t & \mathbf{0} \\ \mathbf{0} & X_t \end{pmatrix} \right)^{-1}$$

is the empirical covariance matrix given $X_t$.

- Alg*: Given $H_t$ from a task with a pool of $|\Gamma|$ environments $\{\gamma_1, \ldots, \gamma_{|\Gamma|}\}$, the action $a_t$ is chosen by the posterior averaging defined in Algorithm 2. To compute the posterior distribution, we follow the notations in ILSE and denote $w = (w_1, w_2)$ as the stacked vector of parameters, then the posterior distribution is

$$\mathcal{P}(\gamma_i|H_t) = \frac{\exp(-\frac{1}{\sigma^2}\sum_{\tau=1}^{t-1}(o_\tau - w_i^\top z_\tau)^2)}{\sum_{i'=1}^{|\Gamma|}\exp(-\frac{1}{\sigma^2}\sum_{\tau=1}^{t-1}(o_\tau - w_{i'}^\top z_\tau)^2)},$$

where $w_i$ is the demand function parameter in environment $\gamma_i$ and $\sigma^2$ is the variance of the noise (which equals to $0.2$ in our experiments).

### E.3.4 NEWSVENDOR

All the benchmark algorithms presented below assume the demand model belongs to the linear demand function family as defined in Appendix E.2 (which can be mis-specified when we deal with newsvendor problems with two demand types).

- Empirical risk minimization (ERM) [7]: Since the optimal action $a_t^*$ is the $\frac{1}{1+h}$ quantile of the random variable $w^\top X_t + \epsilon_t$ [18], ERM conducts a linear quantile regression based on the observed contexts and demands $\{(X_\tau, o_\tau)\}_{\tau=1}^{t-1}$ to predict the $\frac{1}{1+h}$ quantile on $X_t$.

- Feature-based adaptive inventory algorithm (FAI) [18]: FAI is an online gradient descent style algorithm aiming to minimize the cost $\sum_{t=1}^{T} h \cdot (a_t - o_t)^+ + l \cdot (o_t - a_t)^+$ (we set $l = 1$ in our experiments). Specifically, it chooses $a_t = \tilde{w}_t^\top X_t$, where

$$\tilde{w}_t = \begin{cases} \tilde{w}_{t-1} - \frac{h}{\sqrt{t}} \cdot x_{t-1}, & \text{if } o_{t-1} < a_{t-1}, \\ \tilde{w}_{t-1} + \frac{l}{\sqrt{t}} \cdot x_{t-1}, & \text{otherwise}, \end{cases}$$

  is the online gradient descent step and $\tilde{w}_0$ can be randomly sampled in $[0,1]^d$.

- Alg*: Given $H_t$ from a task with a pool of $|\Gamma|$ environments $\{\gamma_1, \ldots, \gamma_{|\Gamma|}\}$, the action $a_t$ is chosen by the posterior median as shown in Algorithm 2. To compute the posterior distribution, we denote $\bar{\epsilon}_\gamma$ and $\beta_\gamma$ as the noise upper bound and demand function parameter of $\gamma$ at $\tau \leq t - 1$, and define the event $\mathcal{E}_{\gamma,\tau} = \left\{ 0 \leq o_\tau - \beta_\gamma^\top X_\tau \leq \bar{\epsilon} \right\}$ to indicate the feasibility of environment $\gamma$ from $(X_\tau, o_\tau)$, and denote $\bar{\mathcal{E}}_{\gamma,t} = \bigcap_{\tau=1}^{\tau-1} \mathcal{E}_{\gamma,\tau}$ to indicate the feasibility at $t$. Then the posterior distribution of the underlying environment is

$$\mathcal{P}(\gamma_i | H_t) = \frac{\mathbb{1}_{\bar{\mathcal{E}}_{\gamma_i,t}} \cdot \bar{\epsilon}_{\gamma_i}^{1-t}}{\sum_{i'=1}^{|\Gamma|} \mathbb{1}_{\bar{\mathcal{E}}_{\gamma_{i'},t}} \cdot \bar{\epsilon}_{\gamma_{i'}}^{1-t}}.$$

## E.4 DETAILS FOR FIGURES

### E.4.1 DETAILS FOR FIGURE 2

**Figure 2 (a).** In Figure 2 (a), we independently implement Algorithm 1 to train two transformer models on a dynamic pricing task, which has an infinite support of the prior environment distribution $\mathcal{P}_\gamma$ and 6-dimensional contexts (more details can be found in Appendix E.2). For both models, the training parameters are identical except for the $M_0$ in Algorithm 1: the first curve (blue, thick line) uses $M_0 = 50$, while the second curve (orange, dashed line) uses $M_0 = 130$, meaning no transformer-generated data is utilized during training. All other parameters follow the configuration detailed in Appendix E.1. The figure shows the mean testing regret at each training iteration across 128 randomly sampled environments, with the shaded area representing the standard deviation. We further provide more results in Appendix C.1.2.

**Figure 2 (b).** In Figure 2 (b), we consider a dynamic pricing task with a pool of 8 linear demand functions and has 6-dimensional contexts. We generate 30 sequences from $\mathtt{TF}_{\theta_m}$ across different training iterations for this task: before using the transformer-generated data in the training ($m = 40, 50$) and after using such data ($m = 70, 80, 110, 120$). All generated sequences share the same context sequence $\{X_t\}_{t=1}^{T}$ to control the effect of contexts on the chosen actions and the resulting observations. We stack the first 20 actions and observations into a single sample point and use t-SNE method to visualize these high-dimension points as shown in Figure 2 (b). We further apply the same method to generate points from Alg* to study the behavior of $\mathtt{TF}$.

We can observe that: (i) compared to the points from $\mathtt{TF}$ which have not been trained on self-generated data (i.e., $\mathtt{TF}_{\theta_{40}}$, $\mathtt{TF}_{\theta_{50}}$), the points from the trained ones are closer to the Alg*'s, i.e., the expected decision rule of a well-trained $\mathtt{TF}$; (ii) When being trained with more self-generated data, the points from $\mathtt{TF}_{\theta_m}$ get closer to the Alg*'s. These observations suggest that utilizing the self-generated data (like in Algorithm 1) can help mitigate the OOD issue as discussed in Section 3: the data from $\mathtt{TF}_{\theta_m}$ can get closer to the target samples from Alg* during the training and thus the pre-training data is closer to the testing data.

### E.4.2 DETAILS FOR FIGURE 3.

For Figure 3, subfigure (a) is based on a dynamic pricing task with 6-dimensional contexts and (b) is based on a newsvendor task with 4-dimensional contexts. Both tasks have a pool of 4 environments, i.e., the support of $\mathcal{P}_\gamma$ only contains 4 environments, with linear demand function type. Each subfigure is based on a sampled environment with a sampled sequence of contexts $\{X_t\}_{t=1}^{30}$ from the corresponding task. For each task, the data generation process follows the description detailed in Appendix E.2, and the architecture of $\text{TF}_{\hat{\theta}}$ and the definition of $\text{Alg}^*$ can be found in Appendix E.1 and Appendix E.3 respectively. We further provide more results and discussions in Appendix C.2.

