# OpenReview forum: "Understanding the Training and Generalization of Pretrained Transformer for Sequential Decision Making"
_ICLR.cc/2025/Conference — Submitted to ICLR 2025_

### Official Review · Reviewer_hvZS · 2024-10-28

**Soundness:** 2
**Presentation:** 1
**Contribution:** 2
**Rating:** 5
**Confidence:** 3

**Summary:**

This paper looks at how supervised pre-trained transformers can be used in sequential decision-making tasks, which is a subfield of reinforcement learning that doesn't involve transition probability matrices. The authors believe that transformers can perform well in tasks like bandits, dynamic pricing, and newsvendor problems if you leverage closed-form or easily computable optimal actions during the pre-training phase. They look at an issue that arises when there are differences between the training and test phases, and suggest an algorithm that lets transformer-generated actions be part of the training, which helps with generalisation. The authors also give some theoretical insights by looking at the transformer model as a near-Bayesian optimal predictor and suggest some numerical experiments comparing its performance against structured algorithms like UCB and Thompson sampling, which show where transformers work well.

**Strengths:**

By showing that transformers can outperform traditional algorithms in some sequential decision-making contexts, this work contributes to advancing the understanding of transformers.

**Weaknesses:**

**Confusing claims and demonstration:**

- The notations abuse. For example, decision function $f$ generate $a$ or $a^*$. In Eq. 5, the authors claim to generate $a^*$  but there is no $a^*$ in Eq. 5. I can only infer that $f$ generate $a^*$. And then below Eq. 7 it becomes $a_{\tau}$.

- In the paragraph, below Eq. 9, the authors claim that OOD in this setting is because "when generating the training data, there is no way we can know the final parameter. This is not an OOD problem to me. Training on some tasks and testing on other tasks, or changing test-time environmental parameters are more likely to be OOD problem.

- Above Eq. 9, pertraining -> pretraining, and this paragraph is unclearly written. Overly referring to the equation make me hard to follow the idea and intuition.



**Lack of Novelty**

- The theory is comprehensively covered, while simply adding a mixed training phase and testing only on simple experimental setup doesn't seem novel enough to me, given the existing works [1-3].

- In addition, given that Transformer is a powerful sequence model, isn't studying pretraining on state-free bandit problems less meaningful? The authors claim that bandit is a special case, but it's to me more like out-of-scope because sequential problems are more related.



Since the motivations and clarity are less satisfying, I didn't look into the proof in details. While it's good to see that mixing online and offline data can preserve a theoretical guarantee.



[1] Lee, Jonathan, et al. "Supervised pretraining can learn in-context reinforcement learning." *Advances in Neural Information Processing Systems* 36 (2024).

[2] Laskin, Michael, et al. "In-context reinforcement learning with algorithm distillation." *arXiv preprint arXiv:2210.14215* (2022).

[3] Sodhani, Shagun, Amy Zhang, and Joelle Pineau. "Multi-task reinforcement learning with context-based representations." *International Conference on Machine Learning*. PMLR, 2021.

**Questions:**

See Weakness.

---

> ### Author Response · Authors · 2024-11-15
> **Response to Reviewer hvZS Part I**
>
> We thank the reviewer for taking the time, but we strongly disagree with these comments. It seems there are fundamental misunderstandings regarding the initial setup of our paper, and we are surprised to see such misunderstandings arise in this well-known RL/decision-making setting. We hope our explanation helps clear up these confusions and would appreciate it if the reviewer could take more time for re-evaluation.
>
> We would like to first address the most fundamental setup question, Weakness 1, as it significantly affects the understanding of the rest of the paper.
>
>
> # Reviewer’s Weakness 1:
>
>
> The notation is confusing. For example, the decision function $f$ generates $a$ or $a^*$. In Eq. 5, the authors claim to generate $a^*$, but there is no $a^*$ in Eq. 5. I can only infer that $f$ generates $a^*$. Then, below Eq. 7, it becomes $a_\tau$.
>
>
> # Our Reply on Weakness 1:
>
>
> We are not abusing the notation; it is both precise and correct.
>
> We NEVER claim that $f$ generates $a^*$ in Equation 5. In fact, we explicitly stated TWICE that the optimal decision $a^*$ is generated by Equation 3 (see lines 178 and 195). Additionally, we clearly specified that $f$ is a pre-specified decision function that generates $a_t$ in trajectories, which is commonly known as the behavior function. Could the reviewer please point out where we supposedly state that $f$ generates the optimal decision $a^*$?
>
> Then, in Eq. 7, $f$ generates $a_\tau$ for $\tau = 1, \cdots, t-1$, because we need a trajectory for every $t = 1, \cdots, T$, and these indices are necessary for each $t$. That is why we use $\tau$ in line 195. We believe this is the most appropriate definition and do not see any part that might cause confusion.
>
> To clarify the general intuition further, $f$ can be thought of as a generating function that produces trajectories for pre-training. The actions in these trajectories are often suboptimal. However, since these trajectories are simulated and the environment is usually known, the optimal action $a^*$ can be determined using Equation 3.

---

> ### Author Response · Authors · 2024-11-16
> **Response to Reviewer hvZS Part II**
>
> ```
> Weakness 2: In the paragraph, below Eq. 9, the authors claim that OOD in this setting is because...
> ```
> We hope that after addressing Weakness 1, the reviewer will understand why the difference in the trajectory between the pre-training stage and the evaluation stage constitutes an OOD issue. Indeed, such an OOD issue has also been mentioned in [1] (the paper reviewer cited, page 5 right before section 4 in the arxiv version). For clarity, let us explain further where the OOD issue come from.
>
> In the pre-training phase, the trajectories are generated by some behavior function $f$, and the trajectory $(X_1, a_1, o_1,\cdots, X_{t-1}, a_{t-1}, o_{t-1}, X_t)$ (notice that $a_{\tau}$ for $\tau = 1,\cdots, t-1$ is generated by $f$) is accompanied by an optimal decision $a_t^*$. The intention is to let the Transformer learn the best action when it sees the trajectory. However, during the evaluation phase, the trajectories are generated incrementally by the trained transformer $TF_{\theta}$. It is highly likely that this type of trajectories encountered during evaluation have not been seen during pre-training, because that requires the transformer $TF_{\theta}$ to be very close to the behavior function $f$ . Consequently, the Transformer $TF_{\theta}$ might not know the right action when the input is a trajectory it generated itself.
>
> We believe this is clearly an OOD problem. As the reviewer mentioned, it can be thought of as a change in environmental parameters or a shift in tasks between training and testing, which will change the distribution of input trajectories.
>
> ```
> Above Eq. 9, pertraining -> pretraining, and this paragraph is unclearly written. Overly referring to the equation make me hard to follow the idea and intuition.
> ```
> Thanks for the suggestion, we will update it in the next version.
> ```
> Question 1: The theory is comprehensively covered, while simply adding a mixed training phase and testing
> ```
> We hope that after addressing Weakness 1, the reviewer will understand why our work is different from other works. In general, we are the first to propose the mix-pretraining scheme, which works empirically and is supported by theoretical results regarding convergence and regret.
>
> First, we disagree with the reviewer's comment that "the theory is comprehensively covered." In [1], the authors mention the OOD issue but do not address it. The plain scheme in [1] does not converge in our setting, where there is a massive number of environments, as shown in Figure 2(a). Moreover, we provide a more meaningful regret analysis with less restrictive assumptions, which could be important for the theory RL/decision-making community.
>
> Next, [2] and [3] focus more on the empirical side. The authors do not mention the OOD problem for the input trajectory, and there are no theoretical results provided. We believe that the reviewer should not downplay the importance of theoretical contributions.
>
> ```
> Question 2: In addition, given that Transformer is a powerful sequence model, isn't studying pretraining on state-free bandit problems less meaningful
> ```
> We believe this is another misunderstanding of our setting. Bandit problems are state-free, but in terms of learning, the information in the current trajectory contains meaningful data (e.g., how many times each arm has been pulled and the historical rewards). The trajectories heavily depend on the behavior function $f$ that generates the pre-training trajectory.  Therefore, it is definitely related to sequential problems, as the trajectory containing historical information changes as the transformer keep making online decisions over time. Other decision-making problems, including pricing and the newsvendor problem, follow the same reasoning.
>
>
> # Conclusion
>
> To summarize, we believe there is a significant misunderstanding of our paper. We hope our explanation helps clear up these confusions and would appreciate it if the reviewer could take more time for re-evaluation.

---

> > ### Comment · Reviewer_hvZS · 2024-11-16
> > **Response to Authors Part II**
> >
> > Thanks very much for the further clarification.
> >
> > - I understand the OOD problem claimed by the authors now. It is caused by self-generated historical trajectories mismatch to the behavior trajectories. However, in the setting of this paper, the training and testing happen in a single task. But according to [1], page 5 " Sources of distribution mismatch." as the authors refer to, distribution mismatch on downstream **test-time tasks** (can be not included in the training data) are what they mean, instead of the OOD caused by self-generation in a single task.
> >
> > - About "The theory is comprehensively covered", sorry for the confusion, I meant "the theory of the problem proposed by the authors is comprehensively covered by the authors".
> >
> > - Thanks for explaining, I understand now it's an auto-regressive procedure, so even in state-free problem it's still a sequence problem.
> >
> > A follow-up question, I saw the authors provide out-of-sample regret in the experiments, but never define it. This should be explained to show whether you are dealing with OOD problem in practice. Overall, I think the problem is studied in a single task, which makes this work less significant. I'm happy to change my thought if any of the understanding above is incorrect.

---

> > > ### Author Response · Authors · 2024-11-16
> > > **Second Round Response to hvZS Part II**
> > >
> > > We thank the reviewer for the quick reply. However, once again, we disagree with all the questions raised. Please see our response below, and feel free to let us know if you have further questions.
> > > ```
> > > ```
> > > - Question 1: I understand the OOD problem claimed by the authors now. It is caused by self-generated historical trajectories mismatch to the behavior trajectories. However, in the setting of this paper, the training and testing happen in a single task. But according to [1], page 5 " Sources of distribution mismatch." as the authors refer to, distribution mismatch on downstream test-time tasks (can be not included in the training data) are what they mean, instead of ...
> > >
> > > ***Response***
> > >
> > > We have to disagree on this point. Please let us assume that the reviewer has read our last reply in "Second Round Response to hvZS Part I", because this response will be partly based on that.
> > >
> > > - First, according to [1], page 5 "Sources of distribution mismatch," the authors mentioned THREE types of distribution mismatch, and the FIRST type they mentioned is EXACTLY the type of OOD problem we previously described. Specifically, the trajectory rolled out by the Transformer itself differs from the trajectory generated by the behavior function. In [1], it is stated: "(1) When deployed, $M_{\theta}$ will execute its learned policy which invariably induces a distribution over states different from $D_{query}$." Here, $M_{\theta}$ refers to the Transformer, and $D_{query}$ refers to the pretraining distribution. The "distribution over states different from $D_{query}$" implies that the trajectories, consisting of different states/actions, are different. Therefore, we disagree that this type of OOD issue is not mentioned in [1].
> > >
> > > - Second, we disagree with the reviewer's statement that "in the setting of this paper, the training and testing happen in a single task." From our last reply in "Second Round Response to hvZS Part I," it should be clear that our design allows training and testing environments come from different task distributions. Our experiments in Figure 12 and 16 demonstrate the effectiveness of our mix-pretraining strategy in handling pre-training and testing distribution mismatches. This is quite an advancement, as Figure 2(a) in our paper shows that the pre-training pipeline in [1] fails to converge even in a much simpler setting. This suggests that our paper also has the potential to addresses the second type of OOD problem mentioned in [1] and by the reviewer: "(2) Pretraining $T_{pre}$ likely differs from the downstream $T_{test}$." More experimental details on this can be found in Figure 12&16 and Appendix E.3.
> > >
> > > ```
> > > ```
> > >
> > > - Question 2: About "The theory is comprehensively covered", sorry for the confusion...
> > >
> > > - Question 3: Thanks for explaining, I understand now it's an auto-regressive procedure...
> > >
> > > ***Response***
> > >
> > > We understand that, given limited time, it is easy to make mistakes in phrasing, which is why communication is essential. We appreciate your time spent on this. We hope our clarification has provided a clearer understanding.
> > >
> > > ```
> > > ```
> > >
> > > - Question 4: A follow-up question, I saw the authors provide out-of-sample regret in the experiments, but never define it. This should be explained to show whether you are dealing with OOD problem in practice. Overall, I think the problem is studied in a single task, which makes this work less significant...
> > >
> > > ***Response***
> > >
> > > We apologize for any possible confusion. Typically, in the terminology of machine learning, out-of-sample regret is defined as the regret computed over environments sampled from the same distribution as the pre-training environment. We can include this definition in an updated version of the paper if it would help clarify this point.
> > >
> > > However, we disagree with the reviewer's claim that "the problem is studied in a single task," and we want to emphasize that our work addresses environments in various tasks and OOD settings. In general, we evaluated the regret in different testing environments:
> > >
> > > - In Figures 4 and 14, we report the standard out-of-sample regret.
> > > - In Figure 12, the regret is computed for environments sampled from a distribution different from the original heterogeneous pre-training distribution. As noted earlier, the pre-training distribution contains tasks with different structures (e.g., stochastic bandits vs. linear bandits), while the testing environment includes only one type of task out of these tasks from pre-training.
> > > - In Figure 16, we test on highly OOD environments, where the parameters in testing environment differ significantly from those in the pre-training distribution.
> > >
> > > # Conclusion
> > >
> > > We hope our explanation helps clear up these confusions and would appreciate it if the reviewer could take time for re-evaluation.
> > >
> > >
> > > [1] Lee, Jonathan, et al. "Supervised pretraining can learn in-context reinforcement learning." Advances in Neural Information Processing Systems 36 (2024).

---

> > > > ### Comment · Reviewer_hvZS · 2024-11-17
> > > > **Response to Authors**
> > > >
> > > > Thanks very much for the clarification.
> > > >
> > > > I basically understand what this work try to address now. I decide to change to 5 at this point considering the paper in general. Addressing the following can help improve the paper:
> > > >
> > > > **Presentation and Writing**
> > > > - Please change period symbol at the end of Eq. 5 to comma.
> > > > - Please revise the paragraph that overly refers to equations rather than explaining as promised.
> > > > - Line 178 and line 185, don't use - symbol to split the bullet points.
> > > > - As for out-of-sample regret, all the experimental details especially like the evaluation metric widely used in the experiments needs to be clearly defined to guarantee the reproducibility. It is not my personal request but a common standard of a machine learning paper. In addition, it can help demonstrate the practical use.
> > > > - Lots of the contents are trivial and less informative. Such as 2.1 and 2.2. The setting is actually simple and common, shouldn't take that long to explain. The space should be saved for more informative demonstration. For example, how severe and how common is the type 1 OOD problem. Analyzing other methods' limitation in dealing with this problem.
> > > > - In general, the paper is a little hard to follow and not well-structured. Please study [1] more to leverage their way of explaining ideas and concepts in blocks, and the use of notations, and also bold to highlight.
> > > >
> > > >
> > > > **Significance**
> > > > - The significance of this work is still less satisfying to me. This work aims to address one out of three OOD problems proposed in [1] via mixing the data. I think it's a less demanding and trivial-looking problem from my understanding (the 2nd T_test should be more important). To mitigate this, the authors should zoom in to demonstrating the need of solving this particular problem. For example, show via experiments that how bad other problems perform, what's the reason behind it, and prove your analysis with plots or data, rather than simply plotting the curves comparison.
> > > > - Limited application and a little bit over-claim. I understand even state-free problem like bandits might form a sequential problem in the pretraining and testing setting. However, the title of this paper claims for sequential decision-making problems, so conducting experiments in bandit problems is less significant, but only a proof of concept.

---

> > > > > ### Author Response · Authors · 2024-11-30
> > > > > **Additional Experiment Part II**
> > > > >
> > > > > Generally speaking, the insights from these experiments are that our algorithm is robust and still outperform the benchmarks under OOD settings, partly because the stable training pipeline we provided and the generalization power of Transformer itself.
> > > > >
> > > > > # Experiment I:
> > > > >
> > > > > In experiment I, we train the decision-maker under environment with pre-training noise $\epsilon_t\sim\mathcal{N}(0,0.2)$, and vary testing noise distributions such that $\sigma^2 = 0.1$, $\sigma^2 = 0.2$, and $\sigma^2 = 0.3$, separately.
> > > > >
> > > > > The result can been seen in https://docs.google.com/document/d/1bt7MN17L8ttBKZHAKPECEy-PHj5Uwli6VKnlNL7TF48/edit?usp=sharing
> > > > >
> > > > > # Insights from experiment I:
> > > > > - Across all three settings, our algorithm consistently outperforms those benchmarks.
> > > > > - As expected our model shows some variation in mean regret across OOD settings (specifically, lower regret when $\sigma^2 = 0.2$ and higher regret when $\sigma^2 = 0.3$), this is partially due to the varying "difficulty" of the underlying tasks. Indeed, for all tested algorithms, higher noise variances will need more data for accurately estimating the demand function compared to lower variance cases. Therefore, these variations should not be interpreted as signs of our algorithms failing to handle OOD issues. In fact, all benchmark algorithms show similar performance fluctuations under these conditions.
> > > > >
> > > > > # Experiment II:
> > > > >
> > > > > We further evaluate our model under distributional shifts in the generation of parameters $(\alpha, \beta)$, considering both out-of-domain and in-domain shifts. During pre-training, these parameters are sampled per environment as $\alpha \sim \text{Unif}([0.5, 1.5]^6)$ and $\beta \sim \text{Unif}([0.05, 1.05]^6)$. In the testing phase, two types of shifts are introduced, parameterized by the shift level $\mu_{\text{shift}}$:
> > > > >
> > > > > # Insights from experiment II
> > > > >
> > > > > - **Out-of-domain shifts**: Here, the test parameters $(\alpha, \beta)$ can be sampled beyond the original training ranges to simulate scenarios where prior knowledge fails to encompass the true environment space. Specifically, we generate $\alpha \sim \text{Unif}([0.5+\mu_{\text{shift}}, 1.5+\mu_{\text{shift}}]^6)$ and $\beta \sim \text{Unif}([0.05+\mu_{\text{shift}}, 1.05+\mu_{\text{shift}}]^6)$. Four shift levels are evaluated: $\mu_{\text{shift}}=0$ (matching the pre-training distribution) and $\mu_{\text{shift}}=0.1, 0.5, 1$.
> > > > > - **In-domain shifts**: For these shifts,  sub-intervals of length $(1 - \mu_{\text{shift}})$ within the original training ranges are randomly selected, and parameters are sampled uniformly within these sub-intervals. Specifically, for a given $\mu_{\text{shift}}$, we sample $\kappa_1 \sim \text{Unif}([0.5, 1.5 - (1 - \mu_{\text{shift}})])$ and $\kappa_2 \sim \text{Unif}([0.05, 1.05 - (1 - \mu_{\text{shift}})])$. Then, we draw $\alpha \sim \text{Unif}([\kappa_1, \kappa_1 + 1 - \mu_{\text{shift}}]^6)$ and $\beta \sim \text{Unif}([\kappa_2, \kappa_2 + 1 - \mu_{\text{shift}}]^6)$. This approach simulates a scenario in which the prior knowledge is conservative, assuming a broader feasible space than the true space that generates the testing environment. A larger $\mu_{\text{shift}}$ corresponds to more conservative prior knowledge, implying a larger expected feasible parameter space than the actual one. We consider four shift levels: $\mu_{\text{shift}}=0$ (matching the pre-training distribution) and $\mu_{\text{shift}}=0.1, 0.2, 0.3$.
> > > > >
> > > > > The result can been seen in https://docs.google.com/document/d/1bt7MN17L8ttBKZHAKPECEy-PHj5Uwli6VKnlNL7TF48/edit?usp=sharing
> > > > >
> > > > > Insights from experiment II:
> > > > > - For out-of-domain shifts, we observe a performance decline at higher shift levels ($\mu_{\text{shift}}=0.5, 1$) in $(\alpha, \beta)$. This effect is reasonable, as $(\alpha, \beta)$ directly determines the optimal price (action), $a^*_t = \frac{\alpha^\top X_t}{2\beta^\top X_t}$. Therefore, the parameters sampled outside the training distribution's support may lead to optimal decisions that the model has never encountered during training, resulting in performance degradation.
> > > > > - For in-domain shifts, we observe no significant indications of failure of our algorithm in this case. Interestingly, the regret performance of our algorithm and benchmark algorithms improves slightly under these shifts. This enhancement may stem from the shifts leading to $(\alpha, \beta)$ values that are more centered within the range, thereby reducing the occurrence of parameters near the original boundaries that would otherwise be too large or too small. As a result, this shift reduces the likelihood of encountering ``corner'' environments where optimal decisions involve extreme values.
> > > > > - This experiment suggest that the performance decline of out-of-domain shifts can be mitigated by incorporating a larger simulator consisting of wider range of environments, such that the out-of-domain shifts become in-domain shifts.

---

> ### Comment · Reviewer_hvZS · 2024-11-16
> **Response to Authors Part I**
>
> Thanks for you response.
>
> - Please see line 192.
>
> Firstly, you claim that you define a distribution to generate a^*_t, but I don't see a^*_t in your Eq. 5 but an a_\tau. This is the reason why I comment "Confusing notation that the decision function f generates a or a*". Secondly, in your response you claim that you explicitly generate a^* with Eq. 3, I saw that as well. This is also why I'm confused about you claim in line 192. And I'm still confused if you want to keep line 192 unchanged.
>
> - There is no f in Eq. 7, do you mean Eq. 5?
>
> - Thanks for your clarification, if such intuitive explanation is not included in the paper, the authors should put it into the paper somewhere obvious for clarification.
>
> - "However, since these trajectories are simulated and the environment is usually known, the optimal action $a^*$ can be determined using Equation 3." First, it's a strong assumption that the environment is known. Do you study the specific case that the environment is known? In addition, I don't see the causality of this sentence. Can you explain more why the trajectories are simulated and the environment is "usually" known, then a^* can be determined?
>
> To me defining a^* as maximizing the reward is fine. But I don't think it's anything to do with simulated trajectories or known environments.

---

> ### Author Response · Authors · 2024-11-16
> **Second Round Response to hvZS Part I**
>
> Thank you for your quick reply. Please see the response below and let us know if there are further confusions.
>
>
> ```
> ```
>
> - Question: For line 192. Firstly, you claim that you define a distribution to generate $a_t^*$, but I don't see $a_t^*$ in your Eq. 5.
>
> ***Response***
>
>  192 is a long sentence, and we claim that we define a distribution $P_{\gamma, f}$ that generates BOTH $H_t$ and $a_t^*$, where the full specification of $H_t$ (with the necessary notation $a_{\tau}$) is defined in line 193-194, which is Eq. 5; and $a_t^*$ is followed immediately in line 195. We think it follows the natural order for introducing the setups.
> ```
> ```
> - Question: For line 192. Secondly, in your response you claim that you explicitly generate $a_t^*$ with Eq. 3, I saw that as well. This is also why I'm confused about you claim in line 192. And I'm still confused if you want to keep line 192 unchanged.
>
> ***Response***
>
>  I think the confusion comes from the fact we are introducing $H_t$ (which involves $a_{\tau}$ for $\tau = 1, \cdots. t-1$) and $a_t^*$ together. We can update the new version where $H_t$ and $a_{\tau}$ are introduced separately. We hope this would make things more clear. Thank you for the suggestion, and please let us know if you have some better ways to do so.
> ```
> ```
> - Question: There is no $f$ in Eq. 7, do you mean Eq. 5?
>
> ***Response***
>
> Yes we mean Eq.5. Sorry for the confusion.
> ```
> ```
> - Question: Thanks for your clarification, if such intuitive explanation is not included in the paper, the authors should put it into the paper somewhere obvious for clarification.
>
> ***Response***
>
>  Thank you for the suggestion, we will be adding another graph with explanations in the updated version.
> ```
> ```
> - Question: "However, since these trajectories are simulated and the environment is usually known, the optimal action
>  can be determined using Equation 3." First, it's a strong assumption that the environment is known. Do you study the specific case that the environment is known? In addition, I don't see the causality of this sentence. Can you explain more why the trajectories are simulated and the environment is "usually" known, then a^* can be determined?
>
> ***Response***
>
> Thank you for raising this question. We are glad to see that we are discussing the core theme of this paper. Generally speaking, our approach represents a different design philosophy for learning decision-making algorithms, one that is more data/simulator-driven and relies on the learning power of Transformers on large corpora of data.
>
> In the traditional UCB setting, we assume that the environment follows a specific class of distributions (e.g., Gaussian). Based on this assumption, UCB bounds are developed, leading to good theoretical regret and empirical performance. However, this also means that for different distributions and structures (e.g., stochastic bandit environments vs. linear bandit environments), different algorithms are required (e.g., UCB vs. LinUCB [1]). Therefore, if at test time we do not know which environment the data is sampled from, there is no established consensus on which algorithm to apply.
>
> In our design philosophy (illustrated in Figure 1), we sample a large number of environments and sample trajectories from them (denoted as the simulator $\mathcal{S}$ in Figure 1). Although at test time, the current environment $\mathcal{E}$ may be unknown at the initial stage, as time progresses and more trajectory data is observed, the Transformer can identify which pre-training environment the current trajectory most closely resembles. Specifically:
>
> - If the current environment is part of $\mathcal{S}$, the Transformer leverages pretraining data from that environment to generate optimal decisions.
> - If the current environment is not part of $\mathcal{S}$, the Transformer generalizes to make an "educated guess" for the optimal decisions.
>
> Thus, we adopt a simulator-based approach with a large number of environments, which is why the optimal actions $a_t^*$ are known during pre-training. The rationale is that the Transformer can effectively learn to generalize and adapt under a wide variety of trajectories, leading to strong empirical performance.
>
> In Figure 12, we demonstrate the performance of the Transformer, pretrained on two different types of environment distributions with different structures (analogous to stochastic bandit vs. linear bandit) in a decision-making setting. These results highlight the effectiveness of our approach in scenarios where traditional methods are suboptimal.
>
> [1] Li, Lihong, et al. "A contextual-bandit approach to personalized news article recommendation." Proceedings of the 19th international conference on World wide web. 2010.

---

> ### Author Response · Authors · 2024-11-17
> **Further Questions for Reviewer hvZS**
>
> Thank you for raising these suggestions and replying quickly. We truly appreciate your time. We still have two questions regarding your reply.
>
> # First Question
>
> First, we would like to mention that there might be some misunderstanding regarding the claim of our contribution to decision-making. Sequential decision-making problems are not equivalent to RL and include many other problems with practical impacts, such as pricing, inventory management, and control of service systems. We understand that the reviewer might come from a different area, but if you are interested, we kindly invite you to take a few minutes to search these papers [1][2][3], which demonstrate that these settings are long-standing and common in decision-making with practical significance.
>
> In short, these problems differ from RL in that state transitions are sometimes deterministic. For example, after making a decision on inventory replenishment, the state on the next day will deterministically depend on the decision made the previous day, and the randomness comes from other areas, for example the demand in a day. While this represents a simpler version of RL, it still involves transition dynamics and is more complex than stochastic bandit problems due to constraints (such as inventory levels) that affect the decision space.
>
> In our paper, we conduct extensive experiments on pricing and newsvendor problems, which we believe are more complex than simple bandit problems. Given these clarifications, we hope this addresses the reviewer’s concerns about the perceived "over-claim."
>
> # Second Question
>
> We thank the reviewer for the suggestion. We are curious to know: if we address the reviewer’s feedback during the rebuttal period, would you consider raising the score one more time? There are certain issues we believe could be easily fixed in the main text in terms of writing. Additionally, for further discussions or additional numerical studies, we could provide updates in a separate file and incorporate final changes into the main text if everything is resolved.
>
>
> # Conclusion
>
> We thank the reviewer again for the time and effort in this round of review. We look forward to receiving further feedback.
>
>
>
> [1] Petruzzi, Nicholas C., and Maqbool Dada. "Pricing and the newsvendor problem: A review with extensions." Operations research 47.2 (1999): 183-194
>
> [2] Den Boer, Arnoud V. "Dynamic pricing and learning: historical origins, current research, and new directions." Surveys in operations research and management science 20.1 (2015): 1-18.
>
> [3] Besbes, Omar, and Assaf Zeevi. "Dynamic pricing without knowing the demand function: Risk bounds and near-optimal algorithms." Operations research 57.6 (2009): 1407-1420.

---

> ### Author Response · Authors · 2024-11-30
> **Additional Experiment Part I**
>
> According to the reviewers request, we conduct another round of experiment, hoping to provide some insights for the reviewers question.
>
> The reviewer would like to see "**How the distribution mismatch between the pre-training**". We conducted another set of systematic experiments on this topic.
>
> # Problem setting
>
> We take the dynamic pricing problem to illustrate this, given that i) the exploration-exploitation of the dynamic pricing problems is a bit more complex compared to bandit problems; and ii) there are more baseline algorithms to compare with than those of the bandit algorithms.
>
> We define the demand function in the pricing problem as $D(a) = \alpha^\top X - \beta^\top X \cdot a + \epsilon,$ where $(\alpha, \beta)$ represents the parameter set associated with the environment, $X$ denotes the contextual features, $\epsilon$ is random noise, and $a$ is the continuous decision variable corresponding to price. The reward is given by $a \cdot D(a)$. Over a horizon of $T$, our goal is to dynamically set $a_t$ for $t \in \{1, 2, \dots, T\}$ to minimize regret, defined as $E\left[\sum_{t=1}^T a_t^* \cdot D(a_t^*) - \sum_{t=1}^T a_t \cdot D(a_t)\right],$
> where $a_t^*$ is the optimal price in hindsight. For this example, the feature is of dimension $6$.
>
> # Source of distributional mismatch
>
> To show the performance under pre-training and evaluation mismatch, We conduct experiments that analyze distribution mismatch in two aspects.
>  - Adjust the generation of testing environments by altering the distributions of the noise $\epsilon$
>  - Adjust the (generation of) parameters $(\alpha, \beta)$ to deviate from those in the pre-training environments.
>
> # Benchmark algorithms
>
> We compare our algorithm with common off-the-shelf pricing algorithms:
>
> - ILSE: Iterative least square estimation [1]
> - CILS: Constrained iterated least squares [2]
> - TS: Thompson sampling for dynamic pricing [3]
>
> # Experiment I setting
>
> In experiment I, we train the decision-maker under environment with pre-training noise $\epsilon_t\sim\mathcal{N}(0,0.2)$, and vary testing noise distributions such that $\sigma^2 = 0.1$, $\sigma^2 = 0.2$, and $\sigma^2 = 0.3$, separately.
>
> Please see part II for results, details and insights.
>
>
> # Experiment II setting
>
> We further evaluate our model under distributional shifts in the generation of parameters $(\alpha, \beta)$, considering both out-of-domain and in-domain shifts. During pre-training, these parameters are sampled per environment as $\alpha \sim \text{Unif}([0.5, 1.5]^6)$ and $\beta \sim \text{Unif}([0.05, 1.05]^6)$. In the testing phase, two types of shifts are introduced, parameterized by the shift level $\mu_{\text{shift}}$:
>
> - **Out-of-domain shifts**: Here, the test parameters $(\alpha, \beta)$ can be sampled beyond the original training ranges to simulate scenarios where prior knowledge fails to encompass the true environment space. Specifically, we generate $\alpha \sim \text{Unif}([0.5+\mu_{\text{shift}}, 1.5+\mu_{\text{shift}}]^6)$ and $\beta \sim \text{Unif}([0.05+\mu_{\text{shift}}, 1.05+\mu_{\text{shift}}]^6)$. Four shift levels are evaluated: $\mu_{\text{shift}}=0$ (matching the pre-training distribution) and $\mu_{\text{shift}}=0.1, 0.5, 1$.
> - **In-domain shifts**: For these shifts,  sub-intervals of length $(1 - \mu_{\text{shift}})$ within the original training ranges are randomly selected, and parameters are sampled uniformly within these sub-intervals. Specifically, for a given $\mu_{\text{shift}}$, we sample $\kappa_1 \sim \text{Unif}([0.5, 1.5 - (1 - \mu_{\text{shift}})])$ and $\kappa_2 \sim \text{Unif}([0.05, 1.05 - (1 - \mu_{\text{shift}})])$. Then, we draw $\alpha \sim \text{Unif}([\kappa_1, \kappa_1 + 1 - \mu_{\text{shift}}]^6)$ and $\beta \sim \text{Unif}([\kappa_2, \kappa_2 + 1 - \mu_{\text{shift}}]^6)$. This approach simulates a scenario in which the prior knowledge is conservative, assuming a broader feasible space than the true space that generates the testing environment. A larger $\mu_{\text{shift}}$ corresponds to more conservative prior knowledge, implying a larger expected feasible parameter space than the actual one. We consider four shift levels: $\mu_{\text{shift}}=0$ (matching the pre-training distribution) and $\mu_{\text{shift}}=0.1, 0.2, 0.3$.
>
> Please see part II for results, details and insights.
>
>
> [1] Keskin, N. Bora, and Assaf Zeevi. "Dynamic pricing with an unknown demand model: Asymptotically optimal semi-myopic policies." Operations research 62.5 (2014): 1142-1167.
>
> [2] Qiang, Sheng, and Mohsen Bayati. "Dynamic pricing with demand covariates." arXiv preprint arXiv:1604.07463 (2016).
>
> [3] Wang, Hanzhao, Kalyan Talluri, and Xiaocheng Li. "On dynamic pricing with covariates." arXiv preprint arXiv:2112.13254 (2021).

---

> ### Author Response · Authors · 2024-11-30
> **Response to hvZS Part III**
>
> Hi reviewer hvZS,
>
> We have spent several days conducting an additional set of experiments, and you can find the results here:
> https://openreview.net/forum?id=CiiLchbRe3&noteId=40PEGFrve4
>
> We hope this provides a relatively detailed analysis regarding the distribution mismatch between the pre-training and evaluation stages.
>
> We are happy to update the writing as you suggested and include these additional experimental results in the revised version of the paper.
>
> Since the conversation window is closing in 3 days, please let us know if there is anything else you would like to know. We also hope the reviewer can re-evaluate our work based on these updated results.

---

### Official Review · Reviewer_uvAS · 2024-11-02

**Soundness:** 3
**Presentation:** 3
**Contribution:** 2
**Rating:** 5
**Confidence:** 2

**Summary:**

The authors explore the implications of using pre-trained transformers on bandit-style decision-making problems. They perform a scientific analysis that yields interesting insights, and provide empirical case studies that show that the use of pre-trained transformers yields competitive results with existing methods.

**Strengths:**

This paper offers an analytical analysis of using pre-trained transformers on bandit-style problems and demonstrates the benefits exceeding the typical asymtotic optimal regret. Along the way, the paper provides interesting insights, and the resulting algorithm is well-motivated and explained clearly.

**Weaknesses:**

The main weakness can be the narrow scope of impact as the sequential decision making problem in the paper is very narrowly defined.

The text does very little to motivate the need for the Learned Decision Function (LDF) described in Section 4. In particular, the relationship between the LDF and the pre-trained methods described in this paper is unclear on a first read. The authors need to better explain why the LDF is necessary. It seems unlikely that there is not an acceptable baseline that can be used from the literature. The confusion is further amplified in the experiments section, where the authors use different baselines to evaluate the pre-trained methods.

Additionally, while the insights generated are interesting, the paper lacks a clear and compelling rationale for why the pre-training is worth the extra computation compared to simpler methods. In particular, the use of a complex transformer architecture on a bandit-style problem would, on the surface, appear to be ‘overkill’, and the paper is not able to clearly articulate a compelling reason that justifies the use of such a complex solution method.

Ovearll, math notation is confusing and seems even incorrect/inconsistent.

Some minor comments:
•	Figures 1 and 2 are not referenced anywhere in the main text.

**Questions:**

1.	Do the benchmarks used need pre-training? If not, then what would the authors argue is the benefit of the pre-trained method given the increased computation that is required for the pre-training?
2.     In line 167, it is stated that "...reinforcement learning algorithms and are usually hard to combine with prior knowledge such as pre-training data." However, I believe there are many options to combine prior knowledge in previously collected data: initializing replay buffer, offline learning, transfer learning, ....

---

> ### Author Response · Authors · 2024-11-14
> **Response to Reviewer uvAS Part I**
>
> We appreciate the reviewer’s questions and suggestions. It seems there may be some misunderstandings about the specific setup and other details of our paper. We hope our response clarifies these points and kindly invite the reviewer to re-evaluate our contributions in light of this clarification.
> ```
> Weakness 1: The main weakness can be the narrow scope of impact as the sequential decision making problem in the paper is very narrowly defined...
> ```
> We acknowledge that our setting does not cover the general RL framework. However, we would like to emphasize that this is a unified approach, encompassing nearly all other settings—such as various types of bandits, dynamic pricing, and inventory control. These combined research areas already represent a significant and impactful community.
>
> Moreover, this work represents an essential step toward artificial general intelligence (AGI), which holds significant implications for the future of AI. If we do not analyze it within the bandit setting as a foundation, how can we extend it to more general RL or practical applications? For instance, the well-known online RL algorithm UCBVI [1] builds on the development of UCB in bandit problems. Therefore, we believe that studying the transformer framework in the bandit setting is both meaningful and necessary.
> ```
> Weakness 2a: The text does very little to motivate the need for the Learned Decision Function (LDF)...
> ```
> We think there might be some misunderstanding regarding the setup. Generally speaking, the Learned Decision Function (LDF), which is the transformer-based decision-maker, is defined in Section 2, and the pretraining method aims to generate an LDF with good performance. The section 4 analyzing LDF is necessary because we need to analyze the theoretical performance of the transformer-based decision-maker, such as the realizability and the regret bound, which are very common and necessary results in the decision-making literature (e.g., bandits, pricing). We were indeed surprised by this question asking for motivations.
>
> To clarify the relationship: In Section 2 (Page 3, Line 126), we define the LDF, which generates actions from the Transformer as $a_t = TF_{\theta}(H_t)$. The training loss is generally defined as $\mathbb{E}[\sum_{t=1}^T l(TF_{\theta}(H_t), a_t^*)]$ (Page 5, Line 220).
>
> In Section 3, we discuss properties of the pretraining and generalization loss, while in Section 4, we analyze the properties of the actions generated by the LDF. Specifically, we identify the Bayes-optimal decision function (Page 7, Line 367), which is the optimal decision. We then show in Proposition 4.1 that the Bayes-optimal decision function minimizes our training loss. This implies that by minimizing the empirical loss, the trained transformer is likely to perform close to the Bayes-optimal decision function. Based on this, Proposition 4.4 provides a regret analysis for the LDF, which is a core component of almost every theoretical decision-making paper.
>
> Therefore, the analysis of the LDF is fundamental to demonstrating important theoretical properties related to decision-making. We hope this clarifies why the LDF is essential.
>
> ```
> Weakness 2b: It seems unlikely that there is not an acceptable baseline that can be used from the literature. The confusion is further amplified in the experiments section, where the authors use different baselines to evaluate the pre-trained methods...
> ```
> In summary, the only baseline in [36], from which our method is developed, does not converge and therefore cannot be used as a baseline. Additionally, we cannot use the same baseline across different problems such as stochastic bandits, linear bandits, pricing, and the newsvendor problem, because each problem is distinct, requiring problem-specific algorithms as baselines.
>
> For the first point, our method, Pre-trained Transformer (PT), originates from [36], which introduces a new architecture and serves as the only benchmark baseline. A key contribution of our paper is improving the pretraining pipeline from [36], enabling our improved PT to converge when pretrained on a large number of environments, while the method in [36] fails to do so (see Figure 2(a)). Therefore, it does not make sense to use the only available baseline in the literature, as it does not converge.
>
> For the second point, we cannot use the same baseline for stochastic bandits, linear bandits, pricing, and the newsvendor problem because these problems differ significantly, and only problem-specific algorithms can serve as appropriate baselines. For instance, UCB for bandit algorithms cannot be directly applied to pricing settings.
>
> ***Reference***
>
>
> [1] Azar, Mohammad Gheshlaghi, Ian Osband, and Rémi Munos. "Minimax regret bounds for reinforcement learning." International conference on machine learning. PMLR, 2017.
>
> [36] Lee, Jonathan, et al. "Supervised pretraining can learn in-context reinforcement learning." Advances in Neural Information Processing Systems 36 (2024).

---

> > ### Comment · Reviewer_uvAS · 2024-11-24
> >
> > The authors said "Moreover, this work represents an essential step toward artificial general intelligence (AGI), which holds significant implications for the future of AI. If we do not analyze it within the bandit setting as a foundation, how can we extend it to more general RL or practical applications?"
> >
> > However, it sounds like a big stretch and I hope to see the value of the findings in more general sequential decision making situations. While authors mention dynamic pricing and news vendor problem, I hope to see a more technical description of problems that the proposed method can be useful.
> >
> > Another question: why there is not transition probabilities for dynamic pricing? I thought that the demands are stochastic and state (usually defined as inventory levels) makes stochastic transitions.

---

> > > ### Author Response · Authors · 2024-11-25
> > > **Response to Reviewer uvAS Round 2**
> > >
> > > We thank the reviewer for the reply and the additional questions. We have provided detailed explanation below, which hopefully will make things clearer. We also hope the reviewer could reconsider the contribution of this paper in the literature, and re-evaluate our work.
> > >
> > > ```
> > > Question 1: I hope to see the value of the findings in more general sequential decision making situations. While authors mention dynamic pricing and newsvendor problem, I hope to see a more technical description of problems that the proposed method can be useful.
> > > ```
> > >
> > > In short, given that this paper already contains many new results (40 pages of work with both theoretical and empirical developments), we think it is reasonable to defer the investigation of RL to future work. Now please see our detailed response to this question.
> > >
> > > - First, to the best of our knowledge, in this simple bandit setting, we have not seen theoretical developments in terms of pre-training loss and regret analysis that are comparable to our setting (all the relevant ones impose strong assumptions), and this already has significant merit. Every foundational RL or sequential decision-making algorithm needs to be based on foundations for simple environments like bandit problems. For example, consider the advancement from UCB in bandit problems to UCBVI in RL—we cannot develop UCBVI from nowhere without the insights gained from bandit problems.
> > >
> > > - Second, we believe that in the RL setting, the idea and formulation would be quite similar, utilizing sampled trajectories to pretrain the model. One aspect that could make the setting more challenging in more general sequential decision-making situations is that the complexity bounds and computational time will depend on the time horizon of the RL environment. This could make pretraining computationally expensive.
> > >
> > > Therefore, given that this paper already contains many new results, we think it is reasonable to defer the investigation of RL to future work.
> > >
> > > ```
> > > Question 2: Another question: why there is not transition probabilities for dynamic pricing? I thought that the demands are stochastic and state (usually defined as inventory levels) makes stochastic transitions.
> > > ```
> > > In short, this is because the randomness are coming from the "reward function" (note that the demand is related to the revenue, which corresponds to the general definition of "reward function"), and are not from the inherent randomness of the transition dynamics. This is similar to the bandit environment, where the randomness comes from the reward of each arm.
> > >
> > > To be more specific, the way we compute the remaining inventory follows a deterministic formula, that is,
> > >
> > > $S_{t+1} = \max${$S_{t} - D_t(p_t), 0$},
> > >
> > > where $S_{t+1}$ is the remaining inventory, $S_t$ is the current inventory, and $D_t$ is the demand for today, which depends on today’s price $p_t$. Given today’s inventory level, the randomness only comes from the demand for today and depends (randomly) on the price $p_t$ we set.
> > >
> > > Therefore, this kind of "structured transition"is different from the RL setting, where the randomness in the transition dynamics is governed by a separate source, distinct from the randomness in the reward function.
> > >
> > > You can think of the randomness in demand as analogous to the randomness in the reward function in the RL setting. Once $D_t(p_t)$ is sampled, the entire dynamic is deterministic. Therefore, this inventory problem behaves more like a bandit problem, where the randomness only comes from the reward, and after the reward is sampled, there is no randomness when transitioning to the next state.
> > >
> > > # Conclusion
> > >
> > > We wish our reply make things clearer, and hope that the reviewer could take some time for re-evaluation. Please let us know if you have further questions.

---

> ### Author Response · Authors · 2024-11-14
> **Response to Reviewer uvAS Part II**
>
> ```
> Weakness 3: Additionally, while the insights generated are interesting, the paper lacks a clear and compelling rationale for why the pre-training is worth the extra computation compared to simpler methods...
> ```
> In terms of this weakness, we think the reviewer may have overlooked two important factors.
>
> 1. **General Algorithm For Decision-Making**
>
> This algorithm is not specifically designed for bandit problems. It is a pipeline intended to be broadly applicable across various settings, including stochastic bandits, linear bandits, dynamic pricing, and newsvendor problems. The general pipeline has its merit in designing unified frameworks as a step toward artificial general intelligence (AGI).
>
> Moreover, the theoretical analysis is a fundamental step in understanding Transformers for decision-making problems. If we do not analyze it within the bandit setting as a foundation, how can we extend it to more general RL or practical applications? For instance, the well-known online RL algorithm UCBVI [1] builds on the development of UCB in bandit problems. Therefore, we believe that studying the transformer framework in this simpler setting is both meaningful and necessary.
>
> 2. **Applicability to Complex Decision-Making Problems**
>
> We would like to highlight that our pipeline is capable of handling more complex yet practical problems beyond simple bandits. In real-world decision-making scenarios, such as dynamic pricing and newsvendor problems, it is often necessary to switch between different models depending on the environment. To make an analogy to the bandit setting, an environment may alternate between a linear bandit problem (requiring the LinUCB [2] algorithm to perform well) and a stochastic bandit problem (requiring the UCB algorithm). A unified framework like our transformer-based approach effectively handles these environments by identifying the environment and selecting the right algorithm when seeing the current trajectory, as demonstrated in Figure 12. This versatility extends the applicability of our method beyond simple bandit problems to more complex decision-making tasks.
> ```
> Weakness 4: Overall, math notation is confusing and seems even incorrect/inconsistent.
> ```
> We would appreciate it if you could provide specific examples of the issues you found. Raising such concerns without explicitly identifying the problems makes it challenging to address them and, in our view, is both irresponsible and unprofessional. Clear feedback would help us resolve any potential misunderstandings more effectively.
> ```
> Question 1: Do the benchmarks used need pre-training? If not, then what would the authors argue...
> ```
> This represents a different design philosophy. Previously, for every problem instance, problem-specific algorithms (which are the benchmark algorithms) had to be developed, such as UCB for bandits and LinUCB [2] for linear bandits. We propose a unified framework that is pretrained on trajectories sampled from a massive number of environments. In the evaluation phase, by observing the current trajectory, it effectively identifies the environment and selects the "right" algorithm.
>
> This unified approach is successful across various decision-making problems, as demonstrated in our numerical examples. It is more flexible in terms of algorithm design, provides better performance than problem-specific benchmarks (figure 4), and is generalizable as the number of environments in the simulators $\mathcal{S}$ increases, allowing us to handle more complex environments (figure 12).
>
> ```
> Question 2: In line 167, it is stated that...
> ```
> We would like to point out that the pretrained transformer is trained on a massive number of trajectories from a large number of environments, whereas previous methods that incorporate prior knowledge from previously collected data—such as replay buffers, offline learning, and transfer learning—cannot effectively leverage this heterogeneous offline data without knowing the current environment. It is straightforward to construct counterexamples where trajectories from unrelated environments adversely impact algorithm performance.
>
> Developing algorithms that can effectively leverage this type of offline data in a online manner requires a dedicated study, which current offline RL and transfer learning techniques do not adequately address. In this setting, our unified framework, pretrained on trajectories sampled from a wide range of environments, can effectively identify the environment and select the "right" action.
>
> ***References***
>
> [1] Azar, Mohammad Gheshlaghi, Ian Osband, and Rémi Munos. "Minimax regret bounds for reinforcement learning." International conference on machine learning. PMLR, 2017.
>
> [2] Li, Lihong, et al. "A contextual-bandit approach to personalized news article recommendation." Proceedings of the 19th international conference on World wide web. 2010.

---

> ### Author Response · Authors · 2024-11-30
> **Reminder and Additional Experiments**
>
> Hi reviewer uvAS,
>
> We hope this message finds you well. Since we haven’t heard back from you in a while, we wanted to remind you that the conversation window is closing in 3 days.
>
> Additionally, as the reviewer hvZS suggested, we have spent several days conducting additional experiments, which provide a detailed analysis of the distribution mismatch between the pre-training and evaluation stages.
>
> You can find the results here: https://openreview.net/forum?id=CiiLchbRe3&noteId=40PEGFrve4
>
> We are happy to update the writing as you suggested and include these additional experimental results in the revised version of the paper.
>
> Please let us know if there is anything else you would like to know. We also hope you can re-evaluate our work based on these updated results.

---

### Official Review · Reviewer_Svgy · 2024-11-04

**Soundness:** 3
**Presentation:** 3
**Contribution:** 2
**Rating:** 5
**Confidence:** 2

**Summary:**

This paper investigates the training and generalization of pre-trained transformers for sequential decision-making tasks without transition probabilities. The authors propose an algorithm that incorporates transformer-generated action sequences during pretraining, establishes a connection to performative prediction, and addresses the challenge of limited exploration. They highlight three advantages of pre-trained transformers over structured algorithms like UCB and TS: improved utilization of pretraining data, robustness to model misspecification, and enhanced short-term performance.

**Strengths:**

- The paper introduces notation and a framework for studying pre-trained transformers in decision-making tasks.
- It proposes a new algorithm and demonstrates its practical performance.
- The connection to performative prediction is intriguing.

**Weaknesses:**

- My biggest concern is the lack of discussion regarding the disadvantages of using transformers for tasks like bandits. The paper does not sufficiently address the computational and memory overhead involved.

- The comparison between the proposed method and UCB/TS seems unfair, as the transformer benefits from extensive prior data while UCB/TS is evaluated in a cold-start setting. Claiming that UCB/TS is less effective at utilizing prior knowledge is misleading since these algorithms aren’t provided with any pre-training data. In fact, UCB/TS algorithms do incorporate observed data as part of their operation, in the form of empirical mean rewards and reward uncertainty.

- The claimed short-term advantage of the proposed method also seems unconvincing. While it’s true that UCB/TS in their original forms is designed for asymptotic optimality rather than short-term performance, there are likely modifications or more state-of-the-art structured algorithms that can improve short-term performance. For instance, restricting the variance (exploration) in these algorithms could make them more greedy. I’m not convinced that this is an inherent limitation of structured algorithms.

- I am not very familiar with related work on applying deep learning or transformer methods to bandits, so a more comprehensive related work section would be helpful. For example, some studies use generated data as priors for algorithms like TS.

- The clarity and organization of the paper could be improved. For instance, while the paper emphasizes reinforcement learning and sequential decision-making settings, I believe the studied problem is more aligned with the contextual bandit setting. I believe some of the notations on pages 3-5 could be streamlined, and a discussion in the main text about how these settings differ and why the setting studied in the paper is more general would be beneficial.

**Questions:**

- What is the difference between the setting described in the paper (reinforcement learning/sequential decision-making without transition probabilities) and the stochastic bandit or contextual bandit settings? If it aligns with an existing setting, that should be clarified. The use of reinforcement learning terminology here seems to add more confusion than clarity.

- What are the disadvantages of using pre-trained transformers compared to structured algorithms like UCB and Thompson Sampling? One notable concern could be the computational and memory cost; can this be quantified? For instance, how would the algorithms compare if the x-axis were aligned with computational metrics? What is the computational or memory overhead compared to simpler methods? Given that the proposed method is intended to be practical but lacks theoretical rigour, it is important to assess whether using these algorithms makes sense in practice.

- If UCB/TS had access to the same amount of pretraining data, how would their performance compare to the proposed method?

---

> ### Author Response · Authors · 2024-11-13
> **Response to Reviewer Svgy**
>
> We appreciate the reviewer’s questions and suggestions. It seems there may be some misunderstandings about the specific setup and other details of our paper. We hope our response clarifies these points and kindly invite the reviewer to re-evaluate our contributions in light of this clarification.
> ```
> Weakness 1: My biggest concern is the lack of discussion ...
> ```
> We do not see significant issues with respect to the computational and memory overhead. In terms of computational overhead, our model scales linearly with respect to the context length, as is the case with many generative language models, which is totally acceptable. In terms of memory, our transformer adopts the GPT-2 architecture, which requires less than 1 gigabyte of VRAM and can run on almost all GPUs compatible with CUDA. Therefore, we don’t think there is a significant issue in terms of inference speed or memory requirements.
> ```
> Weakness 2: The comparison between the proposed method and UCB/TS seems unfair, as the transformer benefits from...
> ```
> We believe there may be a misunderstanding of our setup. Our transformer is pretrained on a vast number of environments, with each environment generating hundreds of thousands of trajectories. In the evaluation or online learning phase, if the current environment and its sampled trajectories were known, we could leverage that data in combination with the UCB/TS algorithms. However, since the environment is unknown beforehand, using the large volume of offline data indiscriminately could potentially degrade performance. While some algorithms may leverage massive pretraining data from heterogeneous environments online, fully understanding their analytical and computational properties would require a dedicated study, which is beyond the scope of this paper. For this reason, we adhere to the original UCB/TS algorithms in our comparisons.
> ```
> Weakness 3: The claimed short-term advantage of the proposed method also seems unconvincing. While it’s true that...
> ```
> To address this question, we would like to reiterate our main contribution: a stable and unified pretraining pipeline for transformers in the context of decision-making, with both theoretical guarantees and strong empirical performance. This pipeline is designed to be broadly applicable across various settings, including stochastic bandits, linear bandits, dynamic pricing, and newsvendor problems.
>
> We believe it would be inappropriate to compare our method with problem-specific algorithms, as those algorithms are tailored to individual problem instances rather than offering a unified framework. The strength of our approach lies in its generality and versatility across different problem domains, which is distinct from optimizing performance in specific scenarios.
> ```
> Weakness 4: I am not very familiar with related work on applying deep learning...
> ```
> We thank the reviewer for suggesting this, and we will put this in the updated version.
> ```
> Weakness 5: The clarity and organization of the paper could be improved...
> ```
> We thank the reviewer for suggesting this, and we will put this in the updated version.
> ```
> Question 1: What is the difference between the setting described in the paper...
> ```
> The setting in our paper encompasses both the stochastic bandit and contextual bandit settings. In Appendix B, we provide detailed examples that fall within this framework. We believe this is the most general formulation, as it includes these specific settings. A contextual bandit framework, by contrast, would be considerably less general.
> ```
> Question 2: What are the disadvantages of using pre-trained transformers compared to structured algorithms like UCB and Thompson Sampling...
> ```
> The computational cost grows linearly (similar to the UCB and TS algorithms) with respect to the context length, as it corresponds to the number of tokens that need to be evaluated by the large language model, which we believe is entirely acceptable in practice. In terms of memory, our transformer is based on the GPT-2 architecture, requiring less than 1 GB of VRAM and running on most CUDA-compatible GPUs. Therefore, we do not anticipate significant practical issues with inference speed or memory requirements.
> ```
> Question 3: If UCB/TS had access to the same amount of pretraining data, how would...
> ```
> As mentioned earlier, for UCB/TS to effectively leverage the pretraining data, they would need to know the current environment and its corresponding trajectories. Without this knowledge, it is straightforward to construct counterexamples where trajectories from unrelated environments could adversely impact UCB/TS performance by distorting the confidence bounds or posterior estimates.

---

> > ### Comment · Reviewer_Svgy · 2024-11-13
> > **Response to author**
> >
> > Thank you for the clarification.
> >
> > My concern about computational and memory overhead (or other potential drawbacks) was specifically in comparison to running TS/UCB in practice, rather than to other generative language models. I remain somewhat unconvinced that, in practice, the more complex pre-trained transformer approach offers a clear advantage or necessity over simpler structured methods like TS/UCB in bandit problems, especially given the costs associated with pre-training on a large number of environments and trajectories. In reinforcement learning, I understand that it’s often challenging to estimate every state-action pair due to the complexities introduced by the transition matrix; therefore, having access to a good simulator can make pre-training models offline to narrow down the scope beneficial. However, in the simpler bandit setting, TS/UCB-type algorithms can already learn optimally from scratch, so I’m not sure why pre-training a more complicated model would be preferable.
> >
> > Again, I do see the conceptual value of the framework itself and its connection to performative prediction, but the concern above raises questions about the practical significance of the contribution. I’m not an expert in the field, but with my best understanding, I’ll maintain my score.

---

> > > ### Author Response · Authors · 2024-11-13
> > > **2nd Round Response**
> > >
> > > We thank the reviewer for their quick feedback. While we understand the concern, we respectfully disagree with the perspective on the necessity of adopting a transformer-based decision-making agent. We hope the following clarification provides a clearer picture and kindly ask the reviewer to reconsider after reviewing our explanation.
> > >
> > > # General Considerations
> > >
> > > The reviewer may have overlooked two important factors.
> > >
> > > 1. **Applicability to Complex Decision-Making Problems**
> > >
> > > We would like to highlight that our pipeline is capable of handling more complex yet practical problems beyond simple bandits. In real-world decision-making scenarios, such as dynamic pricing and newsvendor problems, it is often necessary to switch between different models depending on the environment. To make an analogy to the bandit setting, an environment may alternate between a linear bandit problem (requiring the LinUCB algorithm to perform well [1]) and a stochastic bandit problem (requiring the UCB algorithm). A unified framework like our transformer-based approach effectively handles these environments by identifying the environment and selecting the right algorithm when seeing the current trajectory, as demonstrated in Figure 12. This versatility extends the applicability of our method beyond simple bandit problems to more complex decision-making tasks. We provide a detailed discussion of the pricing settings at the end of this response.
> > >
> > > 2. **Foundational Step Toward General AI**
> > >
> > > This approach is an essential step toward artificial general intelligence (AGI), which holds significant implications for the future of AI. If we do not analyze it within the bandit setting as a foundation, how can we extend it to more general RL or practical applications? For instance, the well-known online RL algorithm UCBVI [2] builds on the development of UCB in bandit problems. Therefore, we believe that studying the transformer framework in the bandit setting is both meaningful and necessary.
> > >
> > > # Practical Relevance of Pretraining in Decision-Making Problems
> > >
> > > Next, we elaborate on why the transformer-based decision-making agent is practical for our decision-making problems.
> > >
> > > In the dynamic pricing example (a widely studied subject, see [3]), imagine a company (e.g., Amazon) that sells products to customers. At time $t$, the company needs to set the price $p_t$, which corresponds to a demand function $d(p_t)$, with the goal of dynamically maximizing revenue $\sum_{t=1}^T p_t \cdot d(p_t)$. In practice, the demand pattern changes from week to week—sometimes $d(p)$ is a linear function, and other times it is polynomial.
> > >
> > > To achieve maximal revenue, two issues need to be addressed:
> > >
> > > * Identify the environment and the right algorithm: this is important because using an algorithm designed for linear demand will perform poorly for an environment with polynomial demand.
> > > * Quickly adapt: Often, it is impractical to start learning from scratch each time since the demand pattern changes frequently. The algorithm must quickly find the near-optimal price.
> > >
> > > Therefore, it is necessary to pretrain the decision-transformer on many simulated environments. As shown in Figure 1, the larger the simulator environment $\mathcal{S}$, the more likely the transformer will have encountered patterns observed in the real environment. Once the transformer identifies these patterns, it can quickly adapt and find the near-optimal price. The transformer we developed is highly effective in addressing this setting (as shown in Figures 4(c) and 4(d), and Figure 12 in the appendix).
> > >
> > >
> > >
> > >
> > >
> > > [1] Azar, Mohammad Gheshlaghi, Ian Osband, and Rémi Munos. "Minimax regret bounds for reinforcement learning." International conference on machine learning. PMLR, 2017.
> > >
> > > [2] Li, Lihong, et al. "A contextual-bandit approach to personalized news article recommendation." Proceedings of the 19th international conference on World wide web. 2010.
> > >
> > > [3] Den Boer, Arnoud V. "Dynamic pricing and learning: historical origins, current research, and new directions." Surveys in operations research and management science 20.1 (2015): 1-18.

---

> ### Author Response · Authors · 2024-11-30
> **Reminder and Additional Experiments**
>
> Hi reviewer Svgy,
>
> We hope this message finds you well. Since we haven’t heard back from you in a while, we wanted to remind you that the conversation window is closing in 3 days.
>
> Additionally, as the reviewer hvZS suggested, we have spent several days conducting additional experiments, which provide a detailed analysis of the distribution mismatch between the pre-training and evaluation stages.
>
> You can find the results here:
> https://openreview.net/forum?id=CiiLchbRe3&noteId=40PEGFrve4
>
> We are happy to update the writing as you suggested and include these additional experimental results in the revised version of the paper.
>
> Please let us know if there is anything else you would like to know. We also hope you can re-evaluate our work based on these updated results.

---

### Official Review · Reviewer_8XDh · 2024-11-05

**Soundness:** 4
**Presentation:** 3
**Contribution:** 3
**Rating:** 6
**Confidence:** 3

**Summary:**

This paper presents a supervised training framework for transformers applied to sequential decision-making problems—a subset of reinforcement learning (RL) tasks that lack an explicit transition matrix. In this framework, transformers are pre-trained on the target task class by simulating multiple environments to generate trajectories consisting of states, actions, and rewards. However, a discrepancy exists between action selection during pre-training and at test time, which leads to a gap between the empirical loss observed in training and the expected loss during testing. This gap can cause out-of-distribution issues, as studied by previous literature, causing the regret bound of the model’s performance to increase exponentially over time. To address this, the authors propose a mixed pre-training algorithm, in which the models learn not only from simulated data but also from trajectories generated by their own actions. This hybrid approach reduces the discrepancy between training and testing distributions. The proposed algorithm is theoretically validated and supported by experimental results.

**Strengths:**

1. This paper applies transformers in the domain of sequential decision-making, typically dominated by traditional RL approaches. The authors ground their framework with rigorous theoretical proofs, establishing conditions under which the mixed pre-training algorithm minimizes cumulative regret. They also prove that the learned decision function asymptotically matches the Bayes-optimal policy as the data approaches infinity.

2. I like how the authors propose the two-phase training strategy to effectively mitigate out-of-distribution problems by including the model's self-generated actions in the second training phase. This solution is simple yet effective and also maintains theoretical rigor.

**Weaknesses:**

1. I think the setup in Section 2 goes on a bit too long and could be more concise. A detail: the way f is used in Equation (4) isn’t fully explained until after Equation (5), which makes it hard to follow. Plus, describing f as "a prespecified decision function used to generate the data" is too vague.

2. The framework leans heavily on simulated environments ($gamma_i$ and $f$) for pre-training, which might limit scalability to real-world cases. Generating high-quality simulations can be both costly and challenging.

**Questions:**

1. How do the authors choose the environments for pre-training, and do variations in the gamma values during training impact the model’s ability to generalize? It would be interesting to know if a big difference between training and testing gamma values affects performance or creates bias.

2. Have the authors thought about using curriculum learning, where self-generated actions gradually increase over time, instead of the two-phase switch? I think it would enjoy less theoretical property, but possibly perform better in numerical experiments.

---

> ### Author Response · Authors · 2024-11-22
> **Response to Reviewer 8XDh Part I**
>
> We thank the reviewer for these comments. Please see our reply below, and we hope this will make things clearer. Please let us know if we have addressed your questions. We would greatly appreciate it if the reviewer could re-evaluate our work based on this round of response.
>
>
> ```
> Weakness 1: I think the setup in Section 2 goes on a bit too long and could be more concise...
> ```
>
> We thank the reviewer for the suggestion. We will update it in the next version.
>
> ```
> Weakness 2: The framework leans heavily on simulated environments (and) for pre-training, which might limit scalability to real-world cases. Generating high-quality simulations can be both costly and challenging.
> ```
>
> We would like to mention that the design philosophy of our approach is quite different from the traditional decision-making approach. We hope the reviewer can read our reply below to gain a better understanding of our approach. In short, our pipeline is different in that:
>
> - It is a unified approach that can solve a wide variety of decision-making problems (same pre-training approach) under different environments (we just sample these environments for pre-training). By sharing a uniform pre-training structure. This does not require users to design instance-specific (bandit vs pricing) or environment-specific (stochastic bandit vs linear bandit) algorithms.
> - Although it works in a simulated environment, the transformer architecture is capable of generalizing and extracting specific patterns from the trajectories it has seen. Therefore, as long as the simulator is large enough to include trajectories that is similar to the current environment, we can count on the transformer to quick identify the environment and adapt to the corresponding optimal algorithms.
>
> While we do agree that it is not scalable for complicated RL problems in practice, it is applicable for real-world business problems including dynamical pricing and inventory problems, as suggested in Figure 12.
>
> # Details
> To be more specific on those two features mentioned above. In the traditional UCB setting, we assume that the environment follows a specific class of distributions (e.g., Gaussian). Based on this assumption, UCB bounds are developed, leading to good theoretical regret and empirical performance. However, this also means that for different distributions and structures (e.g., stochastic bandit environments vs. linear bandit environments), different algorithms are required (e.g., UCB vs. LinUCB [1]). Therefore, if at test time we do not know which environment the data is sampled from, there is no established consensus on which algorithm to apply.
>
> In our design philosophy (illustrated in Figure 1), we sample a large number of environments and sample trajectories from them (denoted as the simulator $\mathcal{S}$ in Figure 1). Although at test time, the current environment $\mathcal{E}$ may be unknown at the initial stage, as time progresses and more trajectory data is observed, the Transformer can identify which pre-training environment the current trajectory most closely resembles. Specifically:
>
> - If the current environment is part of $\mathcal{S}$, the Transformer leverages pretraining data from that environment to generate optimal decisions.
> - If the current environment is not part of $\mathcal{S}$, the Transformer generalizes to make an "educated guess" for the optimal decisions.
>
> Thus, we adopt a simulator-based approach with a large number of environments. The rationale is that the Transformer can effectively learn to generalize and adapt under a wide variety of trajectories, leading to strong empirical performance.
>
>
> [1] Li, Lihong, et al. "A contextual-bandit approach to personalized news article recommendation." Proceedings of the 19th international conference on World wide web. 2010.

---

> ### Author Response · Authors · 2024-11-22
> **Response to Reviewer 8XDh Part II**
>
> Please see the following replies to your questions. We hope our responses make things clearer. Please let us know if we have addressed your questions. We would greatly appreciate it if the reviewer could re-evaluate our work based on this round of responses.
>
> ```
> Question 1 part I: How do the authors choose the environments for pre-training, and do variations in the gamma values during training impact the model’s ability to generalize?
> ```
>
> The environments are chosen in the following ways.
>
> - **Similar environment with different parameters**: This is the most standard setup for pre-training environments. For example, the bandit environments in Figure 4 and Figure 16 are pre-trained using trajectories sampled from one category of distribution (e.g., Gaussian) with different parameter setups (e.g., different means and variances). Our performance under this setting is quite strong.
>
>
> - **Different environment with different structures**: We also include pre-training data from different categories of environments. In practice, we may not know if the current environment follows a stochastic bandit environment or a contextual bandit environment. Standard bandit algorithms require prior knowledge of the environment to design specific algorithms for each case. In contrast, our model in the evaluation phase aims to identify the correct class of environment and adapt to the optimal algorithm during evaluation, enabling in-context model selection. The strong empirical results of this approach are shown in Figure 12.
>
> ```
> Question 1 part II: It would be interesting to know if a big difference between training and testing gamma values affects performance or creates bias.
> ```
>
> If the gamma in the testing environment has been seen in the pre-training environment, our model performs well compared to standard baselines (see Figure 4). If the gamma in the testing environment is very out-of-distribution (OOD), then, as shown in Figure 16, the performance of our approach is slightly affected. However, it is less affected compared to some standard baselines except Thompson Sampling, and still outperforms all of them by a significant margin.
>
>
> ```
> Question 2: Have the authors thought about using curriculum learning, where self-generated actions gradually increase over time, instead of the two-phase switch? I think it would enjoy less theoretical property, but possibly perform better in numerical experiments.
> ```
>
> Yes, we used curriculum learning in pre-training. Please see page 36 (Appendix E) for the details in our curriculum training.

---

> ### Author Response · Authors · 2024-11-30
> **Reminder and Additional Experiments**
>
> Hi reviewer 8XDh,
>
> We hope this message finds you well. Since we haven’t heard back from you in a while, we wanted to remind you that the conversation window is closing in 3 days.
>
> Additionally, as the reviewer hvZS suggested, we have spent several days conducting additional experiments, which provide a detailed analysis of the distribution mismatch between the pre-training and evaluation stages.
>
> You can find the results here:
> https://openreview.net/forum?id=CiiLchbRe3&noteId=40PEGFrve4
>
> We are happy to update the writing as you suggested and include these additional experimental results in the revised version of the paper.
>
> Please let us know if there is anything else you would like to know. We also hope you can re-evaluate our work based on these updated results.

---

> > ### Comment · Reviewer_8XDh · 2024-11-30
> >
> > Thank you for your reply! I appreciate the clarification and extensive experiments from the authors, and will keep my score which is above the accepting threshold.

---

### Author Response · Authors · 2024-11-26
**Summary to AC**

We thank the reviewers for their time. To save the AC’s time, below we provide a brief summary of the discussion period.

# Our contribution

We developed a uniform pre-training approach that is applicable to a wide variety of decision-making problems. We provide theoretical and empirical results for stable training and effective performance. We believe this is foundational work that provides a comprehensive analysis for applying Transformers to decision-making problems.

To be more specific.
- **Unified Approach.** This approach can solve a wide variety of decision-making problems (using the same pre-training approach for bandit, pricing, and inventory problems) under different environments (with different types of reward functions, such as linear, contextual, and stochastic). By sharing a uniform pre-training structure, this method eliminates the need for users to design instance-specific (e.g., bandit vs. pricing) or environment-specific (e.g., stochastic bandit vs. linear bandit) algorithms, and provides general results for theoretical regrets and similar empirical performance.

- **Power of Transformers to Adapt.** The inherent power of Transformers enables the pretraining of decision-making agents in simulated environments. The Transformer architecture is capable of generalizing and extracting specific patterns from the trajectories it has encountered. Consequently, as long as the simulator is sufficiently large to encompass trajectories similar to those of the current environment, the Transformer can effectively identify the environment and adapt to the corresponding optimal algorithms. This approach introduces a new in-context algorithm selection mechanism that is both powerful and flexible in practice.


# Summary of the reviewers comments

In summary, almost all reviewers (except 8XDh) have had misunderstandings regarding the foundational setup. After the first round of discussions, the main issues remain focused on the contribution of our work to the general decision-making community.

The reviewers' concerns mainly focus on the limitations of the scope of this work, as it primarily focus on simpler environments (bandit, pricing, inventory problems) without random transition dynamics, which is common in the general RL setting.

# Our response regarding the reviewers major concerns

- **Merit of (currently missing) fundamental work.** First, we believe that the effort to focus on a simpler environment should not be downplayed, given that this area lacks a solid foundation. Indeed, every foundational RL or sequential decision-making algorithm needs to be based on foundations for simple environments like bandit problems. For example, consider the advancement from UCB in bandit problems to UCBVI in RL—UCBVI cannot be developed from nowhere without the insights gained from bandit problems.

- **Comprehensive results.** Second, we believe we have shown enough new results to contribute to this community. To the best of our knowledge, in this simple bandit setting, we have not seen theoretical developments comparable to our setting. Our paper contains 40 pages of work with both theoretical and empirical developments, and we think it is reasonable to defer the investigation of RL to future work.

---

### Meta-Review · Area_Chair_V6zH · 2024-12-22

**Metareview:**

This paper presents a supervised training framework for transformers applied to sequential decision-making problems—a subset of reinforcement learning (RL) tasks that lack an explicit transition matrix. In this framework, transformers are pre-trained on the target task class by simulating multiple environments to generate trajectories consisting of states, actions, and rewards. However, a discrepancy exists between action selection during pre-training and at test time, which leads to a gap between the empirical loss observed in training and the expected loss during testing. This gap can cause out-of-distribution issues, as studied by previous literature, causing the regret bound of the model’s performance to increase exponentially over time. To address this, the authors propose a mixed pre-training algorithm, in which the models learn not only from simulated data but also from trajectories generated by their own actions. This hybrid approach reduces the discrepancy between training and testing distributions. The proposed algorithm is theoretically validated and supported by experimental results.

The main concern is on the claimed advantage over standard baselines such as UCB or Thompson sampling: (1) UCB and Thompson sampling can also leverage prior information, and (2) no formal theoretical separation between the proposed approach and UCB/TS. Therefore, the AC recommends rejection.

**Additional Comments On Reviewer Discussion:**

The main concern is on the claimed advantage over standard baselines such as UCB or Thompson sampling: (1) UCB and Thompson sampling can also leverage prior information, and (2) no formal theoretical separation between the proposed approach and UCB/TS. This concerns was not fully addressed during the rebuttal.

---

### Decision · Program_Chairs · 2025-01-22

Reject